# A retrospective cohort study of valproate and infertility in men with epilepsy or bipolar disorder using international health data

Gashirai K. Mbizvo [1,2,3,4] ✉, Susan E. Duncan[5], Lewis Nancarrow[6,7], Tessa Sagarino[8], Lance V. Watkins [9,10,11], Michael T. Mbizvo [12], Gregory Y. H. Lip [1,13,14] & Anthony G. Marson[2,3,4,14]

Valproate is highly effective at treating epilepsy and bipolar disorder. It faces prescribing restrictions in men due to concerns it causes testicular dysfunction and infertility. These mostly stem from animal models – the human evidence is limited and conflicting. We report the largest ever retrospective cohort study of infertility in men with epilepsy or bipolar disorder, using real-world healthcare data from TriNetX. 91,917 of the men are exposed to valproate, and 535,803 unexposed. Cohorts are propensity score matched for a comprehensive set of baseline covariates, and survival analysis is undertaken using Cox-proportional hazards models. No significant difference is seen between valproate-exposed and unexposed men across lifetime risks of infertility, testicular hypofunction, testicular atrophy, and a composite of low sperm concentration, motility, vitality, normal forms, and semen volume ($p > 0.05$). Our findings do not support an association between valproate and infertility in men with epilepsy or bipolar disorder in real-world settings.

Epilepsy, one of the most common neurological disorders globally, affects over 70 million people[1]. Similarly, bipolar disorder represents a significant portion of the global burden of psychiatric disease, impacting more than 40 million individuals[2]. Sodium valproate is a highly effective treatment for both disorders. In epilepsy, valproate is used for all seizure types, and it outperforms any other antiseizure medication (ASM) in idiopathic generalised epilepsy (which affects a third of people with epilepsy (PWE))[3,4]. It is also a cornerstone mood-stabilising treatment for bipolar disorder, where efficacy is second only to lithium[5].

Valproate was the most prescribed ASM globally in the early 2000s[6]. At the same time, it dominated the market of mood-stabilising drugs, particularly in the US[7]. However, widespread evidence subsequently demonstrated foetal congenital malformations (11%) and neurodevelopmental disorders (30–40%) associated with prenatal valproate exposure, leading to guidance changes by 2015–2018[8]. These prohibited valproate prescribing in women of childbearing potential with epilepsy or bipolar disorder unless other treatments had failed. A substantial decline in female valproate prescriptions followed[8–13].

[1]Liverpool Centre for Cardiovascular Science at University of Liverpool, Liverpool John Moores University and Liverpool Heart & Chest Hospital, Liverpool L69 7TX, United Kingdom. [2]Liverpool Interdisciplinary Neuroscience Centre, University of Liverpool, Liverpool, United Kingdom. [3]Pharmacology and Therapeutics, Institute of Systems, Molecular and Integrative Biology, University of Liverpool, Liverpool, United Kingdom. [4]The Walton Centre NHS Foundation Trust, Liverpool, United Kingdom. [5]Muir Maxwell Epilepsy Centre, Centre for Clinical Brain Sciences, The University of Edinburgh, Edinburgh, United Kingdom. [6]Centre for Women's Health Research, Department of Women's and Children's Health, Institute of Life Course and Medical Sciences, University of Liverpool, Member of Liverpool Health Partners, Liverpool, United Kingdom. [7]Hewitt Centre for Reproductive Medicine, Liverpool Women's NHS Foundation Trust, Liverpool, United Kingdom. [8]TriNetX LLC, Cambridge, Massachusetts, USA. [9]University of South Wales, Pontypridd, United Kingdom. [10]Mental Health and Learning Disabilities Service Group, Swansea Bay University Health Board, Swansea Bay University Health Board, Port Talbot, Baglan, United Kingdom. [11]Cornwall Intellectual Disability Equitable Research (CIDER), University of Plymouth Peninsula School of Medicine, Truro, United Kingdom. [12]Reproductive Health Programme, Population Council, Lusaka, Zambia. [13]Danish Centre for Health Services Research, Department of Clinical Medicine, Aalborg University, Aalborg, Denmark. [14]These authors jointly supervised this work: Gregory Y.H. Lip, Anthony G. Marson. ✉e-mail: Gashirai.Mbizvo@liverpool.ac.uk

Valproate remains the first-line ASM for men with newly diagnosed generalised epilepsy in most countries, and its use in these countries remains similarly unrestricted for men with bipolar disorder[3,4,7]. In the UK, regulators recently described considering i) the known risks of valproate, including the risk of impaired male fertility; ii) pre-clinical data on possible transgenerational risks with prenatal exposure, and iii) studies in juvenile and adult animals suggesting adverse effects on the testes[14]. They concluded by observing that there is currently limited data available on many of these risks in humans, and that further studies are required[14]. Nevertheless, they proceed with restricting valproate prescribing for both men and women aged <55 years with epilepsy or bipolar disorder unless other treatments have failed, commencing from January 2024[8,14,15]. Similar restrictions are being considered in other countries[16–19]. Together, they are likely to result in withdrawal of valproate from a substantial number of men already taking it, as has been the case for women[8–13].

The extent to which valproate actually causes impaired male fertility and/or adverse effects on the testes is largely unknown[19,20]. The majority of evidence is either in animal models or on small numbers of humans[19,20]. It is unclear how applicable the results from animal models are to the human experience. Indeed, some of those animal studies have had to use valproate doses seven to 33 times the maximum human-equivalent dose to demonstrate effects such as testicular atrophy and reversible abnormalities in sperm count or motility[19,21–23]. The advice given to humans cannot be based entirely on animal models. High-quality grade I–II human studies are required (Oxford 2011 Levels of Evidence)[24], but are as yet unavailable. Supplementary Table 1 provides a structured overview of the available human studies, outcomes, limitations, and levels of evidence[25–42]. All are observational, spanning grades III–V, with most being case reports or unmatched case-control designs with small numbers of up to 32 valproate-exposed cases and 90 healthy controls. There are no studies using methods from which causality can be more confidently inferred from observational data, such as propensity score matching, inverse probability weighting, or target trial emulation[43–45].

Some of the studies report adverse effects of valproate on the sexual and reproductive health of men, describing lower sperm count or concentration[29–31,39–42], percentage normal sperm[27,30,35,38–42], sperm motility[26,29–31,33,37,39–42], semen volume[35], or testicular volume[30,32]. However, other studies do not report lower sperm count or concentration[32,33,35,37,38], percentage normal sperm[29], sperm motility[35], semen volume[30,33,37,41], or testicular volume[34,37]. Many of the studies focus on blood hormone levels, but there is variation in findings between studies for the same hormones (see in Supplementary Table 1)[25,28–30,32–36,38,42], and hormonal profiles do not necessarily correlate with male sexual health and fertility[20]. Furthermore, epilepsy, in and of itself, can affect fertility rates, which are two-thirds lower in men with epilepsy than without[46]. Therefore, it is difficult to be certain if any reduced fertility seen in men on valproate against healthy controls is related to the valproate or the epilepsy. Indeed, other co-prescribed ASMs may be associated with impairment in semen, testicular, or infertility outcomes[27,28,32,34–38], indicating a need for these to be adjusted for statistically before making inferences about the independent effect of valproate on these outcomes; but such small studies have precluded adjustment[25,26,28–39]. Only one study has directly assessed fertility through birth rate[27]. This retrospective cohort of claims and registry data in Finland is the largest reported, assessing PWE on valproate (overall $n = 1546$, monotherapy $n = 1116$), carbamazepine (overall $n = 2689$, monotherapy $n = 2365$), and oxcarbazepine (overall $n = 832$, monotherapy $n = 631$). Comparisons were made to PWE untreated ($n = 2714–2785$), and people without epilepsy ($n = 13,378–13,689$). Authors concluded that the birth rate was decreased among PWE on ASMs in general, more so in men. Among men with epilepsy, not only valproate, but also carbamazepine and oxcarbazepine were all associated with low birth rate compared to men without epilepsy. However, only oxcarbazepine retained a reduced birth rate amongst treated PWE compared with untreated PWE. In a study of 17 infertile men on valproate who were switched to levetiracetam ($n = 9$) or lamotrigine ($n = 8$), three were subsequently able to conceive[33]. However, conception occurred whilst the men were still on a reducing regimen of valproate, making it difficult to confidently infer an association between valproate and infertility. Finally, there is only one reported study, to our knowledge, of reproductive abnormalities in men on valproate with bipolar disorder[25]. This is reported as a full-length article assessing reproductive hormones in 18 men in Turkey with bipolar disorder on valproate monotherapy or in combination with lithium, 15 men with epilepsy on valproate monotherapy, and 21 men with bipolar disorder on lithium monotherapy[25]. Valproate did not negatively impact reproductive hormones in the men with bipolar disorder. Elevated levels of prolactin and follicle-stimulating hormone (FSH) were found in men with epilepsy. This was attributed to the epilepsy itself by the study authors. They wrote a follow-up Letter to the Editor assessing semen parameters in 12 men from the study who had consented to this testing: six men with bipolar disorder on lithium monotherapy, five men with bipolar disorder on valproate monotherapy or in combination with lithium, and one man with epilepsy on valproate monotherapy[26]. They found reduced sperm count and motility in two out of five men with bipolar disorder on valproate monotherapy or in combination with lithium, and reduced sperm count in the one man with epilepsy on valproate monotherapy. It is challenging to know if any associations, if present, were related to bipolar disorder, epilepsy, lithium, valproate, or a combination of these. A much larger study is needed to help account for these and other potential confounders at baseline, allowing more confident inferences to be drawn about any associations.

We report the first international cohort study of real-world healthcare data assessing infertility outcomes in valproate-exposed vs. unexposed men with epilepsy or bipolar disorder. We studied data from a global network of healthcare organisations (HCOs) held in the TriNetX LIVE platform[47,48]. Clinical data, including disease coding, drug prescribing, and laboratory results, are anonymised and uploaded directly from frontline clinical care onto the TriNetX LIVE platform in real-time. We used a combination of validated disease and prescription codes[49–52] to create three large cohorts of living males aged <55 years within the TriNetX Global Collaborative Network[47,48]: A) men with epilepsy or bipolar disorder; B) men with epilepsy; and C) men with bipolar disorder. Lifetime propensity-score-matched risks of the following outcomes were assessed between those exposed and unexposed to valproate:

i. Male infertility (as defined by the World Health Organisation (WHO) through the International Classification of Diseases 10[th] Revision Clinical Modification (ICD-10-CM) code N46[53],–which includes N46.0 (Azoospermia), N46.1 (Oligospermia), N46.8 (Other male infertility–capturing where male infertility has been identified but the cause does not fit into any of the other specified (coded) aetiological categories)[54], N46.9 (Male infertility, unspecified–capturing where male infertility that has been identified but the cause is unclear)[55], and 606 (Infertility, male–which is the ICD-9-CM code equivalent of N46, allowing healthcare systems using older coding to still be represented)[53];

ii. Testicular hypofunction (ICD-10-CM code E29.1)–which includes defective biosynthesis of testicular androgen, 5-delta-reductase deficiency (with male pseudohermaphroditism), and testicular hypogonadism[56];

iii. Testicular atrophy (ICD-10-CM code N50.0–a code which clinicians have the discretion to use when finding evidence of a pathological reduction in size of the testicles, e.g. using an orchidometer or ultrasound)[57];

iv. A composite measure of laboratory results consisting of low sperm concentration ($< 16 \times 10^6$/ml semen), low sperm motility

( < 42% total motility), low sperm vitality ( < 54% alive), low normal forms ( < 4% of sperm have a normal morphology), and low semen volume ( < 1.4 mL)−thresholds taken from the 2021 WHO Laboratory Manual for the Examination and Processing of Human Semen[58–60];

v. Laboratory results for the following reproductive hormone levels in serum: total testosterone (ng/dL), free testosterone (pg/mL), luteinizing hormone (LH, mIU/mL), FSH (mIU/mL), prolactin (ng/mL), and estradiol (pg/mL)[25,28–30,32–36,38,42]. For hormone profiles, the most recent laboratory result reported during follow-up was used, averaged across the cohort.

The index event was defined as the second recorded valproate prescription in men exposed to the drug, conditional on at least two recorded diagnoses of epilepsy or bipolar disorder. For unexposed men, defined as those with no record of valproate use at any time, the index event was the second recorded diagnosis of epilepsy or bipolar disorder. No restrictions were placed on the use of other ASMs or psychotropic medications in either group, reflecting real-world clinical practice after diagnosis. This approach also allowed inclusion of individuals in the unexposed group who chose not to commence treatment after diagnosis, incorporating real-world variation in clinical decision-making, enhancing the study's generalisability. The rationale for using the second valproate prescription in the exposed group was to ensure repeat exposure, providing a more robust indication of chronic rather than incidental or one-off treatment while also minimising prevalent user bias, which increases as the number of treatment exposures required for inclusion increases[61]. We used the proportion of the cohort with detectable serum valproate levels when last checked during follow-up to infer ongoing exposure and adherence. Requiring two diagnostic codes for epilepsy or bipolar disorder in both groups helped minimise the risk of misdiagnosis by confirming the condition through repeated coding[49]. Additionally, this approach enabled us to more accurately balance cohorts based on diagnosis, as the first recorded epilepsy or bipolar disorder code was then used as a matching covariate in the propensity score matching process (including matching by type of epilepsy using the respective G40.- subgroups, as listed in Supplementary Tables 2–14). This approach was designed to help enhance cohort comparability and reduce the potential for bias from misclassification.

Lifetime follow-up commenced at the index. The comparison groups were propensity score matched for a comprehensive set of baseline covariates before index (see Supplementary Tables 2–14 for a full list of these baseline covariates pre- and post-propensity score matching). Selection of the covariates was primarily informed by prior literature[25–42], and guided by our clinical experience as a team specialising in neurology (G.K.M., S.E.D., A.G.M.), psychiatry (L.V.W.), fertility medicine (L.N.), and andrology (M.T.M.). We used domain-specific knowledge to avoid adjusting for colliders and instrumental variables, focusing on adjusting confounders and competing exposures instead[62,63]. We restricted propensity score matching to covariates measured prior to the index to avoid adjusting for mediators, which can also distort effect estimates[62,63]. The final set of covariates included age, ethnicity, the epilepsy or bipolar disorder themselves (i.e. matching by the first recorded epilepsy or bipolar disease code), type of epilepsy (by ICD-10-CM G40.- subgroup), all other ASMs, antipsychotic and mood-stabilising drugs, other mental, behavioural and neurodevelopmental disorders, dependence on nicotine, cannabis, alcohol, opioids, cocaine, and other psychoactive recreational substances, diseases of the genitourinary system, sexually transmitted infections, congenital malformations (including of the male genital organs) and chromosomal abnormalities, scrotal varices, undescended and ectopic testicles, hypospadias, testicular torsion, malignant neoplasms of the male genital organs, diabetes, cardiovascular diseases, respiratory diseases, acute/chronic renal failure, body mass index

(BMI), hypopituitarism, androgen insensitivity, cystic fibrosis, inguinal hernias and repairs, surgical procedures on the male genital organs, radiotherapy, and other drugs associated with infertility including chemotherapy, oestrogens, androgens, progestogens, testosterones, spironolactone, 5α-reductase inhibitors, alpha blockers, anti-hypertensives, and certain antibiotics. Any variable with a standardised mean difference (SMD) of ≤ 0.1 was considered well-matched[64].

A survival analysis was undertaken on the fully matched cohorts using Cox-proportional hazard models, generating hazard ratios (HRs) with 95% confidence intervals (CIs), and Log-Rank $p$-values with a 0.05 level of significance. This was undertaken for the dichotomous outcomes male infertility, testicular hypofunction, testicular atrophy and the composite measure of low sperm concentration, motility, vitality, normal forms, or semen volume. These dichotomous outcomes were also aggregated into a single overall infertility measure over time in order to understand the effect of valproate exposure over a lifetime and at intervals of 30, 60, 90, 180, 360 days, 2 years, 5 years, and 10 years after initiation. Propensity score matching was undertaken for each interval comparison (see Supplementary Tables 2–10). Those who had experienced the outcome prior to index were excluded, as were the deceased (minimising the effect of competing risks between death and outcome between comparison groups). T-tests were used to compare mean laboratory results for total testosterone, free testosterone, LH, FSH, prolactin, and estradiol, using a 0.05 level of significance. Participants were removed from the analysis (censored) after the last clinical data point in their record. In order to protect patient anonymity, TriNetX shields figures to a value of ≤10 per outcome when there are 10 or fewer participants for the outcome.

We undertook a sensitivity analysis for the cohort of men with epilepsy or bipolar disorder in which we excluded those with key modifiable risk factors for infertility (including chemotherapy, androgens and anabolic steroids, urogenital surgeries, smoking, alcohol and recreational substance misuse (see full list of factors excluded in the methods section). Instead, we propensity score matched them for the non-modifiable risk factors predominantly (including age, ethnicity, and various comorbidities−see full list in Supplementary Table 13). We also undertook a database sensitivity analysis on the capacity to capture the male infertility outcomes in TriNetX. This was done by replicating the main survival analysis in men with epilepsy or bipolar disorder exposed vs. unexposed to drugs known to cause infertility (chemotherapy, androgens, spironolactone, oestrogens−see full list in Supplementary Table 19), expecting to see a significant association with exposure to these agents if the database is sensitive.

## Results

Search One yielded a cohort of 627,720 men with epilepsy or bipolar disorder from a pool of 142 HCOs queried. 87% of the HCOs were based in the US (yielding 553,528 of the men). The remaining 10% were from outside the US (including Brazil, Italy, Spain, Taiwan, and the UK), yielding 63,278 of the men. To protect patient anonymity, TriNetX shielded the names of the remaining countries contributing the final 3% of men, as each had less than three HCOs participating. 1,173,165,286 electronic clinical data points were extracted, from which it was identified that 91,917 (15%) of the men were exposed to valproate, and 535,803 (85%) were unexposed. 59,782 (65%) of those exposed to valproate had two or more epilepsy disease codes (ICD-10-CM G40) in their record, 49,537 (53%) had two or more seizure symptom codes (ICD-10-CM R56), and 37,066 (40%) had two or more bipolar disorder codes (ICD-10-CM F31). Of those unexposed to valproate, 248,357 (53%) had two or more epilepsy disease codes (ICD-10-CM G40), 283,116 (60%) had two or more seizure symptom codes (ICD-10-CM R56), and 123,179 (26%) had two or more bipolar disorder codes (ICD-10-CM F31). These proportions indicate that some patients in the cohorts had both epilepsy and bipolar disorder. Valproate was detectible in the serum of 98% of the 44,886 unmatched men on valproate

who had levels checked (Supplementary Table 15), inferring a good level of drug adherence. The propensity score matching results are shown in Supplementary Tables 2–10. SMDs of ≤ 0.1 were achieved across all covariates after matching, suggesting the cohorts were sufficiently balanced to allow meaningful comparison[64]. Outcomes were compared between 78,971 matched men in each cohort. Their mean ages were 25 ± 12 standard deviations (SD) to 26 ± 13 SD years. Their survival analysis results are summarised in Table 1 (overall infertility measure over time), Table 2 (infertility outcome subgroups) and Table 3 (reproductive hormones). Over a lifetime, 104 more men with epilepsy or bipolar disorder experienced the overall infertility measure with valproate exposure than without, corresponding to a < 1% change in risk, and no significant difference in hazards was observed (HR 0.932; 95% CI 0.849 to 1.024). There were also no significant differences in hazards between the valproate-exposed and unexposed groups at any of the assessed post-index intervals of 30, 60, 90, 180, and 360 days, as well as 2, 5, and 10 years (Table 1). For the subgroups (Table 2), 49 more men with epilepsy or bipolar disorder experienced infertility with valproate exposure than without, 64 more experienced testicular hypofunction, 11 more experienced testicular atrophy, and two more experienced a composite low sperm concentration, motility, vitality, normal forms, or semen volume. These changes in absolute risk were all <1%, and no significant difference in hazards was observed: male infertility HR 1.057; 95% CI 0.864 to 1.295, testicular hypofunction HR 0.916; 95% CI 0.824 to 1.019, testicular atrophy HR 1.000; 95% CI 0.682 to 1.465, and composite low sperm concentration, motility, vitality, normal forms, or semen volume HR 0.856; 95% CI 0.613 to 1.196. Mean serum levels for total testosterone were significantly less in those exposed to valproate (365 ± 272 SD ng/dL) compared to those unexposed to valproate (387 ± 252 SD ng/dL, $p = 0.002$). However, both results remained within a normal range for this test (300 to 1,000 ng/dL)[65]. Free testosterone was slightly higher with valproate exposure (93 ± 357 pg/mL) compared to without (74 ± 87 pg/mL, $p = 0.046$), but both remained within normal range (66 to 309 pg/mL)[66]. There were no differences in LH, FSH, prolactin, or estradiol levels between those exposed and unexposed to valproate ($p > 0.05$).

Search Two yielded a cohort of 460,844 men with epilepsy exposed ($n = 60,860$) and unexposed to valproate ($n = 399,984$). Valproate was detectable in the serum of 97% of the 29,619 unmatched men on valproate who had levels checked (Supplementary Table 16), inferring a good level of drug adherence. The propensity score matching results are shown in Supplementary Table 11. SMDs < 0.1 were achieved across all covariates after matching. Outcomes were compared between 50,102 matched men in each cohort, with mean ages of 22 ± 13 SD to 23 ± 14 SD years. Results are summarised in Tables 2 and 3. In absolute terms, 30 more men with epilepsy experienced infertility with valproate exposure than without, 41 more experienced testicular hypofunction, six more experienced testicular atrophy, and six less experienced a composite low sperm concentration, motility, vitality, normal forms, or semen volume. These changes in absolute risk were all <1%, and no significant difference in hazards was observed: male infertility HR 1.099; 95% CI 0.826 to 1.462, testicular hypofunction HR 0.922; 95% CI 0.792 to 1.073, testicular atrophy HR 0.985; 95% CI 0.601 to 1.613, and composite low sperm concentration, motility, vitality, normal forms, or semen volume HR 0.633; 95% CI 0.366 to 1.096. Mean serum levels for total testosterone were significantly less in those exposed to valproate (338 ± 260 SD ng/dL) compared to those unexposed to valproate (385 ± 262 SD ng/dL, $p = 0.000$). However, both results remained within a normal range for this test (300 to 1000 ng/dL)[65]. There were no differences in free testosterone, LH, FSH, prolactin, or estradiol levels between those exposed and unexposed to valproate ($p > 0.05$).

Search Three yielded a cohort of 157,331 men with bipolar disorder exposed ($n = 33,450$) and unexposed to valproate ($n = 123,881$). Valproate was detectable in the serum of 94% of the 17,564 unmatched men on valproate who had levels checked (Supplementary Table 17). The propensity score matching results are shown in Supplementary Table 12. SMDs < 0.1 were achieved across all covariates after matching. Outcomes were compared between 22,826 matched men in each cohort, with mean ages of 31 ± 9 SD years in both. Results are summarised in Tables 2 and 3. In absolute terms, 17 less men with bipolar disorder experienced infertility with valproate exposure than without, one more experienced testicular hypofunction, and ≤ 10 and 12 men exposed and unexposed to valproate, respectively, experienced testicular atrophy. 18 men in both groups experienced a composite low sperm concentration, motility, vitality, normal forms, or semen volume. These changes in absolute risk were all < 1%, and no significant difference in hazards was observed: male infertility HR 0.721; 95% CI 0.510 to 1.019, testicular hypofunction HR 0.943; 95% CI 0.793 to 1.121, testicular atrophy HR 0.703; 95% CI 0.296 to 1.669, and composite low sperm concentration, motility, vitality, normal forms, or semen volume HR 0.948; 95% CI 0.493 to 1.823. There were no differences in total or free testosterone, LH, FSH, prolactin, or estradiol levels between those exposed and unexposed to valproate ($p > 0.05$).

Our sensitivity analysis Search Four yielded a cohort of 293,997 men with epilepsy or bipolar disorder after excluding those with key modifiable risk factors for male infertility, including androgens, chemotherapy, urogenital surgery, and lifestyle factors such as smoking and excess alcohol. Following this, there were 32,885 and 261,112 men exposed and unexposed to valproate, respectively. Valproate was detectable in the serum of 97% of the 16,820 unmatched men on valproate who had levels checked (Supplementary Table 18). The propensity score matching results are shown in Supplementary Table 13. SMDs <0.1 were achieved across all covariates after matching. Outcomes were compared between 27,946 matched men in each cohort, with mean ages of 20 ± 12 SD years to 21 ± 13 SD years. Results are summarised in Tables 4 (infertility outcomes) and 5 (reproductive hormones). There remained no significant differences between men with epilepsy or bipolar disorder who were exposed and unexposed to valproate in terms of the overall infertility measure (HR 0.910; 95% CI 0.680 to 1.218), male infertility (HR 1.042; 95% CI 0.636 to 1.709), testicular hypofunction (HR 0.775; 95% CI 0.527 to 1.142), testicular atrophy (HR 0.906; 95% CI 0.304 to 2.698), and low sperm concentration, motility, vitality, normal forms, or semen volume (HR 3.717; 95% CI 0.802 to 17.221). There were also no differences in total or free testosterone, LH, FSH, prolactin, or estradiol levels between those exposed and unexposed to valproate with bipolar disorder ($p > 0.05$).

The database sensitivity analysis Search Five identified a cohort of 525,802 men with epilepsy or bipolar disorder exposed ($n = 25,173$) and unexposed ($n = 500,629$) to drugs known to cause infertility. The propensity score matching results are shown in Supplementary Table 14. SMDs < 0.1 were achieved across all covariates after matching. Outcomes were compared between 23,728 matched men in each cohort, with mean ages of 29 ± 14 SD to 30 ± 14 SD years. Results are summarised in Supplementary Table 20 and Supplementary Fig. 1. The database was able to detect a significant increase in all male infertility outcomes assessed in association with exposure to these drugs: overall infertility measure HR 5.202; 95% CI 4.534 to 5.968, male infertility HR 2.980; 95% CI 2.192 to 4.053, testicular hypofunction HR 6.272; 95% CI 5.358 to 7.342, testicular atrophy HR 2.401; 95% CI 1.424 to 4.050, and low sperm concentration, motility, vitality, normal forms, or semen volume HR 1.822; 95% CI 1.086 to 3.057.

## Discussion

Although widespread warnings are given to men with epilepsy or bipolar disorder that taking valproate is associated with infertility[14], the evidence surrounding this is in studies of animals or small numbers of humans, with important confounders unaccounted for, such as the epilepsy or bipolar disorder themselves, other ASMs and psychotropic medications, wider comorbidity, lifestyle and treatments[19,20,25–42].

**Table 1 | Overall infertility measure over time in men with epilepsy or bipolar disorder exposed and unexposed to valproate**

| Outcome | Outcome assessment period after index | Cohort | Patient count before matching | Patient count after matching | Patients excluded because they had the outcome prior to follow-up | Remaining patient denominator analysed | Patients with outcome | Risk | Hazard ratio (95% CI) | Log-Rank p-value |
|---|---|---|---|---|---|---|---|---|---|---|
| Composite outcome: -male infertility[a] -testicular hypofunction[b] or atrophy[c] -low sperm concentration, motility, vitality, normal forms, or semen volume[d] | Lifetime | Epilepsy or bipolar disorder exposed to valproate | 91,917 | 78,971 | 617 | 78,354 | 942 | 1.2% | 0.932 (0.849–1.024) | 0.142 |
| | | Epilepsy or bipolar disorder not exposed to valproate | 535,803 | 78,971 | 675 | 78,296 | 838 | 1.1% | | |
| | 30 days | Epilepsy or bipolar disorder exposed to valproate | 90,571 | 77,826 | 621 | 77,205 | 30 | 0.0% | 0.741 (0.461–1.189) | 0.213 |
| | | Epilepsy or bipolar disorder not exposed to valproate | 524,095 | 77,826 | 638 | 77,188 | 40 | 0.1% | | |
| | 60 days | Epilepsy not exposed to valproate | 89,759 | 77,099 | 600 | 76,499 | 48 | 0.1% | 0.703 (0.485–1.018) | 0.061 |
| | | Bipolar disorder exposed to valproate | 514,623 | 77,099 | 619 | 76,480 | 67 | 0.1% | | |
| | 90 days | Bipolar disorder not exposed to valproate | 88,702 | 76,278 | 583 | 75,695 | 73 | 0.1% | 0.848 (0.620– 1.160) | 0.302 |
| | | Epilepsy or bipolar disorder exposed to valproate | 513,666 | 76,278 | 653 | 75,625 | 84 | 0.1% | | |
| | 180 days | Epilepsy or bipolar disorder not exposed to valproate | 88,687 | 75,985 | 590 | 75,395 | 131 | 0.2% | 0.951 (0.747– 1.210) | 0.681 |
| | | Epilepsy exposed to valproate | 514,702 | 75,985 | 638 | 75,347 | 133 | 0.2% | | |
| | 360 days | Epilepsy not exposed to valproate | 84,068 | 71,327 | 511 | 70,816 | 191 | 0.3% | 0.910 (0.746, 1.110) | 0.353 |
| | | Bipolar disorder exposed to valproate | 496,609 | 71,327 | 518 | 70,809 | 199 | 0.3% | | |
| | 2 years | Bipolar disorder not exposed to valproate | 83,222 | 70,346 | 532 | 70,346 | 333 | 0.5% | 0.919 (0.790–1.069) | 0.272 |
| | | Epilepsy or bipolar disorder exposed to valproate | 478,058 | 70,328 | 550 | 70,328 | 338 | 0.5% | | |

**Table 1 (continued) | Overall infertility measure over time in men with epilepsy or bipolar disorder exposed and unexposed to valproate**

| Outcome | Outcome assessment period after index | Cohort | Patient count before matching | Patient count after matching | Patients excluded because they had the outcome prior to follow-up | Remaining patient denominator analysed | Patients with outcome | Risk | Hazard ratio (95% CI) | Log-Rank p-value |
|---|---|---|---|---|---|---|---|---|---|---|
| | 5 years | Epilepsy or bipolar disorder not exposed to valproate | 87,720 | 75,233 | 583 | 74,650 | 633 | 0.8% | 0.949 (0.848–1.061) | 0.358 |
| | | Epilepsy exposed to valproate | 503,955 | 75,233 | 600 | 74,633 | 596 | 0.8% | | |
| | 10 years | Epilepsy not exposed to valproate | 82,046 | 70,425 | 527 | 69,898 | 747 | 1.1% | 0.942 (0.849–1.045) | 0.261 |
| | | Bipolar disorder exposed to valproate | 469,361 | 70,425 | 570 | 69,855 | 677 | 1.0% | | |

a **Male infertility (as defined by the World Health Organisation (WHO) through the International Classification of Diseases 10th Revision Clinical Modification (ICD-10-CM) code N46**—which includes N46.1 (Oligospermia), N46.8 (Other male infertility—capturing where male infertility has been identified but the cause does not fit into any of the other specified (coded) aetiological categories), N46.9 (Male infertility, unspecified—capturing where male infertility that has been identified but the cause is unclear)[55], and 606 (Infertility, male—which is the ICD-9-CM code equivalent of N46, allowing healthcare systems using older coding to still be represented).

b **Testicular hypofunction (ICD-10-CM code E29.1)**—which includes defective biosynthesis of testicular androgen, 5-delta-reductase deficiency (with male pseudohermaphroditism), and testicular hypogonadism.

c **Testicular atrophy (ICD-10-CM code N50.0)**—a code which clinicians have the discretion to use when finding evidence of a pathological reduction in size of the testicles, e.g., using an orchidometer or ultrasound).

d **Low sperm** concentration (<16 ×10$^6$/ml semen semen), low sperm motility (< 42% total motility), low sperm vitality (<54% viable), low normal forms (< 4% of sperm have a normal morphology), or low semen volume (<1.4 mL). CI = Confidence interval.

**N.B:** Statistics = Survival analysis using Cox-proportional hazard models with 95% CIs and two-sided Log-Rank p-values with a 0.05 level of significance. Slight variation in overall patient count denominators over time reflects the fact that each analysis was conducted as an independent query on a live, dynamic dataset, where patient inclusion can vary slightly due to real-time clinical updates, such as newly recorded diagnoses, additional data accrual, or changes in data completeness across healthcare network sites. Overall trends and conclusions have not been impacted.

In this large cohort study of men with epilepsy or bipolar disorder exposed and unexposed to valproate, adjustment is made for these and many other baseline confounders to make the groups comparable. Subsequently, there is minimal evidence to suggest an association between valproate and male infertility. The results suggest that if there are worsening infertility outcomes in men on valproate in these disorders, the magnitude of effect is small (<1% difference to being unexposed to valproate). Reproductive hormone levels do not appear to fall outside of a normal range, whether the men are exposed or unexposed to valproate.

While the absolute risk of abnormal semen parameters in our study may appear low, this reflects the fact that our estimates are based on the entire population at risk—that is, all men with epilepsy or bipolar disorder are included in the denominator. Naturally, this yields a low proportion of abnormal results across the full cohort. However, this conservative approach enables standardised comparisons between exposure groups and facilitates detection of relative differences in risk, where present. To aid interpretation, we have contextualised our findings using global estimates that similarly adopt a population-at-risk approach[67]. When aligned to a comparable scale, the observed lifetime risk of the overall infertility measure in our study is 1202 per 100,000 in the valproate-exposed group and 1070 per 100,000 in the unexposed group (equivalent to the 1.20% and 1.07% reported in Table 1). These figures are broadly comparable to the worldwide male infertility prevalence reported in the Global Burden of Disease Study as 1820.6 per 100,000, or 1.8%[67].

Our results have important implications for helping men make informed choices about their treatment. Reproductive health is a major determinant of quality of life in men, and over 40% do not take their prescribed medication because of fear of side effects[20,68]. This leads to treatment failure, which, in either epilepsy or bipolar disorder, may lead to hospitalisation, injury or death[20,69,70]. Furthermore, lack of treatment adherence results in increased (and perhaps unnecessary) polypharmacy, which then leads to more side effects[20]. These factors make it crucial that clinicians are provided with sufficient evidence to rationalise the advice they provide to men about side effects. Although clinicians are advised to warn men that valproate is linked to infertility[14], the evidence surrounding this is limited and contradictory[25–42]. Our results suggest that more caution is needed when advising men on valproate about this side effect in either epilepsy or bipolar disorder. Prior studies linking valproate to male infertility in epilepsy mainly used healthy men without epilepsy as controls[27–30,32,34–38]. However, the decision men are faced with is whether to accept valproate or another treatment for their epilepsy or bipolar disorder. Therefore, pragmatic evidence that is directly relevant to this decision is needed, meaning comparisons should be between drug-exposed and unexposed men with the disease, as was reported in the current study. In such circumstances, the evidence we present suggests valproate may not be linked to infertility over and above other ASMs or mood-stabilisers, nor over the epilepsy or bipolar disorder themselves. This is consistent with findings in the largest previous study of valproate and male fertility. This was a national study of Finnish health data in which 1116 PWE on valproate were compared to 13,378–13,689 healthy people without epilepsy and 2714–2785 people with epilepsy that was untreated, adjusted for age and marital status[27]. Birth rate was lower in men with epilepsy on valproate (and those on carbamazepine or oxcarbazepine) when compared to men without epilepsy. There was no difference in birth rate between men with epilepsy on valproate (or carbamazepine) and men with epilepsy who were untreated, illustrating the substantial impact epilepsy has as a confounder. Only oxcarbazepine was linked to reduced birth rate amongst treated PWE compared with untreated PWE[27].

The predominant use of healthy controls in many of the prior studies is likely to be one of the reasons our study results differ from many of those prior studies[28–30,32,34–38]. Additional reasons include our

**Table 2 | Infertility outcome subgroups in men with epilepsy or bipolar disorder exposed and unexposed to valproate**

| Outcome | Cohort | Patient count before matching | Patient count after matching | Patients excluded because they had the outcome prior to follow-up | Remaining patient denominator analysed | Patients with outcome | Risk | Hazard ratio (95% CI) | Log-Rank p-value |
|---|---|---|---|---|---|---|---|---|---|
| Male infertility | Epilepsy or bipolar disorder exposed to valproate | 91,917 | 78,971 | 137 | 78,834 | 216 | 0.3% | 1.057 (0.864– 1.295) | 0.589 |
| | Epilepsy or bipolar disorder not exposed to valproate | 535,803 | 78,971 | 136 | 78,835 | 167 | 0.2% | | |
| | Epilepsy exposed to valproate | 60,860 | 50,102 | 58 | 50,044 | 112 | 0.2% | 1.099 (0.826– 1.462) | 0.517 |
| | Epilepsy not exposed to valproate | 399,984 | 50,102 | 76 | 50,026 | 82 | 0.2% | | |
| | Bipolar disorder exposed to valproate | 33,450 | 22,826 | 60 | 22,766 | 57 | 0.3% | 0.721 (0.510– 1.019) | 0.062 |
| | Bipolar disorder not exposed to valproate | 123,881 | 22,826 | 58 | 22,768 | 74 | 0.3% | | |
| Testicular hypofunction | Epilepsy or bipolar disorder exposed to valproate | 91,917 | 78,971 | 460 | 78,511 | 715 | 0.9% | 0.916 (0.824–1.019) | 0.107 |
| | Epilepsy or bipolar disorder not exposed to valproate | 535,803 | 78,971 | 510 | 78,461 | 651 | 0.8% | | |
| | Epilepsy exposed to valproate | 60,860 | 50,102 | 207 | 49,895 | 355 | 0.7% | 0.922 (0.792–1.073) | 0.294 |
| | Epilepsy not exposed to valproate | 399,984 | 50,102 | 244 | 49,858 | 314 | 0.6% | | |
| | Bipolar disorder exposed to valproate | 33,450 | 22,826 | 229 | 22,597 | 258 | 1.1% | 0.943 (0.793– 1.121) | 0.503 |
| | Bipolar disorder not exposed to valproate | 123,881 | 22,826 | 217 | 22,609 | 257 | 1.1% | | |
| Testicular atrophy | Epilepsy or bipolar disorder exposed to valproate | 91,917 | 78,971 | 32 | 78,939 | 59 | 0.1% | 1.000 (0.682–1.465) | 0.998 |
| | Epilepsy or bipolar disorder not exposed to valproate | 535,803 | 78,971 | 39 | 78,932 | 48 | 0.1% | | |
| | Epilepsy exposed to valproate | 60,860 | 50,102 | 16 | 50,086 | 35 | 0.1% | 0.985 (0.601-1.613) | 0.951 |
| | Epilepsy not exposed to valproate | 399,984 | 50,102 | 26 | 50,076 | 29 | 0.1% | | |
| | Bipolar disorder exposed to valproate | 33,450 | 22,826 | 13 | 22,813 | ≤10^ | 0.0% | 0.703 (0.296– 1.669) | 0.422 |
| | Bipolar disorder not exposed to valproate | 123,881 | 22,826 | 12 | 22,814 | 12 | 0.1% | | |

**Table 2 (continued) | Infertility outcome subgroups in men with epilepsy or bipolar disorder exposed and unexposed to valproate**

| Outcome | Cohort | Patient count before matching | Patient count after matching | Patients excluded because they had the outcome prior to follow-up | Remaining patient denominator analysed | Patients with outcome | Risk | Hazard ratio (95% CI) | Log-Rank p-value |
|---|---|---|---|---|---|---|---|---|---|
| Low sperm concentration, motility, vitality, normal forms, or semen volume | Epilepsy or bipolar disorder exposed to valproate | 91,917 | 78,971 | 45 | 78,926 | 70 | 0.1% | 0.856 (0.613–1.196) | 0.362 |
| | Epilepsy or bipolar disorder not exposed to valproate | 535,803 | 78,971 | 59 | 78,912 | 68 | 0.1% | | |
| | Epilepsy exposed to valproate | 60,860 | 50,102 | 19 | 50,083 | 23 | 0.0% | 0.633 (0.366–1.096) | 0.099 |
| | Epilepsy not exposed to valproate | 399,984 | 50,102 | 19 | 50,083 | 29 | 0.1% | | |
| | Bipolar disorder exposed to valproate | 33,450 | 22,826 | 14 | 22,812 | 18 | 0.1% | 0.948 (0.493–1.823) | 0.250 |
| | Bipolar disorder not exposed to valproate | 123,881 | 22,826 | 20 | 22,806 | 18 | 0.1% | | |

**Legend:** low sperm concentration (< 16 × 10⁶/ml semen semen), low sperm motility (< 42% total motility), low sperm vitality (< 54% viable), low normal forms (< 4% of sperm have a normal morphology), or low semen volume (< 1.4 mL). CI (Confidence interval); ^ = In order to protect patient anonymity, TriNetX shield figures to a value of ≤10 per outcome when there are 10 or fewer participants for the outcome.
**N.B.:** Statistics = Survival analysis using Cox-proportional hazard models with 95% CIs and two-sided Log-Rank p-values with a 0.05 level of significance. An expected mismatch is noted between the total number of patients with epilepsy and/or bipolar disorder and the sum of the number of patients in each subgroup. Since cohorts were not designed to be mutually exclusive, this mismatch is due to the ability for patients to have a record of one or both diagnoses (epilepsy and bipolar disorder) in their electronic health record.

study being much larger than the prior studies[25,26,28–42], allowing us to propensity score match cohorts on a comprehensive set of confounders, many of which were either unmeasured or excluded in prior studies (see Supplementary Table 1). This will have decreased the chances of an overestimation or underestimation of the real-world effect of valproate in our study when compared to prior studies. To help illustrate the dangers of drawing inferences from incompletely or poorly matched samples, we conducted a post-hoc analysis in which we re-assessed the relationship between valproate exposure and our overall infertility measure (defined as experiencing at least one of male infertility, testicular hypofunction, testicular atrophy, or a composite of low sperm concentration, motility, vitality, normal forms, or semen volume). In this illustrative analysis, no matching was applied – cohorts were left completely unmatched. Under these conditions, valproate exposure appeared significantly associated with the overall infertility measure: 954 of 85,625 exposed men experienced the outcome, compared to 4382 of 502,095 unexposed men, representing a 0.2% absolute increase in the exposed group (HR 1.151; 95% CI 1.073 to 1.234; Log-Rank $p < 0.0001$). However, this apparent association disappears once the imbalances in baseline characteristics between cohorts are addressed through propensity score matching, as in the main analysis, where SMDs of ≤ 0.1 were achieved across all covariates, making the groups more comparable[64,71]. Indeed, our study's use of a wide range of relevant comorbidity, treatment and lifestyle conditions as propensity-score-matched covariates (as opposed to excluding them from the study – as was done in most prior studies)[25,29,30,32–38] will have increased the real-world generalisability of our study results to frontline clinical settings. This is because many men seeking counselling about valproate use are not without some prior comorbidity, treatment or lifestyle conditions potentially affecting their fertility. The applicability of conclusions drawn from the prior studies excluding such men is thus limited. Nevertheless, we were also able to undertake a sensitivity analysis in which we also excluded men with a prior history of key modifiable treatment and lifestyle risk factors for infertility. This did not alter our overall conclusions. A recently published systematic review was able to undertake a meta-analysis across five out of the 19 human studies we identified in Supplementary Table 1, applying a random effect model to men with epilepsy on valproate against healthy controls for sperm count, motility, normal percentage of sperm, FSH and LH levels[72]. Sperm motility alone was significantly reduced in men on valproate (overall SMD = −1.62, 95% CI: −2.81 to −0.43, $p = 0.033$). There were no significant differences in sperm count (overall SMD = −0.78, 95% CI: −1.58 to 0.03, $p = 0.286$), percentage of abnormal sperm (overall SMD = 0.93, 95% CI: −0.97 to 2.84, $p = 0.616$), levels of FSH (overall SMD = −1.32, 95% CI: −2.93 to 0.29, $p = 0.198$) and LH (overall SMD = −0.96, 95% CI: −1.95 to 0.04, $p = 0.211$). Interestingly, the authors concluded that men with epilepsy on valproate exhibit abnormal male fertility factors, including sperm count, motility, percentage of abnormal sperm, FSH and LH. However, we note that their findings did not actually support a statistically significant change in any of these factors beyond motility.

Our study is strengthened by being an analysis of real-world healthcare data extracted directly from clinical care[47,48], and it is enriched by laboratory results on semen parameters and serum valproate levels, which are usually unavailable for such health data studies. The number of men aged < 55 years with laboratory results available on semen parameters in TriNetX is 65,628 for semen volume, 46,319 for sperm motility, 38,589 for sperm concentration, 3419 for sperm vitality, and 3239 for normal forms. Serum valproate levels are available for 135,585 men aged < 55 years. The study is further strengthened by its large size and the ability to adjust for a comprehensive set of baseline covariates. Our use of propensity score matching was also a strength, as this method can emulate the effect of randomisation by matching individuals with similar probabilities of treatment based on their observed baseline characteristics[71]. Future work should seek to

**Table 3 | Reproductive hormone levels in men with epilepsy or bipolar disorder exposed and unexposed to valproate**

| Outcome | Cohort | Patient count before matching | Patient count after matching | Patients with sex hormone results | Mean ± SD | p-value |
|---|---|---|---|---|---|---|
| Total testosterone (ng/dL) | Epilepsy or bipolar disorder exposed to valproate | 91,917 | 78,971 | 2716 | 365 ± 272 | 0.002** |
| | Epilepsy or bipolar disorder not exposed to valproate | 535,803 | 78,971 | 2747 | 387 ± 252 | |
| | Epilepsy exposed to valproate | 60,860 | 50,102 | 1499 | 338 ± 260 | 0.000** |
| | Epilepsy not exposed to valproate | 399,984 | 50,102 | 1433 | 385 ± 262 | |
| | Bipolar disorder exposed to valproate | 33,450 | 22,826 | 890 | 386 ± 278 | 0.109 |
| | Bipolar disorder not exposed to valproate | 123,881 | 22,826 | 1064 | 405 ± 258 | |
| Free testosterone (pg/mL) | Epilepsy or bipolar disorder exposed to valproate | 91,917 | 78,971 | 1532 | 93 ± 357 | 0.046** |
| | Epilepsy or bipolar disorder not exposed to valproate | 535,803 | 78,971 | 1530 | 74 ± 87 | |
| | Epilepsy exposed to valproate | 60,860 | 50,102 | 786 | 93 ± 433 | 0.070 |
| | Epilepsy not exposed to valproate | 399,984 | 50,102 | 774 | 64 ± 55 | |
| | Bipolar disorder exposed to valproate | 33,450 | 22,826 | 550 | 92 ± 283 | 0.799 |
| | Bipolar disorder not exposed to valproate | 123,881 | 22,826 | 675 | 88 ± 183 | |
| Luteinizing hormone (mIU/mL) | Epilepsy or bipolar disorder exposed to valproate | 91,917 | 78,971 | 1198 | 5 ± 5 | 0.160 |
| | Epilepsy or bipolar disorder not exposed to valproate | 535,803 | 78,971 | 998 | 5 ± 6 | |
| | Epilepsy exposed to valproate | 60,860 | 50,102 | 804 | 5 ± 5 | 0.615 |
| | Epilepsy not exposed to valproate | 399,984 | 50,102 | 632 | 5 ± 5 | |
| | Bipolar disorder exposed to valproate | 33,450 | 22,826 | 274 | 5 ± 6 | 0.825 |
| | Bipolar disorder not exposed to valproate | 123,881 | 22,826 | 287 | 5 ± 4 | |
| Follicle-stimulating hormone (mIU/mL) | Epilepsy or bipolar disorder exposed to valproate | 91,917 | 78,971 | 1182 | 6 ± 9 | 0.320 |
| | Epilepsy or bipolar disorder not exposed to valproate | 535,803 | 78,971 | 952 | 7 ± 10 | |
| | Epilepsy exposed to valproate | 60,860 | 50,102 | 809 | 7 ± 10 | 0.314 |
| | Epilepsy not exposed to valproate | 399,984 | 50,102 | 618 | 6 ± 9 | |
| | Bipolar disorder exposed to valproate | 33,450 | 22,826 | 262 | 6 ± 8 | 0.696 |
| | Bipolar disorder not exposed to valproate | 123,881 | 22,826 | 262 | 6 ± 7 | |
| Prolactin (ng/mL) | Epilepsy or bipolar disorder exposed to valproate | 91,917 | 78,971 | 3034 | 25 ± 194 | 0.118 |
| | Epilepsy or bipolar disorder not exposed to valproate | 535,803 | 78,971 | 2266 | 19 ± 30 | |
| | Epilepsy exposed to valproate | 60,860 | 50,102 | 1853 | 28 ± 248 | 0.151 |
| | Epilepsy not exposed to valproate | 399,984 | 50,102 | 1356 | 19 ± 22 | |
| | Bipolar disorder exposed to valproate | 33,450 | 22,826 | 922 | 26 ± 118 | 0.073 |
| | Bipolar disorder not exposed to valproate | 123,881 | 22,826 | 777 | 18 ± 20 | |
| Estradiol (pg/mL) | Epilepsy or bipolar disorder exposed to valproate | 91,917 | 78,971 | 361 | 65 ± 152 | 0.258 |
| | Epilepsy or bipolar disorder not exposed to valproate | 535,803 | 78,971 | 379 | 86 ± 319 | |
| | Epilepsy exposed to valproate | 60,860 | 50,102 | 180 | 57 ± 149 | 0.505 |
| | Epilepsy not exposed to valproate | 399,984 | 50,102 | 203 | 66 ± 126 | |
| | Bipolar disorder exposed to valproate | 33,450 | 22,826 | 142 | 63 ± 100 | 0.300 |
| | Epilepsy not exposed to valproate | 123,881 | 22,826 | 184 | 75 ± 112 | |

**Legend: ** = Statistically significant result between valproate-exposed and unexposed men (p < 0.05) using two-sided t-test statistics. P-values are unadjusted for multiple comparisons. However, with Bonferroni correction applied across 18 comparisons, only p-values < 0.0028 would be considered statistically significant. SD = Standard deviation.

**N.B:** An expected mismatch is noted between the total number of patients with epilepsy and/or bipolar disorder and the sum of the number of patients in each subgroup. Since cohorts were not designed to be mutually exclusive, this mismatch is due to the ability for patients to have a record of one or both diagnoses (epilepsy and bipolar disorder) in their electronic health record.

**Table 4 | Infertility outcomes in men with epilepsy and or bipolar disorder exposed and unexposed to valproate with key modifiable risk factors excluded**

| Outcome | Cohort | Patient count before matching | Patient count after matching | Patients excluded because they had the outcome prior to follow-up | Remaining patient denominator analysed | Patients with outcome | Risk | Hazard ratio (95% CI) | Log-Rank p-value |
|---|---|---|---|---|---|---|---|---|---|
| Overall infertility measure (composite outcome): -male infertility[a] -testicular hypofunction[b] or atrophy[c] -low sperm concentration, motility, vitality, normal forms, or semen volume[d] | Epilepsy or bipolar disorder exposed to valproate | 32,885 | 27,946 | 50 | 27,896 | 96 | 0.3% | 0.910 (0.680–1.218) | 0.526 |
| | Epilepsy or bipolar disorder not exposed to valproate | 261,112 | 27,946 | 63 | 27,883 | 86 | 0.3% | | |
| Male infertility | Epilepsy or bipolar disorder exposed to valproate | 32,885 | 27,946 | 14 | 27,932 | 36 | 0.1% | 1.042 (0.636–1.709) | 0.869 |
| | Epilepsy or bipolar disorder not exposed to valproate | 261,112 | 27,946 | 15 | 27,931 | 28 | 0.1% | | |
| Testicular hypofunction | Epilepsy or bipolar disorder exposed to valproate | 32,885 | 27,946 | 31 | 27,915 | 50 | 0.2% | 0.775 (0.527–1.142) | 0.196 |
| | Epilepsy or bipolar disorder not exposed to valproate | 261,112 | 27,946 | 42 | 27,904 | 53 | 0.2% | | |
| Testicular atrophy | Epilepsy or bipolar disorder exposed to valproate | 32,885 | 27,946 | 10 | 27,944 | ≤10^ | 0.04% | 0.906 (0.304–2.698) | 0.859 |
| | Epilepsy or bipolar disorder not exposed to valproate | 261,112 | 27,946 | 10 | 27,941 | ≤10^ | 0.04% | | |
| Low sperm concentration, motility, vitality, normal forms, or semen volume | Epilepsy or bipolar disorder exposed to valproate | 32,885 | 27,946 | 10 | 27,940 | ≤10^ | 0.04% | 3.717 (0.802–17.221) | 0.072 |
| | Epilepsy or bipolar disorder not exposed to valproate | 261,112 | 27,946 | 10 | 27,938 | ≤10^ | 0.04% | | |

[a] **Male infertility (as defined by the World Health Organisation (WHO) through the International Classification of Diseases 10th Revision Clinical Modification (ICD-10-CM) code N46**—which includes N46.0 (Azoospermia), N46.1 (Oligospermia), N46.8 (Other male infertility—capturing where male infertility has been identified but the cause does not fit into any of the other specified (coded) aetiological categories), N46.9 (Male infertility, unspecified—capturing where male infertility that has been identified but the cause is unclear)[55], and 606 (Infertility, male—which is the ICD-9-CM code equivalent of N46, allowing healthcare systems using older coding to still be represented).

[b] **Testicular hypofunction (ICD-10-CM code E29.1)**—which includes defective biosynthesis of testicular androgen, 5-delta-reductase deficiency (with male pseudohermaphroditism), and testicular hypogonadism.

[c] **Testicular atrophy (ICD-10-CM code N50.0)**—a code which clinicians have the discretion to use when finding evidence of a pathological reduction in size of the testicles, e.g., using an orchidometer or ultrasound).

[d] **Low sperm** concentration (<16×10⁶/ml semen semen), low sperm motility (<42% total motility), low sperm vitality (<54% viable), low normal forms (<4% of sperm have a normal morphology), or low semen volume (<1.4 mL).

CI = Confidence interval. ^ = In order to protect patient anonymity, TriNetX shields figures to a value of ≤10 per outcome when there are 10 or fewer participants for the outcome.

**N.B.:** Statistics = Survival analysis using Cox-proportional hazard models with 95% CIs and two-sided Log-Rank p-values with a 0.05 level of significance.

**Table 5 | Reproductive hormone levels in men with epilepsy or bipolar disorder, exposed and unexposed to valproate, after excluding those with key modifiable risk factors for infertility**

| Outcome | Cohort | Patient count before matching | Patient count after matching | Patients with sex hormone results | Mean ±SD | p-value |
|---|---|---|---|---|---|---|
| Total testosterone (ng/dL) | Epilepsy or bipolar disorder exposed to valproate | 32,885 | 27,946 | 355 | 358 ± 268 | 0.609 |
| | Epilepsy or bipolar disorder not exposed to valproate | 261,112 | 27,946 | 354 | 367 ± 233 | |
| Free testosterone (pg/mL) | Epilepsy or bipolar disorder exposed to valproate | 32,885 | 27,946 | 158 | 63 ± 66 | 0.258 |
| | Epilepsy or bipolar disorder not exposed to valproate | 261,112 | 27,946 | 199 | 70 ± 51 | |
| Luteinizing hormone (mIU/mL) | Epilepsy or bipolar disorder exposed to valproate | 32,885 | 27,946 | 198 | 4 ± 4 | 0.355 |
| | Epilepsy or bipolar disorder not exposed to valproate | 261,112 | 27,946 | 157 | 5 ± 5 | |
| Follicle-stimulating hormone (mIU/mL) | Epilepsy or bipolar disorder exposed to valproate | 32,885 | 27,946 | 202 | 6 ± 9 | 0.748 |
| | Epilepsy or bipolar disorder not exposed to valproate | 261,112 | 27,946 | 152 | 6 ± 8 | |
| Prolactin (ng/mL) | Epilepsy or bipolar disorder exposed to valproate | 32,885 | 27,946 | 551 | 39 ± 443 | 0.380 |
| | Epilepsy or bipolar disorder not exposed to valproate | 261,112 | 27,946 | 446 | 20 ± 23 | |
| Estradiol (pg/mL) | Epilepsy or bipolar disorder exposed to valproate | 32,885 | 27,946 | 36 | 31 ± 24 | 0.442 |
| | Epilepsy or bipolar disorder not exposed to valproate | 261,112 | 27,946 | 36 | 28 ± 12 | |

**Legend:** Comparisons use two-sided t-test statistics ($p < 0.05$ cut-off). P-values are unadjusted for multiple comparisons. However, with Bonferroni correction applied across 6 comparisons, only p-values < 0.0083 would be considered statistically significant. SD = Standard deviation.

complement our approach by inverse probability weighting the covariates at baseline (another way to emulate randomisation) and applying, for example, marginal structural models to account for subsequent time-varying confounding, and perhaps considering instrumental variable analyses to validate outcomes outside of unmeasured confounding influence[44,45,73,74]. Such methods are not currently available in the TriNetX LIVE platform, but could be applied on a raw data extract in the future, ideally within the causal framework of a target trial emulation[43–45].

We studied male infertility, testicular hypofunction, and testicular atrophy as defined and captured by the ICD-10-CM coding system, which conforms to WHO's established ICD structure and conventions for statistical classification of disease[53]. We also used established WHO 2021 thresholds for low sperm concentration, motility, vitality, normal forms, and semen volume[58–60]. However, ultimately, male infertility is most accurately captured by the success or failure of conception with a female partner known to be fertile, and this information was not directly available for study. Therefore, this study, alongside the majority of prior studies in the valproate field (which have themselves also tended only to measure semen parameters and reproductive hormones—see in Supplementary Table 1)[25,26,28–30,32,34–38], is limited to assessment of surrogate markers of infertility when considering valproate exposure. Beyond some case reports[31,39–42], few studies[27,33] have actually included an assessment of birth rate or conception success on valproate. However, in our study, it remains reasonable, _clinically_, to assume that the frontline physicians who assigned patients an ICD-10-CM code of N46 (male infertility) might have had supportive information, such as a documented history of conception failure with a fertile partner, and that this helped inform their coding decision. Nevertheless, we would not be able to verify this without direct access to clinical records, which is a recognised limitation for all large-scale health data studies. If possible, it will be important for future studies to include conception success and birth rate as outcomes to help build a more complete picture of evidence about valproate and male reproductive health. Such further work could try to incorporate, for example, treatments for infertility as outcomes (such as intrauterine insemination) to more fully capture male infertility. This was not possible in TriNetX as male data are not linked to the interventions their female partners receive to treat infertility.

Another limitation of this study relates to the potential inaccuracies of healthcare coding. From a diagnostic accuracy perspective, we mitigated this by ensuring we used codes that have been validated previously as reliably able to identify epilepsy or bipolar disorder within healthcare datasets[49–52,75]. This helps to minimise diagnostic misclassification risks. For example, it is possible that the use of two prescriptions of valproate and disease coding for epilepsy or bipolar disorder might have generated false positives by misclassifying cases as disease-positive when clinicians have subsequently reconsidered the diagnosis as negative. However, the diagnostic accuracy studies have generally shown that the false-positive rate for epilepsy disease and symptom coding combined with ASMs or bipolar disease coding is low, generating generally high PPVs of >80%, giving us confidence in the probability of few diagnostic misclassifications having occurred[49–52,75]. From an outcome accuracy perspective, we validated this by illustrating that the infertility outcomes assessed can be captured in association with other drugs known to cause infertility under the same study conditions. However, we also acknowledge the potential for selection bias in that men who use drugs traditionally associated with infertility are perhaps more likely to be assessed for subfertility and thus registered with the various study outcomes. However, the growing concerns about subfertility on valproate should act to mitigate the potential or magnitude of this selection bias[8,14–16,18,19,76,77].

Our results can be generalised to men aged <55 years alone. This cut-off was chosen to mirror the age cut-off given by regulators for

restrictions to valproate prescribing in the UK, which do not apply to men aged ≥ 55 years[14,15,76]. Age is an important confounder and was, therefore, propensity score-matched between cohorts in this study, thereby balancing it across the cohorts. However, future studies should still aim to independently assess infertility outcomes in men aged ≥ 55 years on valproate, as well as to compare outcomes between men at different age ranges. Further, our results can only be generalised to living men, as the deceased were excluded to avoid unequal competitive risks between death and infertility outcomes between cohorts. The main source for mortality data in TriNetX is electronic health records (EHRs), capturing >4 million provider-supplied deaths. As data sourced from EHRs alone may underestimate deaths due to some occurring outside of healthcare information workstreams, TriNetX source additional mortality data from outside EHRs, including from government (Social Security Administration's Master Death File, covering 20% of US deaths), private obituaries (covering 40% of US deaths), and private claims (covering 20% of US deaths)[48]. TriNetX reports that their mortality data covers ~85% of the US population through these linked strategies[47]. Therefore, our study is likely to have captured and excluded the vast majority of deaths and sampled the remaining living males adequately between cohorts. As patients move away from HCOs in the TriNetX catchment, their medical records end, and censorship occurs after the last clinical data point. This is differentiated from situations where the clinical record has ended due to death having happened, as information about the death occurring is usually available from one of the external mortality data sources described above. The mortality data derived from EHRs are refreshed every two to four weeks on average, and data from third-party sources are updated monthly. The possibility of some immortal time biases, however, could not be excluded, as censorship occurring after the last clinical data point would not take into account prior intervals during which participants were temporarily deregistered from HCOs in the TriNetX catchment[70].

We acknowledge that the dynamic nature of data acquisition onto the TriNetX LIVE platform serves as both a strength and a limitation. It is a strength that new patients and HCOs are added daily, meaning sample sizes increase with time, and results are representative of real-time frontline clinical decisions and progress. Conversely, this growth can make it challenging to replicate exact counts over time. Nonetheless, we expect that trends with a strong biological basis will retain their direction of effect despite the evolving nature of the dataset, and our findings support this expectation as there were no changes in the direction of effect or conclusions drawn between our original search at submission (conducted on 09/06/2024, capturing 606,785 men with epilepsy or bipolar disorder from 120 HCOs; logged here: https://doi.org/10.6084/m9.figshare.27310551.v1) and our updated search after the first peer review (conducted on 24/10/2024, capturing an expanded 633,405 men from 131 HCOs; logged here: https://doi.org/10.6084/m9.figshare.27310593.v1). The results included in the current version of the manuscript are drawn from updated searches undertaken on 24/03/2025 following a further round of peer review. Conclusions remain unchanged.

Although we had a minimum requirement of valproate exposure on at least two separate occasions and were able to demonstrate detectible levels of valproate in the serum of 94% to 98% of men who had this tested, inferring a good level of drug adherence in these cohorts, we were unable to provide additional data on the dose or duration of treatment as these were not available. Future work should aim to explore dose-response for infertility outcomes on valproate, e.g., using a raw exposure dataset purchased from TriNetX or other large datasets like Clinical Practice Research Datalink (CPRD) in the UK[78]. Finally, the survival analyses undertaken in TriNetX do not generate or have figures available for the population at risk at each time point across the Kaplan-Meier curve, which is limiting for

understanding the progression of risk over time in exposed and unexposed individuals. Although HRs are available with 95% CIs, CIs cannot be integrated into the Kaplan-Meier curves. A raw TriNetX data extract would allow such flexibility in analysis, but would come with a monetary cost for the data acquisition. The findings of our study and their implications would help justify investment in such further work.

In conclusion, our study results do not support an association between valproate use and infertility in men with epilepsy or bipolar disorder in real-world settings. The study is strengthened by its size and robust propensity score matching profile. Comparing valproate-exposed and unexposed men with epilepsy or bipolar disorder in this study has helped to limit the confounding effect that these conditions have on infertility. We investigate infertility outcomes alone in this study (including testicular dysfunction and atrophy). Our results cannot be generalised to any other potential effects of valproate in men, such as whether there are transgenerational risks with prenatal exposure[14]. Separate work will need to be undertaken to investigate this in humans.

## Methods
### Study design and setting
We undertook a retrospective cohort study of de-identified international healthcare data in the TriNetX LIVE platform, designed within a STROBE-compliant framework. TriNetX is the world's largest ecosystem of real-word electronic health data, drawn from ~250 m patients from >200 HCOs across 19 countries predominantly in North America but also South America, Europe, the Middle East, Africa, and Asia Pacific[47,48]. It holds ~70 billion date- and patient-indexed electronic clinical observations[47]. Much of the data imported by TriNetX comes from large academic HCOs, although data from smaller rural hospitals are also imported to better reflect patients from all care backgrounds. Similarly, recruitment of HCOs into TriNetX is agnostic of insurance status, meaning data from both insured and uninsured patients are captured. Data are imported from a variety of healthcare settings, including emergency care, inpatients, outpatients, and primary care. The data imports occur in real-time from HCOs that are online at the time a researcher connects to the TriNetX LIVE network and undertakes a search, meaning results are contemporaneous with frontline clinical progress. Variables held include demographics, diagnoses (using International Classification of Diseases, Tenth Revision, Clinical Modification (ICD-10-CM) codes), drugs (including RxNorm-coded, Anatomical Therapeutic Chemical Classification (ATC), and Veterans Affairs (VA) formulary), procedures (including Current Procedural Terminology (CPT) and SNOMED codes), laboratory results, genomics, tumour registration, and mortality data. The mortality data are enriched with data sourced from outside healthcare settings to increase coverage. More information about TriNetX can be found online (https://trinetx.com)[47,48].

### Search strategy and participants recruited
Searches of the Global Collaborative Network in the TriNetX LIVE platform were undertaken on 24/03/2025 to build cohorts of men aged < 55 years with epilepsy or bipolar disorder exposed and unexposed to valproate using the following algorithm:

#### Search One: epilepsy or bipolar disorder cohort.
A. *Epilepsy:* at least two G40 (epilepsy) and/or R56 (seizure) ICD-10-CM codes recorded anytime in the record
B. *Bipolar disorder:* at least two F31 (bipolar disorder) codes recorded anytime in the record
C. Not deceased
D. (A or B) + C
E. Valproate (RxNorm 40254, any dose) was prescribed on two or more occasions in the record

F. Valproate never prescribed
G. D + either E or F

**Search Two: epilepsy cohort.**
A. *Epilepsy:* at least two G40 (epilepsy) and/or R56 (seizure) ICD-10-CM codes recorded anytime in the record
B. Not deceased
C. A + B
D. Valproate (any dose) prescribed on two or more occasions in the record
E. Valproate never prescribed
F. C + D or E

**Search Three: bipolar disorder cohort.**
A. *Bipolar disorder:* at least two F31 (bipolar disorder) codes recorded anytime in the record
B. Not deceased
C. A + B
D. Valproate (any dose) prescribed on two or more occasions in the record
E. Valproate never prescribed
F. C + D or E

No restrictions were placed on other ASMs or mood-stabilisers prescribed as these and other relevant medications were then propensity score matched at baseline between comparison groups (see statistical analysis). To get an estimate of adherence to valproate, we extracted serum or plasma valproate levels, where recorded, and reported the proportion with detectible levels at the most recent visit in which levels were checked.

A sensitivity analysis was undertaken to establish outcomes in men with epilepsy or bipolar disorder exposed and unexposed to valproate, following exclusion of those with key modifiable risk factors for infertility as follows:

**Search Four: epilepsy or bipolar disorder cohort, excluding those with key modifiable risk factors for infertility.**
A. *Epilepsy:* at least two G40 (epilepsy) and/or R56 (seizure) ICD-10-CM codes recorded anytime in the record
B. *Bipolar disorder:* at least two F31 (bipolar disorder) codes recorded anytime in the record
C. Not deceased
D. Cannot have: finasteride (RxNorm 25025), ketoconazole (RxNorm 6135), trimethoprim (RxNorm 10829), nitrofurantoin (RxNorm 7454), erythromycins/macrolides (VA:AM200), aminoglycosides (VA:AM300), antineoplastics (VA:AN000), androgens/anabolics (VA:HS100), progestins (VA:HS800), estrogens (VA:HS300), spironolactone (RxNorm:9997), cyproterone (RxNorm:3014), Radiation Oncology Treatment (CPT:1010843), Surgical Procedures on the Urinary System (CPT:1008061), Repair initial inguinal hernia, age 5 years or older (CPT:1008011), Surgical Procedures on the Male Genital System (CPT:1008470), Alcohol related disorders (ICD-10-CM F10), Opioid related disorders (ICD-10-CM F11), Cannabis related disorders (ICD-10-CM F12), Sedative, hypnotic, or anxiolytic related disorders (ICD-10-CM F13), Cocaine related disorders (ICD-10-CM F14), Other stimulant related disorders (ICD-10-CM F15) Hallucinogen related disorders (ICD-10-CM F16), Nicotine dependence (ICD-10-CM F17), Inhalant related disorders (ICD-10-CM F18), Other psychoactive substance related disorders (ICD-10-CM F19)
E. (A or B) + C + D
F. Valproate (any dose) prescribed on two or more occasions in the record
G. Valproate never prescribed
H. E + either F or G

**Case-ascertainment accuracy**
Studies validating the diagnostic accuracy of ICD-10 coding for epilepsy or bipolar disorder in TriNetX *specifically* are as yet unavailable. However, more widely, there are a substantial number of studies validating the diagnostic accuracy of these codes, including in regions and settings covered by TriNetX[49]. A case-ascertainment strategy combining disease coding (G40, including epilepsy and status epilepticus) and/or symptom coding (R56.9) with co-prescribed ASMs is a valid way to accurately capture epilepsy within electronic healthcare datasets, and this spans multiple studies in a variety of healthcare settings and countries[49]. This strategy tends to achieve figures of 80–90% across positive predictive values (PPVs) and sensitivity estimates, and negative predictive values (NPV) and specificity estimates approach 100%[49,50]. The corresponding F1 scores are also high at 0.86[70]. F31 codes are also known to accurately capture bipolar disorder, with PPVs, sensitivities and NPVs reaching 84%, 89%, and 98%, respectively[75].

**Outcomes**
The first occurrence of the following coded outcomes was measured after index up to the day of the data search (24/03/2025), excluding any persons who had the outcome coded prior to index (see statistical analysis for index specification) for the dichotomous outcomes:
i) Male infertility (ICD-10-CM code N46)[53];
ii) Testicular hypofunction (ICD-10-CM code E29.1)[53];
iii) Testicular atrophy (ICD-10-CM code N50.0)[53];
iv) A composite measure of low sperm concentration ($< 16 \times 10^6$/ml semen), low sperm motility (< 42% total motility), low sperm vitality (< 54% alive), low normal forms (< 4% of sperm have a normal morphology), and low semen volume (< 1.4 mL)[58–60];
v) Reproductive hormone levels: total testosterone (ng/dL), free testosterone (pg/mL), LH (mIU/mL), FSH (mIU/mL), prolactin (ng/mL), and estradiol (pg/mL)[25,28–30,32–36,38,42]. The most recent result taken during follow-up was used.

To assess the longitudinal effect of valproate exposure, the dichotomous outcomes (i–iv) were also combined into a unitary infertility measure. This was evaluated at multiple intervals after valproate initiation: 30, 60, 90, 180, and 360 days, and at 2, 5, and 10 years.

ICD-10-CM is the International Classification of Diseases, Tenth Revision, Clinical Modification, which is a standardised system that is widely used to code diseases and medical conditions amongst HCOs, particularly in the US[53]. It is curated by the Centres for Disease Control and Prevention (CDC)[53]. ICD-10-CM is approved by the WHO and conforms to the WHO's established ICD structure and conventions for statistical classification of disease[53]. Healthcare providers use ICD-10-CM codes when diagnosing patients, which are then uploaded to TriNetX directly to allow study data to reflect real-world, real-time, clinical decisions and diagnoses[47,48]. Full details about how the WHO and CDC define individual ICD-10-CM codes can be found within their respective reference texts[53].

**Outcome ascertainment accuracy**
There are no studies of the diagnostic accuracy of ICD-10 coding for male infertility, although a study of ICD-9 coding indicates that such codes are likely to be accurate[79]. To assess the validity of the TriNetX database being able to capture male infertility under our study conditions, we repeated the analysis to build cohorts of men aged < 55 years with epilepsy or bipolar disorder exposed and unexposed to drugs known to cause infertility (including chemotherapy, androgens, spironolactone, oestrogens: see Supplementary Table 19 for full list of drugs contained under these subheadings). We excluded anyone ever prescribed valproate to ensure it did not confound the validation results. The following search strategy was used:

**Search Five: epilepsy or bipolar disorder cohort exposed and unexposed to drugs known to cause male infertility.**

A. *Epilepsy:* at least two G40 (epilepsy) and/or R56 (seizure) ICD-10-CM codes recorded anytime in the record
B. *Bipolar disorder:* at least two F31 (bipolar disorder) codes recorded anytime in the record
C. Not deceased
D. Cannot have: valproate
E. (A or B) + C + D
F. Any of the following prescribed on two or more occasions in the record: antineoplastic agents (ATC:L01), spironolactone (RxNorm 9997), estrogens (ATC:G03C), androgens/anabolics (VA:HS100), anti-androgens (ATC:L02BB)
G. Any from F was never prescribed
H. E + either F or G

## Statistical analysis

Three study cohorts were created: A) men with epilepsy or bipolar disorder; B) men with epilepsy; and C) men with bipolar disorder. The baseline covariates listed in Supplementary Tables 2–14 were extracted and propensity score matched between those exposed and unexposed to valproate in each of these cohorts. Propensity scores were estimated using logistic regression (via scikit-learn). Matching was performed on a 1:1 basis using greedy nearest neighbour matching without replacement, with a caliper width of 0.1 pooled standard deviations. Cohorts were randomly shuffled prior to matching to reduce order bias[80,81]. The primary focus of propensity score matching is to achieve balance between covariates amongst the cohorts. As part of the propensity score matching process, *p*-values and SMD were generated for each covariate pre- and post-matching between cohorts. We defined an SMD threshold of ≤ 0.1 to indicate adequate covariate balance[64]. Uncorrected *p*-values were calculated by a two-sided *t*-test for continuous covariates (which were age and BMI) and two-sided Z-test for categorical covariates (which were all the rest). In propensity score matching, *t*-tests are used to compare covariates between the two groups (pre- and post-matching) to ensure that matching has reduced differences in covariates and, therefore, has made groups comparable. This means the *t*-test, in this situation, is applied for descriptive purposes to check for cohort covariate balance, rather than for hypothesis testing or drawing inferences about study outcomes between cohorts. Therefore, a formal requirement for normality testing is less relevant. Furthermore, the large sample sizes typical of TriNetX mean the data generally benefit from the Central Limit Theorem, which states that the sampling distribution of the mean tends to be normal as the sample size increases, regardless of the underlying distribution of the data[82]. In such scenarios, it is generally accepted that the *t*-test can be used even if data are not perfectly normal, because the large sample size compensates for deviations from normality. The same applies to the comparisons we made of reproductive hormone levels (laboratory results), where two-sided *t*-tests were used to test hypotheses. Although TriNetX did not test for normality here, the Central Limit Theorem allows us to reasonably assume normal distribution and/or compensation for deviations from normality owing to large samples[82].

Cohorts were propensity score matched for all of the baseline covariates listed in Supplementary Tables 2–14 at the start follow-up (time 0: corresponding to the index event). The index event was taken as the second recorded valproate prescription in those exposed to the drug, or the second recorded epilepsy or bipolar disease code in those unexposed to the drug. This allowed cohorts to be propensity score matched for epilepsy and bipolar disorder using the first code to appear for these conditions. It also ensured men with only a single exposure to the valproate or a single seizure were excluded.

Lifetime follow-up commenced from day 1 after the index until the date of data search (24/03/2025). Survival analysis was undertaken on the fully matched cohorts using Cox-proportional hazard regression models to calculate HRs with 95% CIs for the risk of each categorical outcome occurring, alongside two-sided Log-Rank *p*-value testing (0.05 significance cut-off). Kaplan-Meier plots were generated for statistically significant results. Two-sided *t*-tests were used for continuous outcomes (reproductive hormone profiles, 0.05 significance cut-off). Participants were removed from the analysis (censored) after the last clinical data point in their record. Data were analysed within the TriNetX LIVE platform using R4.0.2, Python 3.7, and Java 11.0.16.

## Ethics Statement

The University of Liverpool Research Ethics decision tool was used to determine that ethical approval was not required as the study was a secondary analysis of data that were anonymised by an external party (TriNetX) and provided to the research team in a fully anonymised format. Data were de-identified per the de-identification standard defined in Section 164.514(a) of the HIPAA Privacy Rule. TriNetX data are attenuated to ensure participating HCOs remain anonymised. This includes withholding the names of participating countries with less than three HCOs contributing data, and shielding frequencies to ≤10 when ten or less patients experience an outcome, in order to protect their anonymity. No personal data were used or patient recruitment undertaken as part of this study.

## Reporting summary

Further information on research design is available in the Nature Portfolio Reporting Summary linked to this article.

## Data availability

All analyses were conducted within the TriNetX LIVE platform, and results were exported as aggregate data outputs and summary statistics, which we have made publicly available via FigShare (https://doi.org/10.6084/m9.figshare.28728839). These are not the Source Data required to reproduce the figures and tables in our manuscript. The Source Data required to reproduce the figures and tables in our manuscript are real-world clinical records from patients and so are held securely within the TriNetX LIVE platform; they cannot be publicly deposited under TriNetX policy and the U.S. HIPAA Privacy Rule de-identification requirements (§164.514). Other researchers can request access to the TriNetX LIVE platform (www.trinetx.com/about-trinetx/contact) via their institution, follow the methods described in this manuscript within the TriNetX LIVE platform, and in doing so, reproduce the figures and tables, with small variations in counts expected due to real-time data updates from participating healthcare organisations over time. Custom code was not generated for this study. For the purpose of open access, the author has applied a Creative Commons Attribution (CC BY) licence to any Author Accepted Manuscript version arising.

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

## Acknowledgements

G.K.M. is supported by a National Institute for Health and Care Research (NIHR) Clinical Lectureship (CL-2022-07-002), Academy of Medical Sciences (AMS) Starter Grant for Clinical Lecturers (REF: SGL030\1029), and an Epilepsy Research Institute Emerging Leader Fellowship (F2401). A.G.M. is supported by the NIHR Applied Research Collaboration North West Coast (ARC NWC). The funders had no role in the study design, data collection, analysis and interpretation, or writing of this manuscript. The views expressed in this publication are those of the author(s) and not necessarily those of the NIHR, AMS, the Epilepsy Research Institute, or the Department of Health and Social Care. We are also grateful to Iain Buchan, Glen Martin, and Matthew Sperrin for their assistance with our queries regarding causal inference.

## Author contributions

G.K.M., S.E.D.: Conceptualisation; G.K.M.: Methodology; G.K.M., T.S.: Data Curation, Formal Analysis; G.K.M.: Writing – Original Draft; G.K.M., S.E.D., L.N., L.V.W., M.T.M., G.Y.H.L., A.G.M.: Writing—Review & Editing; G.K.M.: Funding Acquisition; G.Y.H.L., A.G.M.: Supervision.

## Competing interests

A.G.M. declares i) a UCB Pharma grant paid to University of Liverpool for the National Audit of Seizure Management in Hospitals (NASH) study, which is unrelated to the submitted work; ii) an Angelini grant to be paid to University of Liverpool as co-applicant for A multi-method PRoject to maximise efficient and equitable pathways tO suPport from a rEgional epiLepsy centre (PROPEL), which is unrelated to the submitted work; iii) Honoraria paid to University of Liverpool for lectures

unrelated to the submitted work given at educational events sponsored by Sanofi, Eiasi, and GSK; iii) Support from Angelini for attendance unrelated to the submitted work at the 2024 International League Against Epilepsy (ILAE) congress. G.K.M. declares an Angelini grant paid to the University of Liverpool as co-applicant on the PROPEL study, which is unrelated to the submitted work; ii) Honoraria paid to the University of Liverpool for delivering a lecture at an educational event sponsored by Angelini, which was unrelated to the submitted work. G.Y.H.L. is a consultant and speaker for BMS/Pfizer, Boehringer Ingelheim, Daiichi-Sankyo, and Anthos, which is unrelated to the submitted work. No fees are received personally. G.Y.H.L. is a National Institute for Health and Care Research (NIHR) Senior Investigator and co-principal investigator of the AFFIRMO project on multimorbidity in AF, which has received funding from the European Union's Horizon 2020 research and innovation programme under grant agreement No. 899871, which is unrelated to the submitted work. T.S. is an employee of TriNetX, which routinely provides data for research studies, including the data used in this study. The involvement of T.S. as a co-author reflects her technical assistance as a data analyst and does not imply any compensation or influence from TriNetX on this study. The remaining authors declare no competing interests.
