## [Transparent Peer Review file · Nature Communications]

An international cohort study of valproate and infertility in men with epilepsy or bipolar disorder

Corresponding Author: Dr Gashirai Mbizvo

Version 0:

Reviewer comments:

Reviewer #1

(Remarks to the Author)

Thank you for the opportunity to review the paper "Infertility outcomes in valproate exposed and unexposed men with epilepsy or bipolar disorder: an international cohort study of real-world healthcare data". Mbizvo and colleagues provide a comprehensive examination of the potential effects of valproate on male fertility in patients with epilepsy or bipolar disorder using a large retrospective cohort study. The article nicely addresses an important clinical question, especially given its widespread use in treating epilepsy and bipolar disorder, providing results from a large sample size and use of real-world data after conducting a comprehensive adjustment for confounders, despite the potential biases deriving from the retrospective design and the limited validation of outcome measures within the database.

I have some comments and suggestions to improve the paper:

Introduction:

- When reporting the current evidence on valproate effects on male fertility the paper could benefit from a more structured presentation of the studies' outcomes and limitations. It would be useful to report the level of evidence (instead of "small human studies" in line 33, 66 and others report the design of the studies) of the available literature.
- Line 43, when referring to bipolar disorder the format is too like the former sentence. Change to improve flow.
- When referring to the main outcome (e.g. "male fertility and/or adverse effects on the testes", line 65 and others) the authors should better clarify these definitions. For instance, how do the definitions used (e.g. male infertility [ICD10-CM N46]) match with the WHO definition of infertility?
- Regarding citation 12 (line 66, 67, 69-72, 75), not the comprehensive review, but the specific studies should be cited, since they report different outcomes.
- Line 79-81: "Valproate did not have a negative effect on male reproductive hormones in the men with bipolar disorder, and elevated prolactin and follicle-stimulating hormone levels in men with epilepsy was attributed to the epilepsy". I suggest revising/rephrasing the sentence for clarity.
- Line 77 to 83. When referring to the Turkish study (citation 14) it appears that in "two out of five men" reduced sperm count and motility were described, but the authors state that the study included "18 men with bipolar disorder on valproate monotherapy or in combination with lithium therapy, and 15 men with epilepsy on valproate monotherapy".

Results and discussions:

The results are presented clearly, with specific outcomes measured and hazard ratios reported. It's good to see both the absolute risk changes and the statistical significance (or lack thereof) clearly presented. It could benefit from a comparison with the findings of former studies and a more critical analysis of why the current results differ.

- Line 93. When referring to the composite measure, please describe how these semen parameters were chosen among others usually taken into consideration (e.g. morphology, vitality).
- Line 94. The authors could clarify why the index event was defined as stated in the article (e.g. second recorded valproate prescription to establish chronic exposition? Second recorded epilepsy or bipolar disease code to exclude misdiagnosis?).
- Line 189. Please state the n. or % of patients who had lab results on semen parameters available.
- Line 279. Authors should add as a possible limitation that considering the two prescriptions of valproate and the ICD code might miss possible epilepsy/BD misdiagnosis, reconsidered during the follow-up time.

Tables and Figures:

The tables summarizing the results are well-organized and informative. Kaplan-Meier plots aid in visualizing the survival analysis results.

Table 1:

- Include p-value along with HR.

- There is a mismatch between the total of patients with epilepsy and/or bipolar disorder and sum of the number of patients in each group. The mismatch is probably due to patients with epilepsy and bipolar disorder. Please confirm that or give an alternative explanation.

Minor Comments:

- Abstract: while not mandatory, reporting p-value (line 39, $p < 0.05$) could be useful when stating the non-significance of the results.
 - Line 274. Standardized mean difference, use SMD as abbreviation.
 - In the notes: "552 patients in Cohort 1 was excluded". Correct with "were".
 - In the notes: "815 patients in Cohort 1 was excluded". Correct with "were".
 - In the notes: "1290 patients in Cohort 1 was excluded". Correct with "were".
 - Line 118. Define HCOs (Healthcare organization [HCO]). Prefer "clinical data" to "clinical facts".
 - Line 122. When expressing mean age use mean (SD).
 - Line 126: "compared to epilepsy or bipolar". Add "patients".
 - Line 127, 149, 156: When expressing the results for the HR and 95% CI for the three groups I suggest introducing the data as follows: "HR with 95% CI for were obtained for the following outcomes: ...". Also, I suggest changing the format to express the numbers. Use for instance "male infertility (HR 1.063; 95% CI 0.828 to 1.364)".
 - Line 164. Authors could consider rephrasing "Following this, there is little evidence demonstrated to suggest an association between valproate and male infertility" with "Subsequently, there is minimal evidence to suggest an association between valproate and male infertility".
 - Line 182. Change "seems to be when..." with "seems to emerge when".
 - Line 97. When introducing the ">120 baseline covariates" please refer to the supplementary table. 6. Same thing when describing drugs known to cause infertility (line 153) used in the sensitivity analysis, refer to Supplementary Table. 5
- Supplementary Table 1:
- Add the total n. of patients with available valproate serum level.
- Supplementary Table 2:
- Add the total n. of patients with available valproate serum level.
- Supplementary Table 3
- Add the total n. of patients with available valproate serum level.
- Supplementary Table 5:
- Please add a note explaining the codes that are reported along with drugs categories and specific drugs (e.g. ATCL01 Antineoplastic agents)

Reviewer #2

(Remarks to the Author)

Thank you for giving me the opportunity to review this manuscript on potential adverse events (reduced fertility) associated with valproate treatment in fertile men. In both psychiatry and neurology, this is an important and urgent issue to address.

I have the following comments:

Comment 1. The section "Main" gives a good introduction to why, restrictions to the use of valproate is problematic and not without potential consequences for the persons with epilepsy and bipolar disorder. Please, use original references e.g., for the declining use of valproate.

Comment 2. Authors mention "A combination of validated disease and prescription codes were used to create three large cohorts of males aged <55 years within the TriNetX Global Collaborative Network" – there are two references: Ref. 15: TriNetX. Welcome to the world's largest, living ecosystem of real-world data and evidence for the life sciences and healthcare industries. Global data, for global health (2023), and Ref 16. Mbizvo, G.K., Bucci, T., Lip, G.Y.H. & Marson, A.G. Morbidity and mortality risks associated with valproate withdrawal in young men and women with epilepsy. Brain (2024). In the latter, they recognize that the dataset used was not validated (TriNetX data). Maybe more relevant to describe that although they use data in the present study that has not been validated, previous studies suggests that the validity of the data is sufficient for the purpose and refer to studies describing validation of epilepsy diagnoses e.g., their previous publication Mbizvo GK, Bennett KH, Schnier C, Simpson CR, Duncan SE, Chin RFM. The accuracy of using administrative healthcare data to identify epilepsy cases: A systematic review of validation studies. Epilepsia. Jul 2020;61(7):1319

Comment 3. The authors mention "The comparison groups were propensity-matched for >120 baseline covariates before index." (Supplementary Table 6: Baseline characteristics matched as covariates). Is there a reference or rationale for the choice of these variables?

Comment 4. The authors mention (line 243) "As we showed previously, a case-ascertainment strategy combining disease coding (G40, including epilepsy and status epilepticus) and/or symptom coding (R56.9) with co-prescribed ASMs is a valid way to accurately capture epilepsy within electronic healthcare datasets, and this spans multiple studies in a variety of healthcare settings and countries." Please describe the code "R56.9" – it is not found in the WHO browser (<https://icd.who.int/browse10/2019/en#/R56>) – is there an error here? Is it possible to provide information on the distribution of cases identified with the various algorithms for epilepsy: "Epilepsy: at least two G40 (epilepsy) and/or R56 (seizure) ICD-10-CM codes recorded anytime in the record" (G40, R56 and the combination of the two)?

Comment 5. The authors describe the dataset "TriNetX is the world's largest ecosystem of real-word electronic health data, drawn from ~250m patients from >120 healthcare organisations (HCOs) across 19 countries predominantly in North America

but also South America, Europe, the Middle East, Africa, and Asia Pacific". This surely is an impressive dataset, but it is difficult to understand the coverage of the data and who may be missing. As far as I understand, the cohort studied is dynamic, which makes it difficult to replicate findings reported. Is there some way that the authors can log the dataset studied to allow for replication of findings?

Comment 6. I fully agree with the authors choice to include mortality in their analyses – but the manuscript provides very limited data: "The mortality data are enriched with data sourced from outside healthcare settings to increase coverage" – could the authors elaborate on quality of the mortality information to give the reader some impression on how precise mortality is accounted for? (e.g., mortality vs. persons who move away from the HCO catchment area).

Comment 7, the authors describe male infertility and how this was captured using ICD10 coding. "There are no studies of the diagnostic accuracy of ICD-10 coding for male infertility, although a study of ICD-9 coding indicates that such codes are likely to be fairly accurate" – although based on ICD-9 coding, the reference provides measures of the validity (as opposed to "fairly accurate") e.g., reference 23 describes a positive predictive value of 85%. However, studies suggest that infertility rates in e.g., the Nordic countries approach 5% (<https://nhwstat.org/populations/fertility>). The algorithm identified Infertility outcomes in men (up to 1%), but although the validity of the diagnoses used to identify infertility may be high, the algorithm may not capture all men with sub-optimal fertility. Would it be possible include treatment for infertility in the analyses (e.g. intrauterine insemination) to more fully capture male infertility?

Comment 8, although the validity of the coding for male infertility is high (e.g., positive predictive value over 85%), the authors do not address completeness of the registration (see comment above). The infertility diagnosis is likely given mainly to males who are actively seeking to become a parent. As mentioned, birth rates are lower in men with epilepsy (Artama, M., Isojarvi, J.I. & Auvinen, A. Antiepileptic drug use and birth rate in patients with epilepsy-a population-based cohort study in Finland. *Hum Reprod* 21, 2290-2295 (2006).). It may be reasonable to assume that males with epilepsy (and possible also bipolar disorder) are less likely seeking to become a parent and thus less likely to be identified with one of the outcomes. This is accounted for in this study by comparing the cohorts exposed and unexposed to valproate. However, subfertility in males with epilepsy may relate to the indication for use of valproate e.g., type of epilepsy (generalized vs. focal epilepsy) and type of epilepsy may be associated with fertility (see e.g., <https://doi.org/10.1177/15357597221135717>). Could the authors elaborate on this potential bias and possibly address this in analyses?

Comment 9. The authors address infertility outcomes in men with epilepsy and or bipolar disorder exposed and unexposed to drugs known to cause infertility (Supplementary Table 4). Males who use drugs known to be associated with infertility will be much more likely to be assessed for subfertility and thus be registered with the various outcomes (male infertility). Thus, this has implications for the use of this analysis as a "positive control" for the outcome. Please, elaborate on the limitations of using this outcome as support for the method used.

Comment 10. I find that figure 1 is hard to read – is it possible to revise the y-axis (survival probability) to capture potential differences? (e.g., focusing survival on 100 – 95%)? Does the survival curve account for mortality? Would it be possible to add the number of patients followed, e.g., by each of the time points (1,000 days, 2,000 days etc.)? Would absolute figures (%) be more illustrative? (e.g., cumulative incidence accounting for mortality?). I was unable to find Supplementary Figure 1 (line 377).

Reviewer #3

(Remarks to the Author)

With the background that the extent to which valproate actually (may) causes impaired male fertility and/or adverse effects on the testes is largely unknown, the manuscript titled "Infertility outcomes in valproate exposed and unexposed men with epilepsy or bipolar disorder: an international cohort study of real-world healthcare data" reports the findings of a retrospective cohort study of "reproductive health" in men with epilepsy or bipolar disorder treated with valproate, using real-world healthcare data from TriNetX. Hence, by measuring lifetime propensity-matched risks of male infertility (ICD10-CM N46), testicular hypofunction (E29.1), testicular atrophy (N50.0), and a composite measure of sperm and semen parameters, the authors depicted a not statistically significant difference between men exposed as compared with those unexposed to valproate (the magnitude of effect was <1% difference to being exposed vs. unexposed to valproate).

These data are the largest ever considered focusing on the specific topic of reproductive health from the male standpoint. In this setting, the authors have complied with the recommendations of the 'Sex and Gender Equity in Research – SAGER – guidelines', clearly explaining that the retrospective data analysis did only apply to men (as per sex assigned at birth). The terms have been properly used both in the title and the abstract.

Overall, the study is well conducted and provides a clinically interesting piece of information, despite several limitations should be mentioned.

Major issues:

1) Over the materials and methods, the authors state that they queried TriNetX to analyze men under 55 years of age. Is there a specific reason why this cut-off was chosen? It would be interesting to explore the outcomes of interest at different age ranges, as aging is a confounding factor. Additionally, the Authors should explain why they chose to require at least two diagnostic codes for epilepsy or bipolar disorder, rather than a single code. This decision could impact the selection of participants and therefore the outcome itself.

2) The Authors did adjust their analyses for testosterone use, anabolic steroid use, chemotherapy treatments etc. via propensity score matching (PSM). As such, these factors should not be part of the PSM but rather should be used as exclusion criteria. The fact that two cohorts are statistically similar in their use of anabolic steroids does not exclude the

influence of anabolic steroids on the outcome (e.g., azoospermia or development of severe oligospermia). Instead, the model should be adjusted for non-modifiable risk factors, such as age. This Reviewer considers that this methodological issue significantly impacts the validity of the reported findings.

3) The results section does not provide information on the duration of valproate exposure or dosage. This is more than crucial. Despite this Reviewer doubts this information can be found on TriNetX, this should be listed over the limitation section. Moreover, the Authors did not provide information on the timing of infertility, oligospermia, testicular dysfunction etc. diagnoses relative to valproate exposure.

4) Overall, the use of coding can limit the validity of results. Among other reasons, It cannot assure adherence to and duration of treatment (up to two prescriptions is recorded).

5) It is not clear what coding N46.8 (Other male infertility), N46.9 (Male infertility, unspecified), and 606 (Infertility, male); Testicular hypofunction(E29.1) actually mean. How is Testicular atrophy (N50.0) defined? Overall, the Authors have not provided information on the specific hormonal profiles in their results. It is crucial to understand how testicular dysfunction is classified within the TriNetX system. More detailed information is required regarding the criteria used to define testicular dysfunction, including which hormones were measured, the threshold values used for classification, and how these measurements were recorded and extracted from the TriNetX database.

6) It is well known that oligospermia is not a surrogate for male infertility (also according to WHO indications); as a whole, it must be clearly discussed that evaluating semen parameters in itself represents only a surrogate for spermatogenetic health, but not for reproductive outcomes; indeed, fertility should be considered as a successful reproduction as measured by a live birth and not with semen/sperm parameters only; this issue deserves to be comprehensively discussed in the manuscript; therefore, the main limitation of this study is the outcomes selected by coding which cannot realistically capture male infertility.

7) The authors do not sufficiently address the potential limitations of using ICD-10 codes for identifying infertility outcomes, particularly given the complex nature of male infertility diagnoses. Limitation section should be definitely implemented.

Other comments

1) The Authors stated "The comparisons were made using chi-squared tests for categorical variables and independent-sample t-tests for continuous variables". This Reviewer does assume data distribution was tested and statistical analysis performed accordingly.

2) The censoring approach (removing participants after the last fact in their record) could introduce immortal time bias if not handled correctly. More detail on how this was managed is needed.

3) Matching does not include important risk factors for male infertility such as varicocele and cryptorchidism, as well as recreational habits (cigarette smoking, alcohol consumption, illicit drug use) which may eventually impact toward sperm parameters.

4) The Authors should use the same units of measure reported in the WHO manual for the examination of human semen (sperm concentration x10⁶/mL)

5) The "pooled estimate of standardized differences" is not a common term and may be confusing to readers. Please rephrase.

6) In the results section, the Authors omitted reporting the number at risk in their Kaplan-Meier analyses. As far as this Reviewer is aware, TriNetX does not provide numbers at risk in their Kaplan-Meier curves by default; although this holds true, a limitation should be explicitly acknowledged if this applies. Alternatively, if the Authors have access to this information, they should include it to enhance the robustness and clarity of their analysis. To this aim, the Authors could consider exporting the data to spreadsheets and recreating the Kaplan-Meier curves using third-party statistical software.

7) The authors do not report Kaplan-Meier risk estimates for specific time points, which is a significant omission even if the results are not statistically significant. These estimates provide important information about the progression of risk over time for exposed vs. unexposed and the outcomes of interest. Lastly, the inclusion of confidence intervals in the Kaplan-Meier curves is needed.

Version 1:

Reviewer comments:

Reviewer #1

(Remarks to the Author)

The authors have revised the manuscript and they have addressed the main concerns raised. I have no additional comments. In my opinion, the paper could be accepted in the present version

Reviewer #4

(Remarks to the Author)

The noteworthy results are that the evidence used by regulators to justify valproate restriction in men is flawed. This is significant as it means that valproate is an effective medication which is denied to men without a sound evidence base.

This is a large population based study and design is superior to other studies. The greatest strength of the study is that it compares men with epilepsy or bipolar disease prescribed valproate versus those not prescribed valproate, reducing the confounders in other studies.

The limitations of the study are transparently and well described in the paper.

Suggestions:

***Abstract:**

This should include the key point about comparison between valproate exposed vs unexposed groups outlined above.

The sentences are long and overly complicated and this detracts from the important messages:

eg Line 31-34: "Use is becoming increasingly restricted for men owing to widespread warnings about it causing testicular dysfunction and infertility. Most of the existing evidence on this comes from animal studies, with limited and inconclusive data available from human research. " ..this could be simplified to "Regulatory restrictions for men are based on warnings about it causing testicular dysfunction and infertility. Evidence for this comes largely from animal studies, with limited and inconclusive human data."

Line 305-307: "Any evidence that valproate is linked to male infertility in epilepsy seems to emerge when it is compared to healthy men without epilepsy, which were the controls used in most of the prior studies." could be simplified to "Prior studies linking valproate to male infertility in epilepsy mainly used healthy men without epilepsy as controls".

The conclusion of the abstract is very weak compared to the strength of the data:

Line 43-44: "Our results suggest more caution is needed in associating valproate use for epilepsy or bipolar disorder with male infertility in humans."

should be something like: "Our results do not support a causal association between valproate use for epilepsy or bipolar disorder and male infertility in humans"

Reviewer #5

(Remarks to the Author)

This manuscript represents a significant effort on authors. Thank you for the many efforts the author team made to update the manuscript based on previous review(s). Specific concerns follow, but my focus on updating this manuscript is on coding concerns, and rate of abnormal semen analysis concerns.

3.2- Appreciate the authors comments and change to manuscript. This reviewer asks authors to add to changes that this age is recommended in UK, not in all countries.

3.3- No concerns with this response.

3.4- Appreciate the authors inclusion of further work with regards to sensitivity analysis.

3.5- This is a significant concern but agree with previous reviewer- this information likely is not attainable, and authors have updated limitations.

3.6- This reviewer remains concerned regarding this issue of timing of these labs and values with respect to use of valproate. Disagree strongly with arbitrary issue below. Sperm for instance is matured over 70-90 days, and it is not clear if semen analyses in these patients are taken shortly after valproate initiation or much later. Also feel it would be straightforward to design studies with volunteers on valproate to do semen analyses 30, 60, 90 180, 360 days after starting. Need further editing to this limitation to suggest inability to know impact of valproate for sure on sperm health.

Also, difficult to follow total N for semen analysis values, but seems risk of abnormal semen analyses was significantly low.

Page 7:

"Similarly, almost equal numbers of 25 and 23 in those exposed and unexposed to valproate, respectively, experienced a composite low sperm concentration, motility, vitality, normal forms, or semen volume (0.05% risk in both groups)."

Authors need to comment on this exceptionally low risk of abnormal SA values in both cohorts. I assume risk of abnormal SA values would approximate 10% in public patients presenting for semen analyses.

3.7- This reviewer feels authors have adequately responded to previous comments.

3.8- This reviewer feels authors have inadequately responded to previous comments. Codes used do not fully search for male infertility (See Z31.81), or other organic causes of subfertility (Q53.9, Q55.1, O55.3- some examples).

3.9- This reviewer feels authors have adequately responded to previous comments.

3.10- This reviewer understands previous reviewer comments/concerns. See my concern above at 3.6. Otherwise appreciate the updates authors have made to this request.

3.11- This reviewer feels authors have adequately responded to previous comments.

3.12- This reviewer feels authors have adequately responded to previous comments.

3.13- This reviewer still has concern regarding coding used (See 3.8 above).

3.14- This reviewer feels authors have adequately responded to previous comments.

3.15- This reviewer feels authors have adequately responded to previous comments.

3.16- Appreciate previous reviewer concerns, but as authors point out these changes are not available, so limitations have been added.

Version 2:

Reviewer comments:

Reviewer #4

(Remarks to the Author)

I have carefully read the letter to referees and revised manuscript & I think the authors have addressed referee comments

very thoroughly. In my opinion the manuscript is suitable for publication and represents a significant contribution to knowledge with major clinical application.

Reviewer #6

(Remarks to the Author)

The authors provide findings that support non-association of valproate use and male infertility. This is significant given frequency of valproate usage for multiple pathologies from seizures to bipolar disorder to migraines. Often these diagnoses are made at a young age, thus any deleterious medication effects on fertility are acutely felt. The study is well designed and its conclusions supported by stated results. We agree with prior reviewer sentiments and believe the authors have adequately responded to their recommendations.

Reviewer #7

(Remarks to the Author)

AUTHOR RESPONSES TO REVIEWER QUESTIONS - NCOMMS-24-38348A

Reviewer #1 (Remarks to the Author):

REVIEWER 1.1:

Thank you for the opportunity to review the paper “Infertility outcomes in valproate exposed and unexposed men with epilepsy or bipolar disorder: an international cohort study of real-world healthcare data”. Mbizvo and colleagues provide a comprehensive examination of the potential effects of valproate on male fertility in patients with epilepsy or bipolar disorder using a large retrospective cohort study. The article nicely addresses an important clinical question, especially given its widespread use in treating epilepsy and bipolar disorder, providing results from a large sample size and use of real-world data after conducting a comprehensive adjustment for confounders, despite the potential biases deriving from the retrospective design and the limited validation of outcome measures within the database.

AUTHOR 1.1:

We are grateful to the reviewer this summary, that is both helpful and kind.

REVIEWER 1.2

I have some comments and suggestions to improve the paper:

Introduction:

- When reporting the current evidence on valproate effects on male fertility the paper could benefit from a more structured presentation of the studies' outcomes and limitations. It would be useful to report the level of evidence (instead of “small human studies” in line 33, 66 and others report the design of the studies) of the available literature.

AUTHOR 1.2:

Thank you for this helpful suggestion. In response, we have now incorporated a structured review of the literature, presenting the studies, outcomes, limitations, and levels of evidence in Supplementary Table 1 (see supplementary material). Additionally, a new section summarising this table has been added on page 3 of the Introduction, which reads as follows:

Human literature review

The extent to which valproate actually causes impaired male fertility and/or adverse effects on the testes is largely unknown.^{19,20} The majority of evidence is either in animal models or on small numbers of humans.^{19,20} It is unclear how applicable the results from animal models are to the human experience. Indeed, some of those animal studies have had to use valproate doses seven to 33 times the maximum human-equivalent dose to demonstrate effects such as testicular atrophy and reversible abnormalities in sperm count or motility.^{19,21-23} The advice given to humans cannot be based entirely on animal models. High-quality grade I–II human studies are required (Oxford 2011 Levels of Evidence),²⁴ but are as yet unavailable. Supplementary Table 1 provides a structured overview of the available human studies, outcomes, limitations, and levels of evidence.²⁵⁻⁴² All are observational, spanning grades III–V, with most being case reports or unmatched case-control designs with small numbers of up to 32 valproate-exposed cases and 90 healthy controls. There are no studies using methods from which causality can be more confidently inferred from observational data, such as propensity-score matching, inverse probability weighting, or target trial emulation.⁴³⁻⁴⁵

Some of the studies report adverse effects of valproate on the sexual and reproductive health of men, describing lower sperm count or concentration,^{29-31,39-42} percentage normal sperm,^{27,30,35,38-42} sperm motility,^{26,29-31,33,37,39-42} semen volume,³⁵ and testicular volume.^{30,32} However, other studies do not report lower sperm count or concentration,^{32,33,35,37,38} percentage normal sperm,²⁹ sperm motility,³⁵ semen volume,^{30,33,37,41} and testicular volume.^{34,37} Many of the studies focus on blood hormone levels, but there is variation in findings between studies for the same hormones (see in Supplementary Table 1),^{25,28-30,32-36,38,42} and hormonal profiles do not necessarily correlate with male sexual health and fertility.²⁰ Furthermore, epilepsy, in and of itself, can affect fertility rates, which are two-thirds lower in men with epilepsy than without.⁴⁶ Therefore, it is difficult to be certain if any reduced fertility seen in men on valproate against healthy controls is related to the valproate or the epilepsy. Indeed, other co-prescribed ASMs may be associated with impairment in semen, testicular, or infertility outcomes,^{27,28,32,34-38} indicating a need for these to be adjusted for statistically before making inferences about the independent effect of valproate on these outcomes; but such small studies have precluded adjustment.^{25,26,28-39} Only one study has directly assessed fertility through birth rate.²⁷ This retrospective cohort of claims and registry data in Finland is the largest reported, assessing PWE on valproate (overall $n = 1,546$, monotherapy $n = 1,116$), carbamazepine (overall $n = 2,689$, monotherapy $n = 2,365$), and oxcarbazepine (overall $n = 832$, monotherapy $n = 631$). Comparisons were made to PWE untreated ($n = 2,714-2,785$), and people without epilepsy ($n = 13,378-13,689$). Authors concluded that birth rate was decreased among PWE on ASMs in general, more so in men. Among men with epilepsy, not only valproate, but also carbamazepine and oxcarbazepine were all associated with low birth rate compared to in men without epilepsy. However, only oxcarbazepine retained a reduced birth rate amongst treated PWE compared with untreated PWE. In a study of 17 infertile men on valproate who were switched to levetiracetam ($n = 9$) or lamotrigine ($n = 8$), three were subsequently able to conceive.³³ However, conception occurred whilst the men were still on a reducing regimen of valproate, making it difficult to confidently infer an association between valproate and infertility. Finally, there is only one reported study of reproductive abnormalities in men on valproate with bipolar

disorder.²⁵ This is reported as a full-length article assessing reproductive hormones in 18 men in Turkey with bipolar disorder on valproate monotherapy or in combination with lithium, 15 men with epilepsy on valproate monotherapy, and 21 men with bipolar disorder on lithium monotherapy.²⁵ Valproate did not negatively impact reproductive hormones in the men with bipolar disorder. Elevated levels of prolactin and follicle-stimulating hormone (FSH) were found in men with epilepsy. This was attributed to the epilepsy itself by the study authors. They wrote a follow-up Letter to the Editor assessing semen parameters in 12 men from the study who had consented to this testing: six men with bipolar disorder on lithium monotherapy, five men with bipolar disorder on valproate monotherapy or in combination with lithium, and one man with epilepsy on valproate monotherapy.²⁶ They found reduced sperm count and motility in two out of five men with bipolar disorder on valproate monotherapy or in combination with lithium, and reduced sperm count in the one man with epilepsy on valproate monotherapy. It is challenging to know if any associations, if present, were related to bipolar disorder, epilepsy, lithium, valproate, or a combination of these. A much larger study is needed to help account for these and other potential confounders (covariates) at baseline, allowing more confident inferences to be drawn about any associations.

Reviewer 1.3

- Line 43, when referring to bipolar disorder the format is too like the former sentence. Change to improve flow.

Author 1.3

We note this with thanks, and have re-worded the sentences to the following, on page 2:

“Epilepsy, one of the most common neurological disorders globally, affects over 70 million people.¹ Similarly, bipolar disorder represents a significant portion of the global burden of psychiatric disease, impacting more than 40 million individuals.^{2”}

Reviewer 1.4:

- When referring to the main outcome (e.g. “male fertility and/or adverse effects on the testes”, line 65 and others) the authors should better clarify these definitions. For instance, how do the definitions used (e.g. male infertility [ICD10-CM N46]) match with the WHO definition of infertility?

Author 1.4

The reviewer raises an important point for general clarification. We have used male infertility, testicular hypofunction, and testicular atrophy as defined by the World Health Organisation (WHO) through their International Classification of Diseases coding system. Our thresholds for low sperm concentration, motility, vitality, normal forms, and semen volume are also taken from the WHO, in their 2021 Laboratory Manual for the Examination and Processing of Human Semen – which was co-edited by one of our study authors (M.T.M). We have now clarified these definitions in the text in several areas as follows:

Page 4:

“Lifetime propensity-score matched risks of the following outcomes were assessed between those exposed and unexposed to valproate: **i) male infertility (as defined by the World Health Organisation (WHO) through the International Classification of Diseases 10th Revision Clinical Modification (ICD-10-CM) code N46,**⁵³ – which includes N46.0 (Azoospermia), N46.1 (Oligospermia), N46.8 (Other male infertility – capturing where male infertility has been identified but the cause does not fit into any of the other specified (coded) aetiological categories),⁵⁴ N46.9 (Male infertility, unspecified – capturing where male infertility that has been identified but the cause is unclear),⁵⁵ and 606 (Infertility, male – which is the ICD-9-CM code equivalent of N46, allowing healthcare systems using older coding to still be represented);⁵³ **ii) Testicular hypofunction (ICD-10-CM code E29.1)** – which includes defective biosynthesis of testicular androgen, 5-delta-reductase deficiency (with male pseudohermaphroditism), and testicular hypogonadism;⁵⁶ **iii) Testicular atrophy (ICD-10-CM code N50.0)** – a code which clinicians have the discretion to use when finding evidence of a pathological reduction in size of the testicles, e.g. using an orchidometer or ultrasound);⁵⁷ **iv) A composite measure of laboratory results consisting of low sperm concentration (<16 x 10⁶/ml semen), low sperm motility (<42% total motility), low sperm vitality (<54% alive), low normal forms (<4% of sperm have a normal morphology), and low semen volume (<1.4 mL)** – thresholds taken from the 2021 WHO Laboratory Manual for the Examination and Processing of Human Semen;⁵⁸⁻⁶⁰ and **v) laboratory results for the following reproductive hormone levels in serum: total testosterone (ng/dL), free testosterone (pg/mL), luteinizing hormone (LH, mIU/mL), FSH (mIU/mL), prolactin (ng/mL), and estradiol (pg/mL).**^{25,28-30,32-36,38,42} For hormone profiles, the most recent laboratory result reported during follow-up was used, averaged across the cohort.”

Page 15, under outcomes, in online methods section:

- “
- i) Male infertility (ICD-10-CM code N46);⁵³
 - ii) Testicular hypofunction (ICD-10-CM code E29.1);⁵³
 - iii) Testicular atrophy (ICD-10-CM code N50.0);⁵³
 - iv) A composite measure of low sperm concentration (<16 x 10⁶/ml semen), low sperm motility (<42% total motility), low sperm vitality (<54% alive), low normal forms (<4% of sperm have a normal morphology), and low semen volume (<1.4 mL);⁵⁸⁻⁶⁰

- v) Reproductive hormone levels: total testosterone (ng/dL), free testosterone (pg/mL), LH (mIU/mL), FSH (mIU/mL), prolactin (ng/mL), and estradiol (pg/mL).^{25,28-30,32-36,38,42} The most recent result taken during follow-up was used.

ICD-10-CM is the International Classification of Diseases, Tenth Revision, Clinical Modification, which is a standardised system that is widely used to code diseases and medical conditions amongst HCOs, particularly in the US.⁵³ It is curated by the Centres for Disease Control and Prevention (CDC).⁵³ ICD-10-CM is approved by the WHO and conforms to the WHO's established ICD structure and conventions for statistical classification of disease.⁵³ Healthcare providers use ICD-10-CM codes when diagnosing patients, which are then uploaded to TriNetX directly to allow study data to reflect real-world, real-time, clinical decisions and diagnoses.^{47,48} Full details about how the WHO and CDC define individual ICD-10-CM codes can be found within their respective reference texts.⁵³

REVIEWER 1.5

- Regarding citation 12 (line 66, 67, 69-72, 75), not the comprehensive review, but the specific studies should be cited, since they report different outcomes.

AUTHOR 1.5

Thank you. We have now cited the specific studies, both in the text and in the new Supplementary Table 1 summary.

REVIEWER 1.6

- Line 79-81: "Valproate did not have a negative effect on male reproductive hormones in the men with bipolar disorder, and elevated prolactin and follicle-stimulating hormone levels in men with epilepsy was attributed to the epilepsy". I suggest revising/rephrasing the sentence for clarity.

AUTHOR 1.6

We note this with thanks, and have rephrased the sentences as follows, on page 4:

"Valproate did not negatively impact reproductive hormones in the men with bipolar disorder. Elevated levels of prolactin and follicle-stimulating hormone (FSH) were found in men with epilepsy. This was attributed to the epilepsy itself by the study authors."

REVIEWER 1.7

- Line 77 to 83. When referring to the Turkish study (citation 14) it appears that in "two out of five men" reduced sperm count and motility were described, but the authors state that the study included "18 men with bipolar disorder on valproate monotherapy or in combination with lithium therapy, and 15 men with epilepsy on valproate monotherapy"

AUTHOR 1.7

We agree with the reviewer, this is confusing and requires further clarification that the same study was actually published in two different forms – once as a full-length study assessing reproductive hormones (reporting 18 men with bipolar disorder on valproate monotherapy or valproate in combination with lithium, and 15 men with epilepsy on valproate monotherapy), and then next as a Letter to the Editor assessing semen in a few men who had consented to this (assessing 5 men with bipolar disorder on valproate monotherapy or lithium and valproate, and one man with epilepsy on valproate monotherapy). We have now clarified these details in the Supplementary Table 1, and also amended text on page 4 to:

"Finally, there is only one reported study of reproductive abnormalities in men on valproate with bipolar disorder.²⁵ This is reported as a full-length article assessing reproductive hormones in 18 men in Turkey with bipolar disorder on valproate monotherapy or in combination with lithium, 15 men with epilepsy on valproate monotherapy, and 21 men with bipolar disorder on lithium monotherapy.²⁵ Valproate did not negatively impact reproductive hormones in the men with bipolar disorder. Elevated levels of prolactin and follicle-stimulating hormone (FSH) were found in men with epilepsy. This was attributed to the epilepsy itself by the study authors. They wrote a follow-up Letter to the Editor assessing semen parameters in 12 men from the study who had consented to this testing: six men with bipolar disorder on lithium monotherapy, five men with bipolar disorder on valproate monotherapy or in combination with lithium, and one man with epilepsy on valproate monotherapy.²⁶ They found reduced sperm count and motility in two out of five men with bipolar disorder on valproate monotherapy or in combination with lithium, and reduced sperm count in the one man with epilepsy on valproate monotherapy."

REVIEWER 1.8

Results and discussions:

The results are presented clearly, with specific outcomes measured and hazard ratios reported. It's good to see both the absolute risk changes and the statistical significance (or lack thereof) clearly presented. It could benefit from a comparison with the findings of former studies and a more critical analysis of why the current results differ.

AUTHOR 1.8

The reviewer's point is well received and will certainly help improve the article. The following has now been added to the discussion section on page 9:

“Although clinicians are advised to warn men that valproate is linked to infertility,¹⁴ the evidence surrounding this is limited and contradictory.²⁵⁻⁴² Our results suggest that more caution is needed when advising men on valproate about this side effect in either epilepsy or bipolar disorder. Any evidence that valproate is linked to male infertility in epilepsy seems to emerge when it is compared to healthy men without epilepsy, which were the controls used in most of the prior studies.^{27-30,32,34-38} However, the decision men are faced with is whether to accept valproate or another treatment for their epilepsy or bipolar disorder. Therefore, pragmatic evidence that is directly relevant to this decision is needed, meaning comparisons should be between drug exposed and unexposed men with the disease, as was reported in the current study. In such circumstances, the evidence we present suggests valproate may not be linked to infertility over and above other ASMs or mood-stabilisers, nor over the epilepsy or bipolar disorder themselves. This is consistent with findings in the largest previous study of valproate and male fertility. This was a national study of Finnish health data in which 1,116 PWE on valproate were compared to 13,378–13,689 healthy people without epilepsy and 2,714–2,785 people with epilepsy that was untreated, adjusted for age and marital status.²⁷ Birth rate was lower in men with epilepsy on valproate (and those on carbamazepine or oxcarbazepine) when compared to men without epilepsy only. There was no difference in birth rate between men with epilepsy on valproate (or carbamazepine) and men with epilepsy who were untreated, illustrating the substantial impact epilepsy has as a confounder. Only oxcarbazepine was linked to reduced birth rate amongst treated PWE compared with untreated PWE.²⁷”

The predominant use of healthy controls in many of the prior studies is likely to be one of the reasons our study results differ from many of those prior studies.^{28-30,32,34-38} Additional reasons include our study being much larger than the prior studies,^{25,26,28-42} and the fact we were able to propensity-score match cohorts by a substantial number of confounders, many of which were unmeasured or excluded in prior studies (see Supplementary Table 1). This will have decreased the chances of an overestimation or underestimation of the real-world effect of valproate in our study when compared to prior studies. Our study's use of a wide range of relevant comorbidity, treatment and lifestyle conditions as propensity-score matched covariates (as opposed to excluding them from study, as was done in most prior studies)^{25,29,30,32-38} will have increased the real-world generalisability of our study results to frontline clinical settings – where many men seeking counselling about valproate use are not without some prior comorbidity, treatment or lifestyle conditions potentially affecting their fertility. The applicability of conclusions drawn from prior studies to patient groups excluded from those studies is limited. Nevertheless, we were also able to undertake a sensitivity analysis in which we excluded men with a prior history of key modifiable treatment and lifestyle risk factors for infertility, and this did not alter our overall conclusions. A recently published systematic review was able to undertake a meta-analysis across five out of the 19 human studies we identified in Supplementary Table 1, applying a random effect model to men with epilepsy on valproate against healthy controls for sperm count, motility, normal percentage of sperm, FSH and LH levels.⁶⁵ Sperm motility alone was significantly reduced in men on valproate (Overall SMD = -1.62, 95% CI: -2.81 to -0.43, $p = 0.033$). There were no significant differences in sperm count (Overall SMD = -0.78, 95% CI: -1.58 to 0.03, $p = 0.286$), percentage of abnormal sperm (Overall SMD = 0.93, 95% CI: -0.97 to 2.84, $p = 0.616$), levels of FSH (Overall SMD = -1.32, 95% CI: -2.93 to 0.29, $p = 0.198$) and LH (Overall SMD = -0.96, 95% CI: -1.95 to 0.04, $p = 0.211$). Interestingly, the authors concluded that men with epilepsy on valproate exhibit abnormal male fertility factors including sperm count, motility, percentage of abnormal sperm, FSH and LH. However, we note that their findings did not actually support a statistically significant change in any of these factors beyond motility.”

REVIEWER 1.9

- Line 93. When referring to the composite measure, please describe how these semen parameters were chosen among others usually taken into consideration (e.g. morphology, vitality).

AUTHOR 1.9

We based our selection of semen parameters on those listed in the 2021 WHO Laboratory Manual for the Examination and Processing of Human Semen, prioritising those with the most available data in TriNetX and routinely utilised in clinical practice (based on our clinical experience). Initially, morphology and vitality were not included due to limited data availability and their less frequent assessment in routine clinical settings outside fertility treatment. However, in response to the reviewer's concern, we have now included these parameters and described the TriNetX data availability for each semen parameter measured in the manuscript. The relevant text excerpts are as follows:

Page 4:

“**iv) A composite measure of laboratory results consisting of low sperm concentration** (<16 x 10⁶/ml semen), **low sperm motility** (<42% total motility), **low sperm vitality** (<54% alive), **low normal forms** (<4% of sperm have a normal morphology), **and low semen volume** (<1.4 mL) – thresholds taken from the 2021 WHO Laboratory Manual for the Examination and Processing of Human Semen;⁵⁸⁻⁶⁰”

Page 6:

“four more experienced a composite low sperm concentration, motility, vitality, normal forms, or semen volume (a 0.02% change in risk).”

And

“composite low sperm concentration, motility, vitality, normal forms, or semen volume (HR 0.913; 95% CI 0.621 to 1.342)”

Page 7:

“Similarly, almost equal numbers of 25 and 23 in those exposed and unexposed to valproate, respectively, experienced a composite low sperm concentration, motility, vitality, normal forms, or semen volume (0.05% risk in both groups).”

And

“and composite low sperm concentration, motility, vitality, normal forms, or semen volume (HR 1.632; 95% CI 0.803 to 3.319).”

Page 8:

“and low sperm concentration, motility, vitality, normal forms, or semen volume (HR 1.594; 95% CI 0.598 to 4.252).”

And

“The database was able to detect a significant increase in all male infertility outcomes assessed in association with exposure to these drugs: male infertility HR 3.036; 95% CI 2.214 to 4.163, testicular hypofunction HR 6.681; 95% CI 5.658 to 7.889, testicular atrophy HR 3.236; 95% CI 1.759 to 5.954, and low sperm concentration, motility, vitality, normal forms, or semen volume HR 2.425; 95% CI 1.348 to 4.362.”

Page 9:

“The number of men aged <55 years with laboratory results available on semen parameters in TriNetX is 58,641 for semen volume, 45,482 for sperm motility, 35,507 for sperm concentration, 3,287 for sperm vitality, and 2,166 for normal forms.”

REVIEWER 1.10

- Line 94. The authors could clarify why the index event was defined as stated in the article (e.g. second recorded valproate prescription to establish chronic exposition? Second recorded epilepsy or bipolar disease code to exclude misdiagnosis?).

AUTHOR 1.10

This is a helpful suggestion. We have now clarified the rationale for this strategy in the text on page 5:

“The index event was defined as the second recorded valproate prescription in participants exposed to the drug, alongside the presence of at least two epilepsy or bipolar disorder codes, or the second recorded epilepsy or bipolar disease code for those unexposed to valproate. The rationale for using the second prescription was to ensure repeat valproate exposure, providing a more robust indication of chronic use rather than incidental or one-off exposure. We then used the proportion of the cohort with detectable serum valproate levels during follow-up to infer ongoing exposure and adherence. Requiring two diagnostic codes for epilepsy or bipolar disorder allowed us to minimise the risk of misdiagnosis by confirming the condition through repeated coding. Additionally, this approach enabled us to more accurately balance cohorts based on diagnosis, as the first recorded epilepsy or bipolar disorder code was then used as a matching covariate in the propensity-score matching process (including matching by type of epilepsy using the respective G40.- subgroups, as listed in Supplementary Tables 2–6)). This approach was designed to help improve cohort comparability and reduce the potential of bias from misclassification.”

The is also reflected in the text on page 17 of the online methods:

“Cohorts were propensity-score matched for all of the baseline covariates listed in Supplementary Tables 2–6 at the start follow-up (time 0: corresponding to the index event). The index event was taken as the second recorded valproate prescription in those exposed to the drug, or the second recorded epilepsy or bipolar disease code in those unexposed to the drug. This allowed cohorts to be propensity-score matched for epilepsy and bipolar disorder using the first code to appear for these conditions. It also ensured men with only a single exposure to the valproate or single seizure were excluded.”

REVIEWER 1.11

- Line 189. Please state the n. or % of patients who had laboratory results on semen parameters available.

AUTHOR 1.11

We have now included this information on page 9, as detailed at the end of the AUTHOR 1.9 response.

REVIEWER 1.12

- Line 279. Authors should add as a possible limitation that considering the two prescriptions of valproate and the ICD code might miss possible epilepsy/BD misdiagnosis, reconsidered during the follow-up time.

AUTHOR 1.12

This limitation is reasonable to consider, albeit the diagnostic accuracy validation work we have undertaken previously suggests the likelihood of this is low. Nevertheless, it is good for us to acknowledge the possibility, as helpfully suggested by the reviewer. We have now included the following text on page 11, as part of the limitations section of the discussion:

“Another limitation of this study relates to the potential inaccuracies of healthcare coding. From a diagnostic accuracy perspective, we mitigated this by ensuring we used codes that have been validated previously as reliably able to identify epilepsy or bipolar disorder within healthcare datasets.^{49-52,84} This helps to minimise diagnostic misclassification risks. For example, it is possible the use of two prescriptions of valproate and disease coding for epilepsy or bipolar disorder might have generated false positives by misclassifying cases as disease-positive when clinicians have subsequently reconsidered the diagnosis as negative. However, the diagnostic accuracy studies have generally shown that the false-positive rate for epilepsy disease and symptom coding combined with ASMs or bipolar disease coding is low, generating generally high PPVs of >80%, giving us confidence in the probability of few diagnostic misclassifications having occurred.^{49-52,84}”

Also relevant to the above points is the section on case-ascertainment accuracy on page 15 of the online methods:

“Case-ascertainment accuracy

Studies validating the diagnostic accuracy of ICD-10 coding for epilepsy or bipolar disorder in TriNetX *specifically* are as yet unavailable. However, more widely, there are a substantial number of studies validating the diagnostic accuracy of these codes, including in regions and settings covered by TriNetX.⁴⁹ As we showed previously, a case-ascertainment strategy combining disease coding (G40, including epilepsy and status epilepticus) and/or symptom coding (R56.9) with co-prescribed ASMs is a valid way to accurately capture epilepsy within electronic healthcare datasets, and this spans multiple studies in a variety of healthcare settings and countries.⁴⁹ This strategy tends to achieve figures of 80–90% across positive predictive values (PPVs) and sensitivity estimates, and negative predictive values (NPV) and specificity estimates approach 100%.^{49,50} The corresponding F1 scores are also high at 0.86.⁸⁷ F31 codes are also known to accurately capture bipolar disorder, with PPVs, sensitivities and NPVs reaching 84%, 89%, and 98%, respectively.⁸⁴”

REVIEWER 1.13

Tables and Figures:

The tables summarizing the results are well-organized and informative. Kaplan-Meier plots aid in visualizing the survival analysis results.

Table 1:

- Include p-value along with HR.

AUTHOR 1.13

We have now included Log-Rank *p*-values alongside the HRs in Table 1.

REVIEWER 1.14

- There is a mismatch between the total of patients with epilepsy and/or bipolar disorder and sum of the number of patients in each group. The mismatch is probably due to patients with epilepsy and bipolar disorder. Please confirm that or give an alternative explanation.

AUTHOR 1.14

The reviewer is correct to point out that there is a mismatch, and the reasoning is as deduced by the reviewer. We have now included this information as a note in table 1, which reads as follows:

“N.B: An expected mismatch is noted between the total number of patients with epilepsy and/or bipolar disorder and sum of the number of patients in each subgroup. Since cohorts were not designed to be mutually exclusive, this mismatch is due to the ability for patients to have record of one or both diagnoses (epilepsy and bipolar disorder) in their electronic health record.”

REVIEWER 1.15

Minor Comments:

- Abstract: while not mandatory, reporting p-value (line 39, $p < 0.05$) could be useful when stating the non-significance of the results.

AUTHOR 1.15

This is a helpful suggestion. We have now included p-values in the abstract. This now includes the following, on page 2:

“Those exposed to valproate demonstrate a <1% increased risk of each of these outcomes compared to those unexposed to valproate, and the difference is not statistically significant (Log-Rank $p > 0.05$ for each outcome).”

REVIEWER 1.16

- Line 274. Standardized mean difference, use SMD as abbreviation.

AUTHOR 1.16

Thanks. We have now used SMD when abbreviating standardized mean difference throughout the text.

REVIEWER 1.17

- In the notes: "552 patients in Cohort 1 was excluded". Correct with "were".
- In the notes: "815 patients in Cohort 1 was excluded". Correct with "were".
- In the notes: "1290 patients in Cohort 1 was excluded". Correct with "were".

AUTHOR 1.17

We thank the reviewer for helpfully noticing these typographical errors, which we have now corrected.

REVIEWER 1.18

- Line 118. Define HCOs (Healthcare organization [HCO]). Prefer "clinical data" to "clinical facts".

AUTHOR 1.18

Thank you for pointing this out. We have now defined healthcare organisations as HCO after the first use. We felt "clinical data points" would work best instead of "clinical facts", so we have made this change as well.

REVIEWER 1.19

- Line 122. When expressing mean age use mean (SD).

AUTHOR 1.19

We note this with thanks, and have now used mean (SD) for all means reported throughout the text.

REVIEWER 1.20

- Line 126: "compared to epilepsy or bipolar". Add "patients".

AUTHOR 1.20

Thank you. We have now written this out to read as follows, on page 6:

"These changes, each less than 1%, were not statistically significant compared to in patients with epilepsy or bipolar disorder"

We thought this would work even better than "epilepsy or bipolar patients"

REVIEWER 1.21

- Line 127, 149, 156: When expressing the results for the HR and 95% CI for the three groups I suggest introducing the data as follows: "HR with 95% CI for were obtained for the following outcomes: ...".

Also, I suggest changing the format to express the numbers. Use for instance "male infertility (HR 1.063; 95% CI 0.828 to 1.364)".

AUTHOR 1.21

We thank the reviewer for these suggestions. The issue we encountered with the suggested structure, "HR with 95% CI for were obtained for the following outcomes: ...," is that the outcomes have already been introduced in the preceding sentence: "In absolute terms, 31 more men with epilepsy or bipolar disorder experienced infertility with valproate exposure (a 0.04% change in risk), 80 more experienced testicular hypofunction (a 0.10% change in risk), 12 more experienced testicular atrophy (a 0.02% change in risk), and four more experienced a composite low sperm concentration, motility, vitality, normal forms, or semen volume (a 0.02% change in risk)." Following this with "HR with 95% CI for were obtained for the following outcomes..." could imply that new or different outcomes are being introduced. Respectfully, we felt these two sentence structures were potentially incompatible. However, we agree a revision was needed for clarity. Accordingly, we incorporated the reviewer's second suggestion and revised the sentences as follows, on page 6:

"In absolute terms, 31 more men with epilepsy or bipolar disorder experienced infertility with valproate exposure (a 0.04% change in risk), 80 more experienced testicular hypofunction (a 0.10% change in risk), 12 more experienced testicular atrophy (a 0.02% change in risk), and four more experienced a composite low sperm concentration, motility, vitality, normal forms, or semen volume (a 0.02% change in risk). These changes, each less than 1%, were not statistically significant compared to in patients with epilepsy or bipolar disorder unexposed to valproate, as evidenced by the corresponding HR with 95% CI results for male infertility (HR 0.978; 95% CI 0.797 to 1.201), testicular hypofunction (HR 0.947; 95% CI 0.850 to 1.056), testicular atrophy (HR 1.051; 95% CI 0.702 to 1.574), and composite low sperm concentration, motility, vitality, normal forms, or semen volume (HR 0.913; 95% CI 0.621 to 1.342)."

REVIEWER 1.22

- Line 164. Authors could consider rephrasing “Following this, there is little evidence demonstrated to suggest an association between valproate and male infertility” with “Subsequently, there is minimal evidence to suggest an association between valproate and male infertility.

AUTHOR 1.22

Thanks. We have made this change exactly as advised, on page 8.

REVIEWER 1.23

- Line 182. Change “seems to be when...” with “seems to emerge when”.

AUTHOR 1.23

Thanks. We have made this change exactly as advised, on page 9.

REVIEWER 1.24

- Line 97. When introducing the “>120 baseline covariates” please refer to the supplementary table. 6. Same thing when describing drugs known to cause infertility (line 153) used in the sensitivity analysis, refer to Supplementary Table. 5

AUTHOR 1.24

We have now referred to the relevant tables as follows:

Page 5:

“The comparison groups were propensity-score matched for >140 baseline covariates before index (see Supplementary Tables 2–6 for a full list of these baseline covariates pre- and post-propensity-score matching).

(Note that we have expanded the covariate list in response to reviewer two’s suggestion to include type of epilepsy subgroups as covariates, and reviewer three’s suggestion to include lifestyle factors as covariates.)

Page 8:

“The database sensitivity analysis Search Five identified a cohort of 530,518 men with epilepsy or bipolar disorder exposed (n = 27,925) and unexposed (n = 502,593) to drugs known to cause infertility. The propensity-score matching results are shown in Supplementary Table 6.”

REVIEWER 1.25

Supplementary Table 1:

- Add the total n. of patients with available valproate serum level.

Supplementary Table 2:

- Add the total n. of patients with available valproate serum level.

Supplementary Table 3

- Add the total n. of patients with available valproate serum level.

AUTHOR 1.25

We note this with thanks. We have now included the followed sentence below supplementary tables 7 to 10

“▪ Total number of men <55 years of age with serum valproate levels available in TriNetX = 125,453”

REVIEWER 1.26

Supplementary Table 5:

- Please add a note explaining the codes that are reported along with drugs categories and specific drugs (e.g. ATCL01 Antineoplastic agents)

AUTHOR 1.26

We appreciate that the reviewer wishes for full details about the codes, drug categories, and specific drugs to be expanded. Whilst this will substantially increase the length and complexity of text in this section, we agree that such details will be helpful for readers. Therefore, we have now added the relevant material to this section, which now reads as follows:

Supplementary Table 11: Drugs known to cause male infertility

- **ATC L01 Antineoplastic agents**
 - ATC L01A Alkylating agents
 - ATC L01AA Nitrogen mustard analogues
 - RxNorm 134547 bendamustine
 - RxNorm 2346 chlorambucil

- RxNorm 2531369 melphalan flufenamide
 - RxNorm 3002 cyclophosphamide
 - RxNorm 5657 ifosfamide
 - RxNorm 6674 mechlorethamine
 - RxNorm 6718 melphalan
- ATC L01AB Alkyl sulfonates
 - RxNorm 1828 busulfan
 - RxNorm 38508 treosulfan
- ATC L01AC Ethylene imines
 - RxNorm 10473 thiotepa
- ATC L01AD Nitrosoureas
 - RxNorm 10114 streptozocin
 - RxNorm 2105 carmustine
 - RxNorm 6466 lomustine
- ATC L01AX Other alkylating agents
 - RxNorm 3098 dacarbazine
 - RxNorm 37776 temozolomide
- ATC L01B Antimetabolites
 - ATC L01BA Folic acid analogues
 - RxNorm 196239 raltitrexed
 - RxNorm 662019 pralatrexate
 - RxNorm 68446 pemetrexed
 - RxNorm 6851 methotrexate
 - ATC L01BB Purine analogues
 - RxNorm 103 mercaptopurine
 - RxNorm 10485 thioguanine
 - RxNorm 24698 fludarabine
 - RxNorm 274771 nelarabine
 - RxNorm 44151 clofarabine
 - RxNorm 44157 Cladribine
 - ATC L01BC Pyrimidine analogues
 - RxNorm 1251 azacitidine
 - RxNorm 12574 gemcitabine
 - RxNorm 15657 decitabine
 - RxNorm 194000 capecitabine
 - RxNorm 3041 cytarabine
 - RxNorm 4488 floxuridine
 - RxNorm 4492 fluorouracil
 - RxNorm 4582 tegafur
- ATC L01C Plant alkaloids and other natural products
 - ATC L01CA Vinca alkaloids and analogues
 - RxNorm 11198 vinblastine
 - RxNorm 11202 vincristine
 - RxNorm 11204 vindesine
 - RxNorm 39541 vinorelbine
 - ATC L01CB Podophyllotoxin derivatives
 - RxNorm 10362 teniposide
 - RxNorm 4179 etoposide
 - ATC L01CD Taxanes
 - RxNorm 56946 paclitaxel
 - RxNorm 72962 docetaxel
 - RxNorm 996051 cabazitaxel
 - ATC L01CE Topoisomerase 1 (TOP1) inhibitors
 - RxNorm 51499 irinotecan
 - RxNorm 57308 topotecan
 - ATC L01CX Other plant alkaloids and natural products
 - RxNorm 1716278 trabectedin
- ATC L01D Cytotoxic antibiotics and related substances
 - ATC L01DA Actinomycines
 - RxNorm 3100 dactinomycin
 - ATC L01DB Anthracyclines and related substances

- RxNorm 3109 daunorubicin
- RxNorm 31435 valrubicin
- RxNorm 3639 doxorubicin
- RxNorm 3995 epirubicin
- RxNorm 5650 idarubicin
- RxNorm 7005 mitoxantrone
- ATC L01DC Other cytotoxic antibiotics
 - RxNorm 1622 bleomycin
 - RxNorm 337523 ixabepilone
 - RxNorm 632 mitomycin
 - RxNorm 6995 plicamycin
- ATC L01E Protein kinase inhibitors
 - ATC L01EA BCR-ABL tyrosine kinase inhibitors
 - RxNorm 1307619 bosutinib
 - RxNorm 1364347 ponatinib
 - RxNorm 2584304 asciminib
 - RxNorm 282388 imatinib
 - RxNorm 475342 dasatinib
 - RxNorm 662281 nilotinib
 - ATC L01EB Epidermal growth factor receptor (EGFR) tyrosine kinase inhibitors
 - RxNorm 1430438 afatinib
 - RxNorm 1721560 osimertinib
 - RxNorm 2058849 dacomitinib
 - RxNorm 2570736 mobocertinib
 - RxNorm 328134 gefitinib
 - RxNorm 337525 erlotinib
 - ATC L01EC B-Raf serine-threonine kinase (BRAF) inhibitors
 - RxNorm 1147220 vemurafenib
 - RxNorm 1424911 dabrafenib
 - RxNorm 2049106 encorafenib
 - ATC L01ED Anaplastic lymphoma kinase (ALK) inhibitors
 - RxNorm 1148495 crizotinib
 - RxNorm 1535457 ceritinib
 - RxNorm 1727455 alectinib
 - RxNorm 1921217 brigatinib
 - RxNorm 2103164 lorlatinib
 - ATC L01EE Mitogen-activated protein kinase (MEK) inhibitors
 - RxNorm 1425099 trametinib
 - RxNorm 1722365 cobimetinib
 - RxNorm 2049122 binimetinib
 - RxNorm 2289380 selumetinib
 - ATC L01EF Cyclin-dependent kinase (CDK) inhibitors
 - RxNorm 1601374 palbociclib
 - RxNorm 1873916 ribociclib
 - RxNorm 1946825 abemaciclib
 - ATC L01EG Mammalian target of rapamycin (mTOR) kinase inhibitors
 - RxNorm 141704 everolimus
 - RxNorm 35302 sirolimus
 - RxNorm 657797 temsirolimus
 - ATC L01EH Human epidermal growth factor receptor 2 (HER2) tyrosine kinase inhibitors
 - RxNorm 1940643 neratinib
 - RxNorm 2361285 tucatinib
 - RxNorm 480167 lapatinib
 - ATC L01EJ Janus-associated kinase (JAK) inhibitors
 - RxNorm 1193326 ruxolitinib
 - RxNorm 2197490 fedratinib
 - RxNorm 2595243 pacritinib
 - RxNorm 2665204 momelotinib
 - ATC L01EK Vascular endothelial growth factor receptor (VEGFR) tyrosine kinase inhibitors
 - RxNorm 1242999 axitinib
 - RxNorm 2534233 tivozanib

- RxNorm 2670179 fruquintinib
- ATC L01EL Bruton's tyrosine kinase (BTK) inhibitors
 - RxNorm 1442981 ibrutinib
 - RxNorm 1986808 acalabrutinib
 - RxNorm 2262435 zanubrutinib
 - RxNorm 2629338 pirtobrutinib
- ATC L01EM Phosphatidylinositol-3-kinase (Pi3K) inhibitors
 - RxNorm 1544460 idelalisib
 - RxNorm 1945077 copanlisib
 - RxNorm 2058509 duvelisib
 - RxNorm 2169285 alpelisib
- ATC L01EN Fibroblast growth factor receptor (FGFR) tyrosine kinase inhibitors
 - RxNorm 2123125 erdafitinib
 - RxNorm 2359268 pemigatinib
 - RxNorm 2550729 infigratinib
 - RxNorm 2628190 futibatinib
- ATC L01EX Other protein kinase inhibitors
 - RxNorm 1098413 vandetanib
 - RxNorm 1312397 regorafenib
 - RxNorm 1363268 cabozantinib
 - RxNorm 1592737 nintedanib
 - RxNorm 1603296 lenvatinib
 - RxNorm 1919083 midostaurin
 - RxNorm 2105628 larotrectinib
 - RxNorm 2105806 gilteritinib
 - RxNorm 2123125 erdafitinib
 - RxNorm 2183102 pexidartinib
 - RxNorm 2197862 entrectinib
 - RxNorm 2272107 avapritinib
 - RxNorm 2359268 pemigatinib
 - RxNorm 2362165 capmatinib
 - RxNorm 2369389 ripretinib
 - RxNorm 2370147 selpercatinib
 - RxNorm 2394936 pralsetinib
 - RxNorm 2477103 tepotinib
 - RxNorm 2478439 umbralisib
 - RxNorm 2643048 quizartinib
 - RxNorm 2669967 capivasertib
 - RxNorm 357977 sunitinib
 - RxNorm 495881 sorafenib
 - RxNorm 714438 pazopanib
- ATC L01F Monoclonal antibodies and antibody drug conjugates
 - ATC L01FA CD20 (Clusters of Differentiation 20) inhibitors
 - RxNorm 121191 rituximab
 - RxNorm 712566 ofatumumab
 - RxNorm 974779 obinutuzumab
 - ATC L01FB CD22 (Clusters of Differentiation 22) inhibitors
 - RxNorm 1942950 inotuzumab ozogamicin
 - RxNorm 2099295 moxetumomab pasudotox
 - ATC L01FC CD38 (Clusters of Differentiation 38) inhibitors
 - RxNorm 1721947 daratumumab
 - RxNorm 2282018 isatuximab
 - ATC L01FD HER2 (Human Epidermal Growth Factor Receptor 2) inhibitors
 - RxNorm 1298944 pertuzumab
 - RxNorm 224905 trastuzumab
 - RxNorm 2473851 margetuximab
 - ATC L01FF PD-1/PDL-1 (Programmed cell death protein 1/death ligand 1) inhibitors
 - RxNorm 1547545 pembrolizumab
 - RxNorm 1597876 nivolumab
 - RxNorm 1792776 atezolizumab
 - RxNorm 1875534 avelumab

- RxNorm 1919503 durvalumab
- RxNorm 2058826 cemiplimab
- RxNorm 2539967 dostarlimab
- RxNorm 2632981 retifanlimab
- RxNorm 2669406 toripalimab
- RxNorm 2677426 tislelizumab
- ATC L01FG VEGF/VEGFR (Vascular Endothelial Growth Factor) inhibitors
 - RxNorm 1535922 ramucirumab
 - RxNorm 253337 bevacizumab
- ATC L01FX Other monoclonal antibodies and antibody drug conjugates
 - RxNorm 1094833 ipilimumab
 - RxNorm 1147320 brentuximab vedotin
 - RxNorm 1294580 gemtuzumab ozogamicin
 - RxNorm 1597258 blinatumomab
 - RxNorm 1726104 elotuzumab
 - RxNorm 1855735 olaratumab
 - RxNorm 2054068 mogamulizumab
 - RxNorm 2174090 polatuzumab vedotin
 - RxNorm 2268307 enfortumab
 - RxNorm 2360231 sacituzumab
 - RxNorm 2387211 tafasitamab
 - RxNorm 2387834 belantamab mafodotin
 - RxNorm 2474039 naxitamab
 - RxNorm 2540964 loncastuximab tesirine
 - RxNorm 2549199 amivantamab
 - RxNorm 2571095 tisotumab
 - RxNorm 2619313 tremelimumab
 - RxNorm 2619426 teclistamab
 - RxNorm 2621548 mirvetuximab soravtansine
 - RxNorm 2625122 mosunetuzumab
 - RxNorm 2637392 epcoritamab
 - RxNorm 2639782 glofitamab
 - RxNorm 2644447 talquetamab
- ATC L01FY Combinations of monoclonal antibodies and antibody drug conjugates
 - RxNorm 2596773 relatlimab
- ATC L01X Other antineoplastic agents
 - ATC L01XA Platinum compounds
 - RxNorm 2555 cisplatin
 - RxNorm 32592 oxaliplatin
 - RxNorm 40048 carboplatin
 - ATC L01XB Methylhydrazines
 - RxNorm 8702 procarbazine
 - ATC L01XC Monoclonal antibodies (deprecated 2022)
 - RxNorm 1094833 ipilimumab
 - RxNorm 1147320 brentuximab vedotin
 - RxNorm 121191 rituximab
 - RxNorm 1294580 gemtuzumab ozogamicin
 - RxNorm 1298944 pertuzumab
 - RxNorm 1371041 ado-trastuzumab emtansine (deprecated 2020)
 - RxNorm 1535922 ramucirumab
 - RxNorm 1547545 pembrolizumab
 - RxNorm 1597258 blinatumomab
 - RxNorm 1597876 nivolumab
 - RxNorm 1721947 daratumumab
 - RxNorm 1723738 necitumumab
 - RxNorm 1726104 elotuzumab
 - RxNorm 1792776 atezolizumab
 - RxNorm 1855735 olaratumab
 - RxNorm 1875534 avelumab
 - RxNorm 1919503 durvalumab
 - RxNorm 1942950 inotuzumab ozogamicin

- RxNorm 2054068 mogamulizumab
- RxNorm 2058826 cemiplimab
- RxNorm 2099295 moxetumomab pasudotox
- RxNorm 2174090 polatuzumab vedotin
- RxNorm 224905 trastuzumab
- RxNorm 2268307 enfortumab
- RxNorm 2282018 isatuximab
- RxNorm 2387211 tafasitamab
- RxNorm 2387834 belantamab mafodotin
- RxNorm 253337 bevacizumab
- RxNorm 2539967 dostarlimab
- RxNorm 263034 panitumumab
- RxNorm 318341 cetuximab
- RxNorm 712566 ofatumumab
- RxNorm 974779 obinutuzumab
- ATC L01XD Sensitizers used in photodynamic/radiation therapy
 - RxNorm 23066 dihematoporphyrin ether
 - RxNorm 337068 methyl 5-aminolevulinate
 - RxNorm 683 aminolevulinic acid
- ATC L01XE Protein kinase inhibitors (deprecated 2021)
 - RxNorm 1098413 vandetanib
 - RxNorm 1147220 vemurafenib
 - RxNorm 1148495 crizotinib
 - RxNorm 1193326 ruxolitinib
 - RxNorm 1242999 axitinib
 - RxNorm 1307619 bosutinib
 - RxNorm 1312397 regorafenib
 - RxNorm 1363268 cabozantinib
 - RxNorm 1364347 ponatinib
 - RxNorm 141704 everolimus
 - RxNorm 1424911 dabrafenib
 - RxNorm 1425099 trametinib
 - RxNorm 1430438 afatinib
 - RxNorm 1442981 ibrutinib
 - RxNorm 1535457 ceritinib
 - RxNorm 1592737 nintedanib
 - RxNorm 1601374 palbociclib
 - RxNorm 1603296 lenvatinib
 - RxNorm 1721560 osimertinib
 - RxNorm 1722365 cobimetinib
 - RxNorm 1727455 alectinib
 - RxNorm 1873916 ribociclib
 - RxNorm 1919083 midostaurin
 - RxNorm 1921217 brigatinib
 - RxNorm 1940643 neratinib
 - RxNorm 1946825 abemaciclib
 - RxNorm 1986808 acalabrutinib
 - RxNorm 2049106 encorafenib
 - RxNorm 2049122 binimetinib
 - RxNorm 2058849 dacomitinib
 - RxNorm 2103164 lorlatinib
 - RxNorm 2105628 larotrectinib
 - RxNorm 2105806 gilteritinib
 - RxNorm 2197490 fedratinib
 - RxNorm 2197862 entrectinib
 - RxNorm 282388 imatinib
 - RxNorm 328134 gefitinib
 - RxNorm 337525 erlotinib
 - RxNorm 357977 sunitinib
 - RxNorm 475342 dasatinib

- RxNorm 480167 lapatinib
- RxNorm 495881 sorafenib
- RxNorm 657797 temsirolimus
- RxNorm 662281 nilotinib
- RxNorm 714438 pazopanib
- ATC L01XF Retinoids for cancer treatment
 - RxNorm 10753 tretinoin
 - RxNorm 233272 bexarotene
 - RxNorm 81864 alitretinoin
- ATC L01XG Proteasome inhibitors
 - RxNorm 1302966 carfilzomib
 - RxNorm 1723735 ixazomib
 - RxNorm 358258 bortezomib
- ATC L01XH Histone deacetylase (HDAC) inhibitors
 - RxNorm 1543543 belinostat
 - RxNorm 1603350 panobinostat
 - RxNorm 194337 vorinostat
 - RxNorm 877510 romidepsin
- ATC L01XJ Hedgehog pathway inhibitors
 - RxNorm 1242987 vismodegib
 - RxNorm 1659191 sonidegib
 - RxNorm 2105845 glasdegib
- ATC L01XK Poly (ADP-ribose) polymerase (PARP) inhibitors
 - RxNorm 1597582 olaparib
 - RxNorm 1862579 rucaparib
 - RxNorm 1918231 niraparib
 - RxNorm 2099704 talazoparib
- ATC L01XL Antineoplastic cell and gene therapy
 - RxNorm 1316105 herpesvirus 1, human
 - RxNorm 1986438 tisagenlecleucel
 - RxNorm 1987398 axicabtagene ciloleucel
 - RxNorm 2387277 brexucabtagene autoleucel
 - RxNorm 2479136 lisocabtagene maraleucel
 - RxNorm 2536430 idecabtagene vicleucel
 - RxNorm 2594775 ciltacabtagene autoleucel
 - RxNorm 2644436 nadofaragene firadenovec
- ATC L01XX Other antineoplastic agents
 - RxNorm 1045453 eribulin
 - RxNorm 1156 asparaginase
 - RxNorm 1232150 aflibercept
 - RxNorm 1316105 herpesvirus 1, human
 - RxNorm 140587 celecoxib
 - RxNorm 1494066 miltefosine
 - RxNorm 1747556 venetoclax
 - RxNorm 18330 arsenic trioxide
 - RxNorm 1940332 enasidenib
 - RxNorm 1986438 tisagenlecleucel
 - RxNorm 1987398 axicabtagene ciloleucel
 - RxNorm 2049873 ivosidenib
 - RxNorm 2109055 tagraxofusp
 - RxNorm 2178390 selinexor
 - RxNorm 2274378 tazemetostat
 - RxNorm 2374729 lurbinectedin
 - RxNorm 2550714 sotorasib
 - RxNorm 2567226 belzutifan
 - RxNorm 2590743 tebentafusp
 - RxNorm 2625882 adagrasib
 - RxNorm 27100 omacetaxine mepesuccinate
 - RxNorm 34132 pegaspargase
 - RxNorm 4089 estramustine
 - RxNorm 5296 altretamine

- RxNorm 5552 hydroxyurea
 - RxNorm 569 eflornithine
 - RxNorm 596724 anagrelide
 - RxNorm 7004 mitotane
 - RxNorm 739 amsacrine
 - RxNorm 8011 pentostatin
- **ATC G03B Androgens**
 - ATC G03BA 3-oxoandrogen (4) derivatives
 - RxNorm 10379 testosterone
 - RxNorm 4494 fluoxymesterone
 - RxNorm 6904 methyltestosterone
 - ATC G03BB 5-androstanon (3) derivatives
 - RxNorm 6781 mesterolone
 - **ATC G03C Oestrogens**
 - RxNorm 9066 quinestrol
 - ATC G03CA Natural and semisynthetic oestrogens, plain
 - RxNorm 4083 estradiol
 - RxNorm 4094 estriol
 - RxNorm 4099 estrogens, conjugated (USP)
 - RxNorm 4103 estrone
 - RxNorm 4124 ethinyl estradiol
 - ATC G03CB Synthetic oestrogens, plain
 - RxNorm 3368 dienestrol
 - RxNorm 3390 diethylstilbestro
 - ATC G03CC Oestrogens, combinations with other drugs
 - RxNorm 3368 dienestrol
 - RxNorm 3390 diethylstilbestrol
 - RxNorm 4094 estriol
 - RxNorm 4103 estrone
 - ATC G03CX Other oestrogens
 - RxNorm 38260 tibolone
 - **Spirolactone**
 - RxNorm 9997 spironolactone
 - **ATC:L02BB Anti-androgens**
 - RxNorm 1307298 enzalutamide
 - RxNorm 1999574 apalutamide
 - RxNorm 2180325 darolutamide
 - RxNorm 31805 nilutamide
 - RxNorm 4508 flutamide
 - RxNorm 83008 bicalutamide

Key:

ATC = Anatomical Therapeutic Chemical Classification (ATC). ATC is an internationally accepted classification system for medicines that is maintained by the World Health Organisation (WHO). TriNetX use the ATC system to code parent drug groups.

RxNorm = RxNorm provides standardised names for clinical drugs and links its names to many of the drug vocabularies commonly used in pharmacy management and drug interaction software. RxNorm is part of Unified Medical Language System (UMLS) terminology and is maintained by the United States National Library of Medicine (NLM). TriNetX use RxNorm for individual drugs.

REVIEWER 2.1

Reviewer #2 (Remarks to the Author):

Thank you for giving me the opportunity to review this manuscript on potential adverse events (reduced fertility) associated with valproate treatment in fertile men. In both psychiatry and neurology, this is an important and urgent issue to address.

AUTHOR 2.1

We are grateful to the reviewer for kindly taking the time to make these comments.

REVIEWER 2.2

I have the following comments:

Comment 1. The section “Main” gives a good introduction to why, restrictions to the use of valproate is problematic and not without potential consequences for the persons with epilepsy and bipolar disorder. Please, use original references e.g., for the declining use of valproate.

AUTHOR 2.2

Thank you for this helpful comment. We have now included original references as advised, both in the main text and in the newly added literature review (Supplementary Table 1), with accompanying text on pages 3-4 of the Human Literature Review, where all original references are cited.

REVIEWER 2.3

Comment 2. Authors mention “A combination of validated disease and prescription codes were used to create three large cohorts of males aged <55 years within the TriNetX Global Collaborative Network” – there are two references: Ref. 15: TriNetX. Welcome to the world’s largest, living ecosystem of real-world data and evidence for the life sciences and healthcare industries. Global data, for global health (2023), and Ref 16. Mbizvo, G.K., Bucci, T., Lip, G.Y.H. & Marson, A.G. Morbidity and mortality risks associated with valproate withdrawal in young men and women with epilepsy. *Brain* (2024). In the latter, they recognize that the dataset used was not validated (TriNetX data). Maybe more relevant to describe that although they use data in the present study that has not been validated, previous studies suggests that the validity of the data is sufficient for the purpose and refer to studies describing validation of epilepsy diagnoses e.g., their previous publication Mbizvo GK, Bennett KH, Schnier C, Simpson CR, Duncan SE, Chin RFM. The accuracy of using administrative healthcare data to identify epilepsy cases: A systematic review of validation studies. *Epilepsia*. Jul 2020;61(7):1319

AURTHOR 2.3

This is a helpful suggestion, thank you. We have updated and improved the references on page 4 as follows:

“We used a combination of validated disease and prescription codes⁴⁹⁻⁵² to create three large cohorts of living males aged <55 years within the TriNetX Global Collaborative Network.^{47,48}”

Where 49-52 are:

49 Mbizvo, G. K. et al. The accuracy of using administrative healthcare data to identify epilepsy cases: A systematic review of validation studies. *Epilepsia* 61, 1319-1335, doi:10.1111/epi.16547 (2020).

50 Mbizvo, G. K., Schnier, C., Simpson, C. R., Duncan, S. E. & Chin, R. F. M. Validating the accuracy of administrative healthcare data identifying epilepsy in deceased adults: A Scottish data linkage study. *Epilepsy Res* 167, 106462, doi:10.1016/j.epilepsyres.2020.106462 (2020).

51 Mbizvo, G. K., Simpson, C. R., Duncan, S. E., Chin, R. F. M. & Lerner, A. J. Critical success index or F measure to validate the accuracy of administrative healthcare data identifying epilepsy in deceased adults in Scotland. *Epilepsy Res* 199, 107275, doi:10.1016/j.epilepsyres.2023.107275 (2024).

52 Mbizvo, G. K. et al. Using Critical Success Index or Gilbert Skill score as composite measures of positive predictive value and sensitivity in diagnostic accuracy studies: weather forecasting informing epilepsy research. *Epilepsia*, doi:10.1111/epi.17537 (2023).

And 47, 48 are:

47 TriNetX. Welcome to the world’s largest, living ecosystem of real-world data and evidence for the life sciences and healthcare industries. Global data, for global health (2023). <www.trinetx.com>.

48 Palchuk, M. B. et al. A global federated real-world data and analytics platform for research. *JAMIA Open* 6, ooad035, doi:10.1093/jamiaopen/ooad035 (2023).

We also include a discussion on the diagnostic accuracy validation aspects as follows:

Page 11, discussion:

“Another limitation of this study relates to the potential inaccuracies of healthcare coding. From a diagnostic accuracy perspective, we mitigated this by ensuring we used codes that have been validated previously as reliably able to identify epilepsy or bipolar disorder within healthcare datasets.^{49-52,84} This helps to minimise diagnostic misclassification risks. For example, it is possible the use of two prescriptions of valproate and disease coding for epilepsy or bipolar disorder might have generated false positives by misclassifying cases as disease-positive when clinicians have subsequently reconsidered the diagnosis as negative. However, the diagnostic accuracy studies have generally shown that the false-positive rate for epilepsy disease and symptom coding combined with ASMs or bipolar disease coding is low, generating generally high PPVs of >80%, giving us confidence in the probability of few diagnostic misclassifications having occurred.^{49-52,84}”

Page 15, online methods:

“*Case-ascertainment accuracy*

Studies validating the diagnostic accuracy of ICD-10 coding for epilepsy or bipolar disorder in TriNetX *specifically* are as yet unavailable. However, more widely, there are a substantial number of studies validating the diagnostic accuracy of these codes, including in regions and settings covered by TriNetX.⁴⁹ As we showed previously, a case-ascertainment strategy combining disease coding (G40, including epilepsy and status epilepticus) and/or symptom coding (R56.9) with co-prescribed ASMs is a valid way to accurately capture epilepsy within electronic healthcare datasets, and this spans multiple studies in a variety of healthcare settings and countries.⁴⁹ This strategy tends to achieve figures of 80–90% across positive predictive values (PPVs) and sensitivity estimates, and negative predictive values (NPV) and specificity estimates approach 100%.^{49,50} The corresponding F1 scores are also high at 0.86.⁸⁷ F31 codes are also known to accurately capture bipolar disorder, with PPVs, sensitivities and NPVs reaching 84%, 89%, and 98%, respectively.⁸⁴

REVIEWER 2.4

Comment 3. The authors mention “The comparison groups were propensity-matched for >120 baseline covariates before index.” (Supplementary Table 6: Baseline characteristics matched as covariates). Is there a reference or rationale for the choice of these variables?

AUTHOR 2.4

This is an entirely reasonable question and we have since clarified the text in response, as follows (page 5):

“The comparison groups were propensity-score matched for >140 baseline covariates before index (see Supplementary Tables 2–6 for a full list of these baseline covariates pre- and post-propensity-score matching). Selection of the covariates was primarily informed by prior literature,²⁵⁻⁴² and guided by our clinical experience as a team specialising in neurology (G.K.M., S.E.D., A.G.M.), psychiatry (L.V.W.), fertility medicine (L.N.), and andrology (M.T.M.).”

REVIEWER 2.5

Comment 4. The authors mention (line 243) “As we showed previously, a case-ascertainment strategy combining disease coding (G40, including epilepsy and status epilepticus) and/or symptom coding (R56.9) with co-prescribed ASMs is a valid way to accurately capture epilepsy within electronic healthcare datasets, and this spans multiple studies in a variety of healthcare settings and countries.” Please describe the code “R56.9” – it is not found in the WHO browser (<https://icd.who.int/browse10/2019/en#/R56>) – is there an error here?

AUTHOR 2.5

The reviewer will not find R56.9 in the linked ICD-10 browser, as R56.9 is an ICD-10-CM code for seizures. ICD-10-CM, or the International Classification of Diseases, Tenth Revision, Clinical Modification, is a clinical adaptation of the ICD-10 system widely used in healthcare settings, especially in the United States, and adopted by TriNetX based on its wide use among most of the healthcare organisations in their network. The ICD-10-CM system is approved by the WHO and aligns fully with ICD-10's structure and disease classification conventions. The R56.9 code can be found at the following ICD-10-CM browser: <https://icd10cmtool.cdc.gov/?fy=FY2024>

We have now added further details about the ICD-10-CM system to assist any readers who may also be unfamiliar with this variation. Thank you for highlighting this point. The changes are included below:

Page 15, online methods:

“ICD-10-CM is the International Classification of Diseases, Tenth Revision, Clinical Modification, which is a standardised system that is widely used to code diseases and medical conditions amongst HCOs, particularly in the US.⁵³ It is curated by the Centres for Disease Control and Prevention (CDC).⁵³ ICD-10-CM is approved by the WHO and conforms to the WHO's established ICD structure and conventions for statistical classification of disease.⁵³ Healthcare providers use ICD-10-CM codes when diagnosing patients, which are then uploaded to TriNetX directly to allow study data to reflect real-world, real-time, clinical decisions and diagnoses.^{47,48} Full details about how the WHO and CDC define individual ICD-10-CM codes can be found within their respective reference texts.⁵³”

REVIEWER 2.6

Is it possible to provide information on the distribution of cases identified with the various algorithms for epilepsy: “Epilepsy: at least two G40 (epilepsy) and/or R56 (seizure) ICD-10-CM codes recorded anytime in the record” (G40, R56 and the combination of the two)?

AUTHOR 2.6

Excellent idea. We have now included this as follows, on page 6:

“57,420 (64%) of those exposed to valproate had two or more epilepsy disease codes (ICD-10-CM G40) in their record, 49,092 (55%) had two or more seizure symptom codes (ICD-10-CM R56), and 35,882 (40%) had two or more bipolar disorder codes (ICD-10-CM F31). Of those unexposed to valproate, 244,322 (53%) had two or more epilepsy disease codes (ICD-10-CM G40), 280,963 (60%) had two or more seizure symptom codes (ICD-10-CM R56), and 120,585 (26%) had two or more bipolar disorder codes (ICD-10-CM F31). These proportions indicate that some patients in the cohorts had both epilepsy and

REVIEWER 2.7

Comment 5. The authors describe the dataset “TriNetX is the world’s largest ecosystem of real-word electronic health data, drawn from ~250m patients from >120 healthcare organisations (HCOs) across 19 countries predominantly in North America but also South America, Europe, the Middle East, Africa, and Asia Pacific”. This surely is an impressive dataset, but it is difficult to understand the coverage of the data and who may be missing. As far as I understand, the cohort studied is dynamic, which makes it difficult to replicate findings reported. Is there some way that the authors can log the dataset studied to allow for replication of findings?

AUTHOR 2.7

We appreciate the reviewer’s insights regarding the dynamic nature of the TriNetX platform, which indeed offers both benefits and challenges. We have now reflected this in the limitations section and have now logged the dataset output at submission and after the updated analysis following reviewer comments. Reassuringly, despite a significant increase in database size during this period, the results remained consistent, supporting the robustness and replicability of our findings. The updated text is as follows:

Page 12, limitations discussion:

“We acknowledge that the dynamic nature of data acquisition onto the TriNetX platform serves as both a strength and a limitation. It is a strength that new patients and HCOs are added daily, meaning sample sizes increase with time and results are representative of real-time frontline clinical decisions and progress. Conversely, this growth can make it challenging to replicate exact counts over time. Nonetheless, we expect that trends with a strong biological basis will retain their direction of effect despite the evolving nature of the dataset, and our findings support this expectation as there were no changes in the direction of effect or conclusions drawn between our original search at submission (conducted on 09/06/2024, capturing 606,785 men with epilepsy or bipolar disorder from 120 HCOs; logged here: <https://doi.org/10.6084/m9.figshare.27310551.v1>) and our updated search after peer review (conducted on 24/10/2024, capturing an expanded 633,405 men from 131 HCOs; logged here: <https://doi.org/10.6084/m9.figshare.27310593.v1>).”

REVIWER 2.8

Comment 6. I fully agree with the authors choice to include mortality in their analyses – but the manuscript provides very limited data: “The mortality data are enriched with data sourced from outside healthcare settings to increase coverage” – could the authors elaborate on quality of the mortality information to give the reader some impression on how precise mortality is accounted for? (e.g., mortality vs. persons who move away from the HCO catchment area).

AUTHOR 2.8

We would like to clarify that deceased individuals were intentionally excluded from the analysis in both groups to prevent death from competing with the likelihood of experiencing the study outcome at different rates between study groups. This exclusion avoids potential competitive risk bias, thus maintaining comparability between the groups. In response to the reviewer’s concerns, we have now expanded on the quality of mortality data in TriNetX to help reassure readers of its comprehensive nature. This additional context should mitigate concerns regarding any risk of misclassifying deceased individuals as those who may have moved away from HCOs.

The following text was originally included on page 6:

“Those who had experienced the outcome prior to index were excluded, as were the deceased (minimising the effect of competing risks between death and outcome between comparison groups).”

The following text has now been added, on page 12:

“Further, our results can only be generalised to living men, as the deceased were excluded to avoid unequal competitive risks between death and infertility outcomes between cohorts. The main source for mortality data in TriNetX is electronic health records (EHRs), capturing >4 million provider-supplied deaths. As data sourced from EHRs alone may underestimate deaths due to some occurring outside of healthcare information workstreams, TriNetX source additional mortality data from outside EHRs, including from government (Social Security Administration’s Master Death File, covering 20% of US deaths), private obituaries (covering 40% of US deaths), and private claims (covering 20% of US deaths).⁴⁸ TriNetX report that their mortality data covers ~85% of the US population through these linked strategies.⁴⁷ Therefore, our study is likely to have captured and excluded the vast majority of deaths and sampled the remaining living males adequately between cohorts. As patients move away from HCOs in the TriNetX catchment, their medical record ends and censorship occurs after the last clinical data point. This is differentiated from situations where the clinical record has ended due to death having happened as information about the death occurring is usually available from one of the external mortality data sources described above. The mortality data derived from EHRs are refreshed every two to four weeks on average, and data from third party sources are updated monthly.”

REVIEWER 2.9

Comment 7, the authors describe male infertility and how this was captured using ICD10 coding. “There are no studies of the diagnostic accuracy of ICD-10 coding for male infertility, although a study of ICD-9 coding indicates that such codes are likely to be fairly accurate” – although based on ICD-9 coding, the reference provides measures of the validity (as opposed to “fairly accurate”) e.g., reference 23 describes a positive predictive value of 85%. However, studies suggest that infertility rates in e.g., the Nordic countries approach 5% (<https://nhwstat.org/populations/fertility>). The algorithm identified Infertility outcomes in men (up to 1%), but although the validity of the diagnoses used to identify infertility may be high, the algorithm may not capture all men with sub-optimal fertility. Would it be possible include treatment for infertility in the analyses (e.g. intrauterine insemination) to more fully capture male infertility?

AUTHOR 2.9

We thank the reviewer for these helpful points. Whilst it is possible that the algorithm may not capture all men with sub-optimal fertility, it is important to note that comparing people with epilepsy or bipolar disorder directly to a general population in terms of real-world infertility rates is challenging. This is because, unfortunately, people with epilepsy or bipolar disorder suffer substantial stigma surrounding their conditions, which frequently leads to concerns about passing on the epilepsy or bipolar disorder to their children. This, coupled with feelings of shame or hesitation about seeking fertility support, may discourage them from pursuing testing and treatment altogether. We have now incorporated all these points into the text, and clarified the need for future work to consider integrating e.g. intrauterine insemination as an outcome, which was not possible in this dataset. The text now reads as follows, on page 10:

“We studied male infertility, testicular hypofunction, and testicular atrophy as defined and captured by the ICD-10-CM coding system, which conforms to WHO's established ICD structure and conventions for statistical classification of disease.⁵³ We also used established WHO 2021 thresholds for low sperm concentration, motility, vitality, normal forms, and semen volume.⁵⁸⁻⁶⁰ However, ultimately, male infertility is most accurately captured by the success or failure of conception with a female partner known to be fertile, and this information was not directly available for study. Therefore, this study, alongside the majority of prior studies in the valproate field (which have themselves also tended only to measure semen parameters and reproductive hormones – see in Supplementary Table 1)^{25,26,28-30,32,34-38} are limited to assessment of surrogate markers of infertility when considering valproate exposure. Beyond some case reports,^{31,39-42} few studies^{27,33} have actually included an assessment of birth rate or conception success on valproate. However, in our study, it remains reasonable, *clinically*, to assume that the frontline physicians who assigned patients an ICD-10-CM code of N46 (male infertility) might have had supportive information, such as a documented history of conception failure with a fertile partner, and that this helped inform their coding decision. Nevertheless, we would not be able to verify this without direct access to clinical records, which is a recognised limitation for all largescale healthcare data studies. Indeed, the absolute infertility rates in both groups of our study were perhaps lower than one might expect in the general population.⁶⁸ This could indicate that some infertile men were not captured by the dataset, overall. Alternatively, it could indicate that in real-world settings like those represented by TriNetX data (i.e., where patients are not subject to the selection bias of actively consenting to participate in a fertility study), men with epilepsy or bipolar disorder may be less inclined to seek fertility assessments and treatment. Unfortunately, this may be supported by the evidence that PWE tend to isolate themselves from society, conceal their disease, and avoid marriage and having children owing to stigma.⁶⁹⁻⁷⁶ The reduced marriage rates may be compounded by parental reactions to the diagnosis.⁷⁵ As high as 18% of parents object to the marriage of their own son or daughter to a person with epilepsy, and 11% object to contact of their children with PWE.^{70,72} These negative attitudes are thought to affect marriage to men to a greater degree as epilepsy is associated with fewer employment opportunities, and the perception of economic dependence is traditionally less acceptable for men.^{73,75} Bipolar disorder is also associated with lower marriage rates and a shorter duration of marriage,⁷⁷⁻⁷⁹ with divorce and separation found to be two to three times more likely than in the general population.⁸⁰⁻⁸² These factors may be compounded by general trends for falling fertility intentions and opportunities for young men in the western world.⁸³ If possible, it will be important for future studies to include conception success and birth rate as outcomes to help build a more complete picture of evidence about valproate and male reproductive health. Such further work could try to incorporate, for example, treatments for infertility as outcomes (such as intrauterine insemination) to more fully capture male infertility. This was not possible in TriNetX as male data are not linked to the interventions their female partners receive to treat infertility.”

REVIEWER 2.10

Comment 8, although the validity of the coding for male infertility is high (e.g., positive predictive value over 85%), the authors do not address completeness of the registration (see comment above). The infertility diagnosis is likely given mainly to males who are actively seeking to become a parent. As mentioned, birth rates are lower in men with epilepsy (Artama, M., Isojarvi, J.I. & Auvinen, A. Antiepileptic drug use and birth rate in patients with epilepsy-a population-based cohort study in Finland. *Hum Reprod* 21, 2290-2295 (2006).). It may be reasonable to assume that males with epilepsy (and possible also bipolar disorder) are less likely seeking to become a parent and thus less likely to be identified with one of the outcomes. This is accounted for in this study by comparing the cohorts exposed and unexposed to valproate. However, subfertility in males with epilepsy may relate to the indication for use of valproate e.g., type of epilepsy (generalized vs. focal epilepsy) and type of epilepsy may be associated with fertility (see e.g., <https://doi.org/10.1177/15357597221135717>). Could the authors elaborate on this potential bias and possibly address this in analyses?

AUTHOR 2.10

We acknowledge that the type of epilepsy could influence fertility outcomes and act as a confounder. Thank you for raising this important point. To address it, we have now included type of epilepsy as a baseline covariate, and balanced it across comparison groups using propensity-score matching. This approach will mitigate the potential bias from differences in type of epilepsy between groups. With this adjustment, we believe that concerns about bias from differences in type of epilepsy are reasonably accounted for. The relevant text excerpts are provided below:

Page 5:

“The covariates included age, ethnicity, the epilepsy or bipolar disorder themselves (i.e. matching by the first recorded epilepsy or bipolar disease code), type of epilepsy (by ICD-10-CM G40.- subgroup), all other ASMs...”

Supplementary Tables 2 – 6 include the G40.- codes used to match by type of epilepsy, as follows:

- G40.9 Epilepsy, unspecified
- G40.3 Generalized idiopathic epilepsy and epileptic syndromes
- G40.4 Other generalized epilepsy and epileptic syndromes
- G40.2 Localization-related (focal) (partial) symptomatic epilepsy and epileptic syndromes with complex partial seizures
- G40.8 Other epilepsy and recurrent seizures
- G40.1 Localization-related (focal) (partial) symptomatic epilepsy and epileptic syndromes with simple partial seizures
- G40.A Absence epileptic syndrome
- G40.0 Localization-related (focal) (partial) idiopathic epilepsy and epileptic syndromes with seizures of localized onset
- G40.5 Epileptic seizures related to external causes
- G40.B Juvenile myoclonic epilepsy [impulsive petit mal]

REVIEWER 2.11

Comment 9. The authors address infertility outcomes in men with epilepsy and or bipolar disorder exposed and unexposed to drugs known to cause infertility (Supplementary Table 4). Males who use drugs known to be associated with infertility will be much more likely to be assessed for subfertility and thus be registered with the various outcomes (male infertility). Thus, this has implications for the use of this analysis as a “positive control” for the outcome. Please, elaborate on the limitations of using this outcome as support for the method used.

AUTHOR 2.11

This is a reasonable assumption to make, and we have now acknowledged this issue in the text for readers in the limitations section on page 11, as follows:

“From an outcome accuracy perspective, we validated this by illustrating that the infertility outcomes assessed can be captured in association with other drugs known to cause infertility under the same study conditions. However, we also acknowledge the potential for selection bias in that men who use drugs traditionally associated with infertility are perhaps more likely to be assessed for subfertility and thus registered with the various study outcomes. However, the growing concerns about subfertility on valproate should act to mitigate the potential or magnitude of this selection bias.^{8,14-16,18,19,85,86”}

REVIEWER 2.12

Comment 10. I find that figure 1 is hard to read – is it possible to revise the y-axis (survival probability) to capture potential differences? (e.g., focusing survival on 100 – 95%)? Does the survival curve account for mortality? Would it be possible to add the number of patients followed, e.g., by each of the time points (1,000 days, 2,000 days etc.)? Would absolute figures (%) be more illustrative? (e.g., cumulative incidence accounting for mortality?). I was unable to find Supplementary Figure 1 (line 377).

AUTHOR 2.12

Thank you for these suggestions. We would like to clarify that deceased individuals were excluded from both study groups to prevent death from competing with the primary outcome and potentially biasing the analysis (see Author Response 2.8). This exclusion allows for a balanced comparison between groups by removing differential mortality rates as a confounding factor. Mortality cannot be added as a covariate in TriNetX, so excluding it at recruitment is necessary to maintain comparability.

In line with the reviewer's suggestions, we have revised the y-axis to display percentages. While additional changes (such as cumulative incidence or patient numbers at each time point) are not possible within the constraints of the survival analysis function in TriNetX, we have annotated the figure with each outcome studied, along with hazard ratios, confidence intervals, and Log-Rank p-values. We have also improved the contrast of the x and y axes for better readability. Additionally, we have included text on page 12 discussing limitations in TriNetX regarding patient numbers at each time point. This now includes the following:

“Finally, the survival analyses undertaken in TriNetX do not generate or have figures available for the population at risk at each timepoint across the Kaplan-Meier curve, which is limiting for understanding the progression of risk over time in exposed and unexposed individuals.”

Supplementary Figure 1 is located at the end of the Supplementary Materials.

REVIEWER 3.1

Reviewer #3 (Remarks to the Author):

With the background that the extent to which valproate actually (may) causes impaired male fertility and/or adverse effects on the testes is largely unknown, the manuscript titled “Infertility outcomes in valproate exposed and unexposed men with epilepsy or bipolar disorder: an international cohort study of real-world healthcare data” reports the findings of a retrospective cohort study of “reproductive health” in men with epilepsy or bipolar disorder treated with valproate, using real-world healthcare data from TriNetX. Hence, by measuring lifetime propensity-matched risks of male infertility (ICD10-CM N46), testicular hypofunction (E29.1), testicular atrophy (N50.0), and a composite measure of sperm and semen parameters, the authors depicted a not statistically significant difference between men exposed as compared with those unexposed to valproate (the magnitude of effect was <1% difference to being exposed vs. unexposed to valproate). These data are the largest ever considered focusing on the specific topic of reproductive health from the male standpoint. In this setting, the authors have complied with the recommendations of the ‘Sex and Gender Equity in Research – SAGER – guidelines’, clearly explaining that the retrospective data analysis did only apply to men (as per sex assigned at birth). The terms have been properly used both in the title and the abstract. Overall, the study is well conducted and provides a clinically interesting piece of information, despite several limitations should be mentioned.

AUTHOR 3.1

We are grateful to the reviewer for kindly providing this helpful summary of the study.

REVIEWER 3.2

Major issues:

1) Over the materials and methods, the authors state that they queried TriNetX to analyze men under 55 years of age. Is there a specific reason why this cut-off was chosen? It would be interesting to explore the outcomes of interest at different age ranges, as aging is a confounding factor.

AUTHOR 3.2

Although looking at all ages would be an interesting approach more broadly, we have focused this particular study on the age group at which regulatory restrictions have been enforced for men on valproate, which is those under the age of 55 years (see e.g.: <https://tinyurl.com/55j6cu8b>). This makes our study directly relevant to current affairs. It will, nevertheless, be interesting for further work to be undertaken on those 55 years and over in future. We have now acknowledged this in the text. Furthermore, age is, of course, a confounding factor. We have, therefore, addressed age as a confounder by including it as a baseline covariate in our propensity-score matching, balancing it across cohorts.

The relevant changes in the text are as follows:

Page 12 (limitations section):

“Our results can be generalised to men aged <55 years alone. This cut-off was chosen to mirror the age cut-off given by regulators for restrictions to valproate prescribing, which do not apply to men aged ≥ 55 years.^{14,15,85} Age is an important confounder and was, therefore, propensity-score matched between cohorts in this study, thereby balancing it across the cohorts. However, future studies should still aim to independently assess infertility outcomes in men aged ≥ 55 years on valproate, as well as to compare outcomes between men at different age ranges.”

REVIWER 3.3:

Additionally, the Authors should explain why they chose to require at least two diagnostic codes for epilepsy or bipolar disorder, rather than a single code. This decision could impact the selection of participants and therefore the outcome itself.

AUTHOR 3.3

This is a helpful point raised. We have now fully clarified reasoning behind this decision in the text at the following locations:

Page 5:

“The index event was defined as the second recorded valproate prescription in participants exposed to the drug, alongside the presence of at least two epilepsy or bipolar disorder codes, or the second recorded epilepsy or bipolar disease code for those unexposed to valproate. The rationale for using the second prescription was to ensure repeat valproate exposure, providing a more robust indication of chronic use rather than incidental or one-off exposure. We then used the proportion of the cohort with detectable serum valproate levels during follow-up to infer ongoing exposure and adherence. Requiring two diagnostic codes for epilepsy or bipolar disorder allowed us to minimise the risk of misdiagnosis by confirming the condition through repeated coding. Additionally, this approach enabled us to more accurately balance cohorts based on diagnosis, as the first recorded epilepsy or bipolar disorder code was then used as a matching covariate in the propensity-score matching process

(including matching by type of epilepsy using the respective G40.- subgroups, as listed in Supplementary Tables 2–6)). This approach was designed to help improve cohort comparability and reduce the potential of bias from misclassification.”

Page 17:

Cohorts were propensity-score matched for all of the baseline covariates listed in Supplementary Tables 2–6 at the start follow-up (time 0: corresponding to the index event). The index event was taken as the second recorded valproate prescription in those exposed to the drug, or the second recorded epilepsy or bipolar disease code in those unexposed to the drug. This allowed cohorts to be propensity-score matched for epilepsy and bipolar disorder using the first code to appear for these conditions. It also ensured men with only a single exposure to the valproate or single seizure were excluded.

REVIEWER 3.4

2) The Authors did adjust their analyses for testosterone use, anabolic steroid use, chemotherapy treatments etc. via propensity score matching (PSM). As such, these factors should not be part of the PSM but rather should be used as exclusion criteria. The fact that two cohorts are statistically similar in their use of anabolic steroids does not exclude the influence of anabolic steroids on the outcome (e.g., azoospermia or development of severe oligospermia). Instead, the model should be adjusted for non-modifiable risk factors, such as age. This Reviewer considers that this methodological issue significantly impacts the validity of the reported findings.

AUTHOR 3.4

We appreciate the reviewer’s insights on this matter. We have chosen to propensity-score match the cohorts across a broad range of >140 variables that may confound infertility outcomes, encompassing both modifiable and non-modifiable factors (see Supplementary Tables 2-6). In our view, a limitation in the current valproate literature (see Supplementary Table 1) is that strict exclusion criteria have reduced the applicability of these studies to real-world patient groups –particularly the multi-morbid or "imperfect" patients we see in clinical practice who seek information on valproate safety. Our pragmatic approach of propensity-score matching for this comprehensive set of variables is, therefore, intended to enhance the real-world generalisability of this study, which uses real-world healthcare data, taken directly from the frontline. Stringent exclusions would limit that generalisability. Nevertheless, in response to the reviewer’s concerns, which are duly acknowledged, we have now also included a sensitivity analysis that excludes men with key modifiable treatment and lifestyle risk factors for infertility at baseline. This addition provides readers with both approaches: our real-world-focused analysis and the reviewer’s more controlled experimental design. Importantly, the results and conclusions remain consistent across both analyses. Excerpts of the changes made are reported below:

Page 6

“We undertook a sensitivity analysis for the cohort of men with epilepsy or bipolar disorder in which we excluded those with key modifiable risk factors for infertility (including chemotherapy, androgens and anabolic steroids, urogenital surgeries, smoking, alcohol and recreational substance misuse (see full list of factors excluded in the online methods section) and, instead, we propensity-score matched them for the non-modifiable risk factors predominantly (including age, ethnicity, and various comorbidities – see full list in Supplementary Table 5).

Page 14

A sensitivity analysis was undertaken to establish outcomes in men with epilepsy or bipolar disorder exposed and unexposed to valproate following exclusion of those with key modifiable risk factors for infertility as follows:

Search Four: epilepsy or bipolar disorder cohort excluding those with key modifiable risk factors for infertility

- A. *Epilepsy*: at least two G40 (epilepsy) and/or R56 (seizure) ICD-10-CM codes recorded anytime in the record
- B. *Bipolar disorder*: at least two F31 (bipolar disorder) codes recorded anytime in the record
- C. Not deceased
- D. Cannot have: finasteride (RxNorm 25025), ketoconazole (RxNorm 6135), trimethoprim (RxNorm 10829), nitrofurantoin (RxNorm 7454), erythromycins/macrolides (VA:AM200), aminoglycosides (VA:AM300), antineoplastics (VA:AN000), androgens/anabolics (VA:HS100), progestins (VA:HS800), estrogens (VA:HS300), spironolactone (RxNorm:9997), cyproterone (RxNorm:3014), Radiation Oncology Treatment (CPT:1010843), Surgical Procedures on the Urinary System (CPT:1008061), Repair initial inguinal hernia, age 5 years or older (CPT:1008011), Surgical Procedures on the Male Genital System (CPT:1008470), Alcohol related disorders (ICD-10-CM F10), Opioid related disorders (ICD-10-CM F11), Cannabis related disorders (ICD-10-CM F12), Sedative, hypnotic, or anxiolytic related disorders (ICD-10-CM F13), Cocaine related disorders (ICD-10-CM F14), Other stimulant related disorders (ICD-10-CM F15) Hallucinogen related disorders (ICD-10-CM F16), Nicotine dependence (ICD-10-CM F17), Inhalant related disorders (ICD-10-CM F18), Other psychoactive substance related disorders (ICD-10-CM F19)
- E. (A or B) + C + D
- F. Valproate (any dose) prescribed on two or more occasions in the record
- G. Valproate never prescribed
- H. E + either F or G

“Our sensitivity analysis Search Four yielded a cohort of 311,520 men with epilepsy or bipolar disorder after excluding those with key modifiable risk factors for male infertility, including androgens, chemotherapy, urogenital surgery, and lifestyle factors such as smoking and excess alcohol. Following this, there were 32,310 and 279,210 men exposed and unexposed to valproate, respectively. Valproate was detectable in the serum of 97% of the 16,201 unmatched men on valproate who had levels checked (Supplementary Table 10). The propensity-score matching results are shown in Supplementary Table 5. Outcomes were compared between 27,873 matched men in each cohort, with mean ages of 20 ± 12 SD years to 21 ± 13 SD years. Results are summarised in Tables 3 (infertility outcomes) and 4 (reproductive hormones). There remained no significant differences between men with epilepsy or bipolar disorder who were exposed and unexposed to valproate in terms of male infertility (HR 1.479; 95% CI 0.868 to 2.522), testicular hypofunction (HR 1.054; 95% CI 0.728 to 1.526), testicular atrophy (HR 2.110; 95% CI 0.559 to 7.957), and low sperm concentration, motility, vitality, normal forms, or semen volume (HR 1.594; 95% CI 0.598 to 4.252). There were also no differences in total or free testosterone, LH, FSH, prolactin, or estradiol levels between those exposed and unexposed to valproate with bipolar disorder ($p > 0.05$).

REVIEWER 3.5

3) The results section does not provide information on the duration of valproate exposure or dosage. This is more than crucial. Despite this Reviewer doubts this information can be found on TriNetX, this should be listed over the limitations section.

AUTHOR 3.5

The reviewer is correct, this information was not readily available to us from TriNetX. We have now updated the limitations on page 12 to highlight this, as follows:

“Although we had a minimum requirement of valproate exposure on at least two separate occasions and were able to demonstrate detectable levels of valproate in the serum of 94% to 97% of men who had this tested, inferring a good level of drug adherence in these cohorts, we were unable to provide additional data on the dose or duration of treatment as these were not available. Future work should aim to explore dose-response for infertility outcomes on valproate, e.g. using a raw exposure dataset purchased from TriNetX or CPRD.⁶⁷”

REVIEWER 3.6

Moreover, the Authors did not provide information on the timing of infertility, oligospermia, testicular dysfunction etc. diagnoses relative to valproate exposure.

AUTHOR 3.6

The reviewer raises an interesting point, which we have now discussed in more detail in the limitations section. This now includes the following, on page 12:

“We were also unable to report the timing of outcome measurement relative to the start of valproate treatment. However, this is perhaps somewhat an arbitrary issue because the start of male infertility, biologically, may develop much earlier during the course of a man’s treatment and become evident only when conception is tried and fertility testing is undertaken. It would be difficult to set up a study able to capture the start of infertility itself in men exposed to valproate as testing their fertility repeatedly over the course of their treatment is unlikely to be feasible.”

REVIEWER 3.7

4) Overall, the use of coding can limit the validity of results. Among other reasons, It cannot assure adherence to and duration of treatment (up to two prescriptions is recorded).

AUTHOR 3.7

We appreciate the reviewer’s observation regarding coding limitations in healthcare data studies. In this study, however, we were able to access laboratory results showing detectable serum valproate levels in nearly all men tested (94–97%), which suggests good adherence within this population. This is a unique advantage of our setup in TriNetX, as most healthcare data studies lack drug-level data.

We acknowledge that accurately capturing treatment duration is challenging. To address this, we required at least two prescriptions to exclude individuals with only incidental or one-off valproate use.

The drug level and adherence clarifications are noted in the text (see Author Response 3.5, above).

We also include specific details about the drug levels for each analysis, as follows:

Page 6, in regard to men with epilepsy or bipolar disorder:

“Valproate was detectable in the serum of 96% of the 45,036 unmatched men on valproate who had levels checked (Supplementary Table 7), inferring a good level of drug adherence.”

Page 7, in regard to men with epilepsy:

“Valproate was detectable in the serum of 97% of the 31,293 unmatched men on valproate who had levels checked (Supplementary Table 8), inferring a good level of drug adherence.”

Page 7, in regard to men with bipolar disorder:

“Valproate was detectable in the serum of 94% of the 16,654 unmatched men on valproate who had levels checked (Supplementary Table 9).”

Page 8, in regard to men with epilepsy or bipolar disorder after excluding those with key modifiable risk factors for male infertility, including androgens, chemotherapy, urogenital surgery, and lifestyle factors such as smoking and excess alcohol:

“Valproate was detectable in the serum of 97% of the 16,201 unmatched men on valproate who had levels checked (Supplementary Table 10).”

REVIEWER 3.8

5) It is not clear what coding N46.8 (Other male infertility), N46.9 (Male infertility, unspecified), and 606 (Infertility, male); Testicular hypofunction (E29.1) actually mean. How is Testicular atrophy (N50.0) defined? It is crucial to understand how testicular dysfunction is classified within the TriNetX system. More detailed information is required regarding the criteria used to define testicular dysfunction, including which hormones were measured, the threshold values used for classification, and how these measurements were recorded and extracted from the TriNetX database.

AUTHOR 3.8

We appreciate the reviewer’s attention to the coding details. All ICD-10-CM codes, including the ones mentioned by the reviewer (N46.8, N46.9, 606, and E29.1) are WHO-defined and approved and used internationally to classify male infertility, testicular hypofunction, and testicular atrophy. Each code can be fully explored online through ICD-10-CM coding browsers (e.g., N46.8 for "Other male infertility" here - <https://icdlist.com/icd-10/N46.8> and N46.9 for "Male infertility, unspecified" – here <https://icdlist.com/icd-10/N46.9> for N46.9). These codes are applied by frontline clinicians based on their assessment of the patient, and TriNetX simply collects and aggregates these codes from the participating healthcare organizations (HCOs). TriNetX itself does not perform measurements or define ICD-10 coding criteria; it relies on the clinical coding performed within each HCO.

In line with the reviewer’s suggestion, we have now expanded on the meaning and definitions of these codes within the study text for readers who may not have immediate access to online resources. Additionally, we have provided a brief overview of the ICD-10-CM system and the TriNetX platform to clarify how data is pooled from HCOs. Relevant excerpts are included below:

Page 4:

“We studied data from a global network of healthcare organisations (HCOs) held in the TriNetX platform.^{47,48} Clinical data, including disease coding, drug prescribing, and laboratory results are anonymised and uploaded directly from frontline clinical care onto the TriNetX platform in real-time.”

Page 4:

“Lifetime propensity-score matched risks of the following outcomes were assessed between those exposed and unexposed to valproate: **i) male infertility (as defined by the World Health Organisation (WHO) through the International Classification of Diseases 10th Revision Clinical Modification (ICD-10-CM) code N46,⁵³** – which includes N46.0 (Azoospermia), N46.1 (Oligospermia), N46.8 (Other male infertility – capturing where male infertility has been identified but the cause does not fit into any of the other specified (coded) aetiological categories),⁵⁴ N46.9 (Male infertility, unspecified – capturing where male infertility that has been identified but the cause is unclear),⁵⁵ and 606 (Infertility, male – which is the ICD-9-CM code equivalent of N46, allowing healthcare systems using older coding to still be represented);⁵³ **ii) Testicular hypofunction (ICD-10-CM code E29.1)** – which includes defective biosynthesis of testicular androgen, 5-delta-reductase deficiency (with male pseudohermaphroditism), and testicular hypogonadism;⁵⁶ **iii) Testicular atrophy (ICD-10-CM code N50.0** – a code which clinicians have the discretion to use when finding evidence of a pathological reduction in size of the testicles, e.g. using an orchidometer or ultrasound);^{57”}

Page 15:

“ICD-10-CM is the International Classification of Diseases, Tenth Revision, Clinical Modification, which is a standardised system that is widely used to code diseases and medical conditions amongst HCOs, particularly in the US.⁵³ It is curated by the Centres for Disease Control and Prevention (CDC).⁵³ ICD-10-CM is approved by the WHO and conforms to the WHO’s

established ICD structure and conventions for statistical classification of disease.⁵³ Healthcare providers use ICD-10-CM codes when diagnosing patients, which are then uploaded to TriNetX directly to allow study data to reflect real-world, real-time, clinical decisions and diagnoses.^{47,48} Full details about how the WHO and CDC define individual ICD-10-CM codes can be found within their respective reference texts.⁵³

REVIEWER 3.9

Overall, the Authors have not provided information on the specific hormonal profiles in their results.

AUTHOR 3.9

This is a very helpful suggestion, which would add to the completeness of our study. We have now added the most commonly measured reproductive hormone profiles to the results. These are included in results tables 2 and 4, and the accompanying text as follows:

Page 5:

“(v) **laboratory results for the following reproductive hormone levels in serum: total testosterone (ng/dL), free testosterone (pg/mL), luteinizing hormone (LH, mIU/mL), FSH (mIU/mL), prolactin (ng/mL), and estradiol (pg/mL).**^{25,28-30,32-36,38,42} For hormone profiles, the most recent laboratory result reported during follow-up was used, averaged across the cohort.”

Page 6:

“T-tests were used to compare mean laboratory results for total testosterone, free testosterone, LH, FSH, prolactin, and estradiol, using a 0.05 level of significance.”

Page 7: in regard to men with epilepsy or bipolar disorder:

“Mean serum levels for total testosterone were significantly less in those exposed to valproate (360 ± 265 SD ng/dL) compared to those unexposed to valproate (389 ± 272 SD ng/dL, $p = 0.000$). However, both results remained within a normal range for this test (300 to 1,000 ng/dL).⁶² There were no differences in free testosterone, LH, FSH, prolactin, or estradiol levels between those exposed and unexposed to valproate ($p > 0.05$).

Page 7: in regard to men with epilepsy

“Mean serum levels for total testosterone were significantly less in those exposed to valproate (334 ± 254 SD ng/dL) compared to those unexposed to valproate (379 ± 260 SD ng/dL, $p = 0.000$). However, both results remained within a normal range for this test (300 to 1,000 ng/dL).⁶² There were no differences in free testosterone, LH, FSH, prolactin, or estradiol levels between those exposed and unexposed to valproate ($p > 0.05$).

Page 8: in regard to men with bipolar disorder

“There were no differences in total or free testosterone, LH, FSH, prolactin, or estradiol levels between those exposed and unexposed to valproate ($p > 0.05$).

Page 8: discussion

“Lower total testosterone levels are seen in men exposed to valproate, particularly those with epilepsy, but the levels are not generally low enough to be outside of a normal range. The remaining reproductive hormone levels assessed, including free testosterone, did not change.”

REVIEWER 3.10

6) It is well known that oligospermia is not a surrogate for male infertility (also according to WHO indications); as a whole, it must be clearly discussed that evaluating semen parameters in itself represents only a surrogate for spermatogenetic health, but not for reproductive outcomes; indeed, fertility should be considered as a successful reproduction as measured by a live birth and not with semen/sperm parameters only; this issue deserves to be comprehensively discussed in the manuscript; therefore, the main limitation of this study is the outcomes selected by coding which cannot realistically capture male infertility.

7) The authors do not sufficiently address the potential limitations of using ICD-10 codes for identifying infertility outcomes, particularly given the complex nature of male infertility diagnoses. Limitation section should be definitely implemented.

AUTHOR 3.10

The reviewer makes interesting linked points here, that are, in effect, relevant to almost the entire body of evidence surrounding valproate and reproductive health in men. As the reviewer will see from our structured review of the literature,

that has now been added following recommendation to do so from another reviewer, nearly all of the studies commenting on infertility with valproate use in men measure what this reviewer would consider only surrogate markers. The entire field needs to be improved, and we hope our paper is the first to help guide that by discussing these issues and making the limitations of such approaches clear. The relevant text excerpts are included below:

Supplementary Table 1 includes a structured summary of the prior human literature.

Page 3, introduction:

Human literature review

“The extent to which valproate actually causes impaired male fertility and/or adverse effects on the testes is largely unknown.^{19,20} The majority of evidence is either in animal models or on small numbers of humans.^{19,20} It is unclear how applicable the results from animal models are to the human experience. Indeed, some of those animal studies have had to use valproate doses seven to 33 times the maximum human-equivalent dose to demonstrate effects such as testicular atrophy and reversible abnormalities in sperm count or motility.^{19,21-23} The advice given to humans cannot be based entirely on animal models. High-quality grade I–II human studies are required (Oxford 2011 Levels of Evidence),²⁴ but are as yet unavailable. Supplementary Table 1 provides a structured overview of the available human studies, outcomes, limitations, and levels of evidence.²⁵⁻⁴² All are observational, spanning grades III–V, with most being case reports or unmatched case-control designs with small numbers of up to 32 valproate-exposed cases and 90 healthy controls. There are no studies using methods from which causality can be more confidently inferred from observational data, such as propensity-score matching, inverse probability weighting, or target trial emulation.⁴³⁻⁴⁵

Some of the studies report adverse effects of valproate on the sexual and reproductive health of men, describing lower sperm count or concentration,^{29-31,39-42} percentage normal sperm,^{27,30,35,38-42} sperm motility,^{26,29-31,33,37,39-42} semen volume,³⁵ and testicular volume.^{30,32} However, other studies do not report lower sperm count or concentration,^{32,33,35,37,38} percentage normal sperm,²⁹ sperm motility,³⁵ semen volume,^{30,33,37,41} and testicular volume.^{34,37} Many of the studies focus on blood hormone levels, but there is variation in findings between studies for the same hormones (see in Supplementary Table 1),^{25,28-30,32-36,38,42} and hormonal profiles do not necessarily correlate with male sexual health and fertility.²⁰ Furthermore, epilepsy, in and of itself, can affect fertility rates, which are two-thirds lower in men with epilepsy than without.⁴⁶ Therefore, it is difficult to be certain if any reduced fertility seen in men on valproate against healthy controls is related to the valproate or the epilepsy. Indeed, other co-prescribed ASMs may be associated with impairment in semen, testicular, or infertility outcomes,^{27,28,32,34-38} indicating a need for these to be adjusted for statistically before making inferences about the independent effect of valproate on these outcomes; but such small studies have precluded adjustment.^{25,26,28-39} Only one study has directly assessed fertility through birth rate.²⁷ This retrospective cohort of claims and registry data in Finland is the largest reported, assessing PWE on valproate (overall $n = 1,546$, monotherapy $n = 1,116$), carbamazepine (overall $n = 2,689$, monotherapy $n = 2,365$), and oxcarbazepine (overall $n = 832$, monotherapy $n = 631$). Comparisons were made to PWE untreated ($n = 2,714-2,785$), and people without epilepsy ($n = 13,378-13,689$). Authors concluded that birth rate was decreased among PWE on ASMs in general, more so in men. Among men with epilepsy, not only valproate, but also carbamazepine and oxcarbazepine were all associated with low birth rate compared to in men without epilepsy. However, only oxcarbazepine retained a reduced birth rate amongst treated PWE compared with untreated PWE. In a study of 17 infertile men on valproate who were switched to levetiracetam ($n = 9$) or lamotrigine ($n = 8$), three were subsequently able to conceive.³³ However, conception occurred whilst the men were still on a reducing regimen of valproate, making it difficult to confidently infer an association between valproate and infertility. Finally, there is only one reported study of reproductive abnormalities in men on valproate with bipolar disorder.²⁵ This is reported as a full-length article assessing reproductive hormones in 18 men in Turkey with bipolar disorder on valproate monotherapy or in combination with lithium, 15 men with epilepsy on valproate monotherapy, and 21 men with bipolar disorder on lithium monotherapy.²⁵ Valproate did not negatively impact reproductive hormones in the men with bipolar disorder. Elevated levels of prolactin and follicle-stimulating hormone (FSH) were found in men with epilepsy. This was attributed to the epilepsy itself by the study authors. They wrote a follow-up Letter to the Editor assessing semen parameters in 12 men from the study who had consented to this testing: six men with bipolar disorder on lithium monotherapy, five men with bipolar disorder on valproate monotherapy or in combination with lithium, and one man with epilepsy on valproate monotherapy.²⁶ They found reduced sperm count and motility in two out of five men with bipolar disorder on valproate monotherapy or in combination with lithium, and reduced sperm count in the one man with epilepsy on valproate monotherapy. It is challenging to know if any associations, if present, were related to bipolar disorder, epilepsy, lithium, valproate, or a combination of these. A much larger study is needed to help account for these and other potential confounders (covariates) at baseline, allowing more confident inferences to be drawn about any associations.”

Page 10, limitations:

“We studied male infertility, testicular hypofunction, and testicular atrophy as defined and captured by the ICD-10-CM coding system, which conforms to WHO's established ICD structure and conventions for statistical classification of disease.⁵³ We also used established WHO 2021 thresholds for low sperm concentration, motility, vitality, normal forms, and semen volume.⁵⁸⁻⁶⁰ However, ultimately, male infertility is most accurately captured by the success or failure of conception with a female partner known to be fertile, and this information was not directly available for study. Therefore, this study, alongside the majority of prior studies in the valproate field (which have themselves also tended only to measure semen parameters and

reproductive hormones – see in Supplementary Table 1)^{25,26,28-30,32,34-38} are limited to assessment of surrogate markers of infertility when considering valproate exposure. Beyond some case reports,^{31,39-42} few studies^{27,33} have actually included an assessment of birth rate or conception success on valproate. However, in our study, it remains reasonable, *clinically*, to assume that the frontline physicians who assigned patients an ICD-10-CM code of N46 (male infertility) might have had supportive information, such as a documented history of conception failure with a fertile partner, and that this helped inform their coding decision. Nevertheless, we would not be able to verify this without direct access to clinical records, which is a recognised limitation for all largescale healthcare data studies. Indeed, the absolute infertility rates in both groups of our study were perhaps lower than one might expect in the general population.⁶⁸ This could indicate that some infertile men were not captured by the dataset, overall. Alternatively, it could indicate that in real-world settings like those represented by TriNetX data (i.e., where patients are not subject to the selection bias of actively consenting to participate in a fertility study), men with epilepsy or bipolar disorder may be less inclined to seek fertility assessments and treatment. Unfortunately, this may be supported by the evidence that PWE tend to isolate themselves from society, conceal their disease, and avoid marriage and having children owing to stigma.⁶⁹⁻⁷⁶ The reduced marriage rates may be compounded by parental reactions to the diagnosis.⁷⁵ As high as 18% of parents object to the marriage of their own son or daughter to a person with epilepsy, and 11% object to contact of their children with PWE.^{70,72} These negative attitudes are thought to affect marriage to men to a greater degree as epilepsy is associated with fewer employment opportunities, and the perception of economic dependence is traditionally less acceptable for men.^{73,75} Bipolar disorder is also associated with lower marriage rates and a shorter duration of marriage,⁷⁷⁻⁷⁹ with divorce and separation found to be two to three times more likely than in the general population.⁸⁰⁻⁸² These factors may be compounded by general trends for falling fertility intentions and opportunities for young men in the western world.⁸³ If possible, it will be important for future studies to include conception success and birth rate as outcomes to help build a more complete picture of evidence about valproate and male reproductive health. Such further work could try to incorporate, for example, treatments for infertility as outcomes (such as intrauterine insemination) to more fully capture male infertility. This was not possible in TriNetX as male data are not linked to the interventions their female partners receive to treat infertility.”

REVIEWER 3.11

Other comments

1) The Authors stated “The comparisons were made using chi-squared tests for categorical variables and independent-sample t-tests for continuous variables”. This Reviewer does assume data distribution was tested and statistical analysis performed accordingly.

AUTHOR 3.11

We thank the reviewer for querying this. We now include the following clarifications to the statistical methods, on page 16 in response:

“The primary focus of propensity-score matching is to achieve balance between covariates amongst the cohorts. As part of the propensity-score matching process, *p*-values and SMD were generated for each covariate pre- and post-matching between cohorts. We set a SMD threshold of < 0.1 to indicate adequate matching.⁶¹ Uncorrected *p*-values were calculated by T-test for continuous covariates (which were age and BMI) and Z-test for categorical covariates (which were all the rest). In propensity-score matching, T-tests are used to compare covariates between the two groups (pre- and post-matching) to ensure that matching has reduced differences in covariates and, therefore, has made groups comparable. This means the T-test, in this situation, is applied for descriptive purposes to check for cohort covariate balance, rather than for hypothesis testing or drawing inferences about study outcomes between cohorts. Therefore, a formal requirement for normality testing is less relevant. Furthermore, the large sample sizes typical of TriNetX mean the data generally benefit from the Central Limit Theorem, which states that the sampling distribution of the mean tends to be normal as the sample size increases, regardless of the underlying distribution of the data.⁸⁹ In such scenarios, it is generally accepted that the T-test can be used even if data are not perfectly normal, because the large sample size compensates for deviations from normality. The same applies for the comparisons we made of reproductive hormone levels (laboratory results), where T-tests were used to test hypotheses. Although TriNetX did not test for normality here, the Central Limit Theorem allows us to reasonably assume normal distribution and/or compensation for deviations from normality owing to large samples.⁸⁹”

REVIEWER 3.12

2) The censoring approach (removing participants after the last fact in their record) could introduce immortal time bias if not handled correctly. More detail on how this was managed is needed.

AUTHOR 3.12

This is an important point to duly acknowledge as a limitation of TriNetX, which we have now done on page 12, and in our previous TriNetX-based study, which is referenced alongside:

“The possibility of some immortal time biases, however, could not be excluded as censorship occurring after the last clinical data point would not take into account prior intervals during which participants were temporarily deregistered from HCOs in the TriNetX catchment.⁸⁷”

REVIEWER 3.13

3) Matching does not include important risk factors for male infertility such as varicocele and cryptorchidism, as well as

recreational habits (cigarette smoking, alcohol consumption, illicit drug use) which may eventually impact toward sperm parameters.

AUTHOR 3.13

We appreciate the reviewer's attention to these risk factors and apologise if it was unclear that these factors were, in fact, included in our original matching profile on submission. Specifically, the ICD-10-CM codes for these conditions were used as follows:

- **ICD-10-CM I86.1** = Scrotal varices (varicocele)
- **ICD-10-CM Q53** = Undescended and ectopic testicle (cryptorchidism)
- **ICD-10-CM F10-F19** = Mental and behavioural disorders due to psychoactive substance use, covering:
 - F10 Alcohol related disorders
 - F11 Opioid related disorders
 - F12 Cannabis related disorders
 - F13 Sedative, hypnotic, or anxiolytic related disorders
 - F14 Cocaine related disorders
 - F15 Other stimulant related disorders
 - F16 Hallucinogen related disorders
 - F17 Nicotine dependence
 - F18 Inhalant related disorders
 - F19 Other psychoactive substance related disorders

These codes encompass all recreational habits referenced by the reviewer, including smoking, alcohol, and illicit drug use. To further clarify for readers, we have now broken down the matching codes by each specific code within the broader F10-F19 grouping in Supplementary Tables 2, 3, 4, and 6. This should help ensure these factors are more clearly identified within the propensity-score matching profile by readers.

The covariates are also now highlighted in the list on page 5 as follows: "... dependence on nicotine, cannabis, alcohol, opioids, cocaine, and other psychoactive recreational substances, diseases of the genitourinary system, sexually transmitted infections, congenital malformations (including of the male genital organs) and chromosomal abnormalities, scrotal varices, undescended and ectopic testicles..."

REVIEWER 3.14

4) The Authors should use the same units of measure reported in the WHO manual for the examination of human semen (sperm concentration x10⁶/mL)

AUTHOR 3.14

We have now changed the units to exactly as suggested by the reviewer throughout the text.

REVIEWER 3.15

5) The "pooled estimate of standardized differences" is not a common term and may be confusing to readers. Please rephrase.

AUTHOR 3.15

We note this suggestion with thanks, and have re-phrased the sentences to now read more clearly, as follows (page 16):

"The primary focus of propensity-score matching is to achieve balance between covariates amongst the cohorts. As part of the propensity-score matching process, *p*-values and SMD were generated for each covariate pre- and post-matching between cohorts. We set a SMD threshold of < 0.1 to indicate adequate matching.⁶¹"

REVIEWER 3.16

6) In the results section, the Authors omitted reporting the number at risk in their Kaplan-Meier analyses. As far as this Reviewer is aware, TriNetX does not provide numbers at risk in their Kaplan-Meier curves by default; although this holds true, a limitation should be explicitly acknowledged if this applies. Alternatively, if the Authors have access to this information, they should include it to enhance the robustness and clarity of their analysis. To this aim, the Authors could consider exporting the data to spreadsheets and recreating the Kaplan-Meier curves using third-party statistical software.

7) The authors do not report Kaplan-Meier risk estimates for specific time points, which is a significant omission even if the results are not statistically significant. These estimates provide important information about the progression of risk over time for exposed vs. unexposed and the outcomes of interest.

Lastly, the inclusion of confidence intervals in the Kaplan-Meier curves is needed.

AUTHOR 3.16

We thank the reviewer for making these important and related points. The reviewer is correct, it is not possible to add these factors to the Kaplan-Meier plots in TriNetX, and we do not have access to a raw data spreadsheet to recreate them in the manner suggested. In response, we have now added these limitations to the text, as follows (page 12):

“Finally, the survival analyses undertaken in TriNetX do not generate or have figures available for the population at risk at each timepoint across the Kaplan-Meier curve, which is limiting for understanding the progression of risk over time in exposed and unexposed individuals. Although HRs are available with 95% CIs, CIs cannot be integrated into the Kaplan-Meier curves. A raw TriNetX data extract would allow such flexibility in analysis but would come with a monetary cost for the data acquisition. The findings of our study and their implications would help justify investment in such further work.”

AUTHOR RESPONSES TO REVIEWER QUESTIONS - NCOMMS-24-38348A

Reviewer #1 (Remarks to the Author):

Reviewer 1.1

The authors have revised the manuscript and they have addressed the main concerns raised. I have no additional comments. In my opinion, the paper could be accepted in the present version

Author 1.1

We are grateful to the reviewer for their supportive comments and a positive evaluation.

Reviewer #4 (Remarks to the Author):

Reviewer 4.1

The noteworthy results are that the evidence used by regulators to justify valproate restriction in men is flawed. This is significant as it means that valproate is an effective medication which is denied to men without a sound evidence base. This is a large population-based study and design is superior to other studies. The greatest strength of the study is that it compares men with epilepsy or bipolar disease prescribed valproate versus those not prescribed valproate, reducing the confounders in other studies. The limitations of the study are transparently and well described in the paper.

Author 4.1

We thank the reviewer for this very helpful summary of the study and its potential implications.

Reviewer 4.2

Suggestions:

*Abstract:

This should include the key point about comparison between valproate exposed vs unexposed groups outlined above.

Author 4.2

Thanks. We have now added the following to the abstract:

“The evidence for this comes largely from animal studies, with limited and inconclusive human data. Comparisons have mostly involved men with epilepsy and healthy controls, notwithstanding the confounding effect of epilepsy itself on male fertility. Few studies have examined infertility outcomes in men with bipolar disorder treated with valproate.”

“Our study results do not support an association between valproate use and infertility in men with epilepsy or bipolar disorder. The study is strengthened by its size and robust propensity score matching profile. Comparing valproate-exposed and unexposed men with epilepsy or bipolar disorder in this study has helped to limit the confounding effect that these conditions have on male fertility.”

Reviewer 4.3

The sentences are long and overly complicated and this detracts from the important messages: eg Line 31-34: "Use is becoming increasingly restricted for men owing to widespread warnings about it causing testicular dysfunction and infertility. Most of the existing evidence on this comes from animal studies, with limited and inconclusive data available from human research. " ..this could be simplified to "Regulatory restrictions for men are based on warnings about it causing testicular dysfunction and infertility. Evidence for this comes largely from animal studies, with limited and inconclusive human data."

Author 4.3

This is a very helpful suggestion. We have now amended the abstract text to read as follows:

“Use is becoming increasingly restricted for men owing to widespread warnings about it causing testicular dysfunction and infertility. The evidence for this comes largely from animal studies, with limited and inconclusive human data.”

Reviewer 4.4

Line 305-307: “Any evidence that valproate is linked to male infertility in epilepsy seems to emerge when it is compared to healthy men without epilepsy, which were the controls used in most of the prior studies.” could be simplified to "Prior studies linking valproate to male infertility in epilepsy mainly used healthy men without epilepsy as controls".

Author 4.4

Another helpful suggestion. We are grateful to the reviewer.

The text on page 9 now reads:

“Prior studies linking valproate to male infertility in epilepsy mainly used healthy men without epilepsy as controls.”

Reviewer 4.5

The conclusion of the abstract is very weak compared to the strength of the data: Line 43-44: "Our results suggest more caution is needed in associating valproate use for epilepsy or bipolar disorder with male infertility in humans." should be something like: "Our results do not support a causal association between valproate use for epilepsy or bipolar disorder and male infertility in humans"

Author 4.5

We thank the reviewer for this helpful suggestion and support. We have opted to retain a cautious interpretation by not using the word “causal” here as we did not undertake a target trial emulation with inverse probability weighting, but we have otherwise adopted the reviewer’s suggestion as follows:

“Our study results do not support an association between valproate use and infertility in men with epilepsy or bipolar disorder.”

Reviewer #5 (Remarks to the Author):**Reviewer 5.1**

This manuscript represents a significant effort on authors. Thank you for the many efforts the author team made to update the manuscript based on previous review(s). Specific concerns follow, but my focus on updating this manuscript is on coding concerns, and rate of abnormal semen analysis concerns.

Author 5.1

We thank the reviewer for taking the time to review our work and for their constructive feedback. We have addressed the coding and abnormal semen analysis concerns in the relevant sections below.

Reviewer 5.2

3.2- Appreciate the authors comments and change to manuscript. This reviewer asks authors to add to changes that this age is recommended in UK, not in all countries.

Author 5.2

Thanks. We have now added this to the text on page 12, which reads:

“This cut-off was chosen to mirror the age cut-off given by regulators for restrictions to valproate prescribing in the UK, which do not apply to men aged ≥ 55 years.^{14,15,87”}

Reviewer 5.3

3.3- No concerns with this response.

Author 5.3

Noted with thanks.

Reviewer 5.4

3.4- Appreciate the authors inclusion of further work with regards to sensitivity analysis.

Author 5.4

Noted with thanks.

Reviewer 5.5

3.5- This is a significant concern but agree with previous reviewer- this information likely is not attainable, and authors have updated limitations.

Author 5.5

Noted with thanks.

Reviewer 5.6

3.6- This reviewer remains concerned regarding this issue of timing of these labs and values with respect to use of valproate. Disagree strongly with arbitrary issue below. Sperm for instance is matured over 70-90 days, and it is not clear if semen analyses in these patients are taken shortly after valproate initiation and at later timepoints. Specifically, we have incorporated the intervals suggested by the reviewer (30, 60, 90, 180, and 360 days) and have also included longer-term intervals (2, 5, and 10 years). We are grateful to the reviewer for this very helpful suggestion. To ensure clarity and ease of interpretation, we have aggregated the relevant dichotomous outcomes into a single overall infertility measure over time. This approach has allowed us to explore the longitudinal effects of valproate exposure without overburdening the reader with numerous additional appendices and repeated propensity score matching outputs. Importantly, our conclusions remain unchanged over time.

Author 5.6

We thank the reviewer for highlighting this important issue regarding the timing of outcome measurement in relation to valproate initiation. In response, we have removed the previous reference to this being an arbitrary issue and have revised the analysis to examine outcomes both shortly after initiation and at later timepoints. Specifically, we have incorporated the intervals suggested by the reviewer (30, 60, 90, 180, and 360 days) and have also included longer-term intervals (2, 5, and 10 years). We are grateful to the reviewer for this very helpful suggestion. To ensure clarity and ease of interpretation, we have aggregated the relevant dichotomous outcomes into a single overall infertility measure over time. This approach has allowed us to explore the longitudinal effects of valproate exposure without overburdening the reader with numerous additional appendices and repeated propensity score matching outputs. Importantly, our conclusions remain unchanged over time.

We have added the following to the abstract:

“We measure lifetime risks of infertility, testicular hypofunction, testicular atrophy, and a composite measure of low sperm concentration, motility, vitality, normal forms, and semen volume. An aggregate of these outcomes is also looked at 30, 60, 90, 180, and 360 days after valproate initiation, as well as at 2, 5, and 10 years.”

We have added the following to the Main section (Page 6):

“A survival analysis was undertaken on the fully matched cohorts using Cox-proportional hazard models, generating hazard ratios (HRs) with 95% confidence intervals (CIs), and Log-Rank p-values with a 0.05 level of significance. This was undertaken for the dichotomous outcomes male infertility, testicular hypofunction, testicular atrophy and the composite measure of low sperm concentration, motility, vitality, normal forms, or semen volume. These dichotomous outcomes were also aggregated into a single “overall infertility measure” over time in order to understand the effect of valproate exposure over a lifetime and at intervals of 30, 60, 90, 180, 360 days, 2 years, 5 years, and 10 years after initiation.”

We have added the following to the online methods section (Page 16, final paragraph)

“The first occurrence of the following coded outcomes was measured after index up to the day of the data search (24/03/2025), excluding any persons who had the outcome coded prior to index (see statistical analysis for index specification) for the dichotomous outcomes:

- i) Male infertility (ICD-10-CM code N46);⁵³
- ii) Testicular hypofunction (ICD-10-CM code E29.1);⁵³
- iii) Testicular atrophy (ICD-10-CM code N50.0);⁵³
- iv) A composite measure of low sperm concentration (<16 x 10⁶/ml semen), low sperm motility (<42% total motility), low sperm vitality (<54% alive), low normal forms (<4% of sperm have a normal morphology), and low semen volume (<1.4 mL);⁵⁸⁻⁶⁰
- v) Reproductive hormone levels: total testosterone (ng/dL), free testosterone (pg/mL), LH (mIU/mL), FSH (mIU/mL), prolactin (ng/mL), and estradiol (pg/mL).^{25,28-30,32-36,38,42} The most recent result taken during follow-up was used.

To assess the longitudinal effect of valproate exposure, the dichotomous outcomes (i–iv) were also combined into a unitary infertility measure. This was evaluated at multiple intervals after valproate initiation: 30, 60, 90, 180, and 360 days, and at 2, 5, and 10 years.”

We have added the following to the Main section (page 7):

“Over a lifetime, 104 more men with epilepsy or bipolar disorder exposed to valproate experienced the overall infertility measure compared to those unexposed – a difference of 0.1% in absolute risk, which was not statistically significant (HR 0.932; 95% CI 0.849 to 1.024). No statistically significant differences were observed between exposed and unexposed groups at any of the assessed intervals: 30, 60, 90, 180, and 360 days, as well as 2, 5, and 10 years post-index (Table 1).”

We have also added a new table showing these results (Table 1).

Reviewer 5.7

Also, difficult to follow total N for semen analysis values, but seems risk of abnormal semen analyses was significantly low. Page 7: “Similarly, almost equal numbers of 25 and 23 in those exposed and unexposed to valproate, respectively, experienced a composite low sperm concentration, motility, vitality, normal forms, or semen volume (0.05% risk in both groups).” Authors need to comment on this exceptionally low risk of abnormal SA values in both cohorts. I assume risk of abnormal SA values would approximate 10% in public patients presenting for semen analyses.

Author 5.7

We thank the reviewer for these helpful observations. We have now clarified the total number of men with available semen analysis data in the Discussion (page 11), which reads:

“The number of men aged <55 years with laboratory results available on semen parameters in TriNetX is 65,628 for semen volume, 46,319 for sperm motility, 38,589 for sperm concentration, 3,419 for sperm vitality, and 3,239 for normal forms.”

While the absolute risk of abnormal semen parameters in our study may appear low, this reflects the fact that our estimates are based on the entire population at risk—that is, all men with epilepsy or bipolar disorder are included in the denominator. Naturally, this yields a low proportion of abnormal results across the full cohort. However, this conservative approach enables standardised comparisons between exposure groups and facilitates detection of relative differences in risk, where present. To aid interpretation, we have contextualised our findings using global estimates that similarly adopt a population-at-risk approach.⁶⁴ When aligned to a comparable scale, the observed lifetime risk of the overall infertility measure in our study is 1,202 per 100,000 in the valproate-exposed group and 1,070 per 100,000 in the unexposed group (equivalent to the 1.20% and 1.07% reported in Table 1). These figures are broadly comparable to the worldwide male infertility prevalence reported in the Global Burden of Disease Study as 1,820.6 per 100,000, or 1.8%.⁶⁴

The above text has now been added to the discussion (page 9).

Reviewer 5.8

3.7- This reviewer feels authors have adequately responded to previous comments.

Author 5.8

Noted with thanks.

Reviewer 5.9

3.8- This reviewer feels authors have inadequately responded to previous comments. Codes used do not fully search for male infertility (See Z31.81), or other organic causes of subfertility (Q53.9, Q55.1, O55.3- some examples).

Author 5.9

We thank the reviewer for these thoughtful observations.

With regard to ICD-10-CM code **Z31.81**, we respectfully note that this refers to “*Encounter for male factor infertility in female*,” as confirmed via the CDC’s ICD-10-CM browser (<https://icd10cmtool.cdc.gov/?fy=FY2025>). This code is used specifically for female patients undergoing evaluation related to male partner infertility, and therefore would not be appropriate for identifying male infertility outcomes within a male cohort. The suggested **Q codes**, including **Q53.9** (*Undescended testicle, unspecified*), **Q55.1** (*Hypoplasia of testis and scrotum*), and **Q55.3** (*Atresia of vas deferens*) are part of Chapter 17: *Congenital malformations, deformations and chromosomal abnormalities (Q00–Q99)*, and specifically the **Q50–Q56** category for *congenital malformations of genital organs*. These codes represent structural conditions typically present from birth, and would not be expected to develop as acquired conditions in adulthood. As such, they are not appropriate as outcome measures in a study examining adult-onset infertility potentially associated with a medication exposure such as valproate. However, we agree that these congenital conditions may be relevant as baseline risk factors for infertility. To account for this, we have included the Q53 and Q55 codes as covariates in our propensity score matching process (see Supplementary Tables 2–14).

Reviewer 5.10

3.9- This reviewer feels authors have adequately responded to previous comments.

Author 5.10

Noted with thanks.

Reviewer 5.11

3.10- This reviewer understands previous reviewer comments/concerns. See my concern above at 3.6. Otherwise appreciate the updates authors have made to this request.

Author 5.11

Noted with thanks. We have now addressed the concern in 3.6 above (see section 5.6).

Reviewer 5.11

3.11- This reviewer feels authors have adequately responded to previous comments.

Author 5.10

Noted with thanks.

Reviewer 5.12

3.12- This reviewer feels authors have adequately responded to previous comments.

Author 5.12

Noted with thanks.

Reviewer 5.13

3.13- This reviewer still has concern regarding coding used (See 3.8 above).

Author 5.13

Thank you. We have now clarified the coding issues in our response to comment 3.8 (see section 5.9).

Reviewer 5.14

3.14- This reviewer feels authors have adequately responded to previous comments.

Author 5.14

Noted with thanks.

Reviewer 5.15

3.15- This reviewer feels authors have adequately responded to previous comments.

Author 5.15

Noted with thanks.

Reviewer 5.16

3.16- Appreciate previous reviewer concerns, but as authors point out these changes are not available, so limitations have been added.

Author 5.16

Noted with thanks.